# Higgs Phases and Boundary Criticality

Kristian Tyn Kai Chung[1*], Rafael Flores-Calderón[1,2], Rafael C. Torres[3], Pedro Ribeiro[3]
Sergej Moroz[4,5] Paul McClarty[1,6]

1 Max Planck Institute for the Physics of Complex Systems, Dresden, Germany
2 Max Planck Institute for Chemical Physics of Solids, Dresden, Germany
3 CeFEMA, LaPMET, Instituto Superior Técnico, Universidade de Lisboa, Lisboa, Portugal
4 Department of Engineering and Physics, Karlstad University, Karlstad, Sweden
5 Nordita, KTH Royal Institute of Technology and Stockholm University, Stockholm, Sweden
6 Laboratoire Léon Brillouin, CEA, CNRS, Université Paris-Saclay, CEA Saclay, Gif-sur-Yvette, France

* ktchung@pks.mpg.de

## Abstract

Motivated by recent work connecting Higgs phases to symmetry protected topological (SPT) phases, we investigate the interplay of gauge redundancy and global symmetry in lattice gauge theories with Higgs fields in the presence of a boundary. The core conceptual point is that a global symmetry associated to a Higgs field, which is pure-gauge in a closed system, acts physically at the boundary under boundary conditions which allow electric flux to escape the system. We demonstrate in both Abelian and non-Abelian models that this symmetry is spontaneously broken in the Higgs regime, implying the presence of gapless edge modes. Starting with the U(1) Abelian Higgs model in 4D, we demonstrate a boundary phase transition in the 3D XY universality class separating the bulk Higgs and confining regimes. Varying the boundary coupling while preserving the symmetries shifts the location of the boundary phase transition. We then consider non-Abelian gauge theories with fundamental and group-valued Higgs matter, and identify the analogous non-Abelian global symmetry acting on the boundary generated by the total color charge. For SU($N$) gauge theory with fundamental Higgs matter we argue for a boundary phase transition in the O($2N$) universality class, verified numerically for $N = 2, 3$. For group-valued Higgs matter, the boundary theory is a principal chiral model exhibiting chiral symmetry breaking. We further demonstrate this mechanism in theories with higher-form Higgs fields. We show how the higher-form matter symmetry acts at the boundary and can spontaneously break, exhibiting a boundary confinement-deconfinement transition. We also study the electric-magnetic dual theory, demonstrating a dual magnetic defect condensation transition at the boundary. We discuss some implications and extensions of these findings and what they may imply for the relation between Higgs and SPT phases.

# 1 Introduction

Gauge fields and gauge invariance have a long and complex history in theoretical physics, deeply interwoven with the advent of quantum field theory, the formulation of the Standard Model of particle physics, and firmly embedded in the modern theory of quantum many body systems. In fundamental physics, gauge theories arise naturally in Lorentz covariant theories of massless particles where they resolve a mismatch between the physical degrees of freedom admitted by the Wigner little group—such as photon polarizations—and the vector potential used to describe the particle states. This is accomplished by rendering the surplus degrees of freedom redundant. As such they arise naturally in field theories of gravity, nuclear forces and electromagnetism. In condensed matter physics, gauge fields play a key role in describing a plethora of physical phenomena including highly-entangled emergent states of matter such as spin liquids and fractional quantum Hall fluids.

The redundancy inherent to gauge theories leads to important subtleties. In particular, while the theories are local, the physical gauge-invariant objects are non-local Wegner-Wilson string loops [1,2]. This implies a tension in describing the spontaneous breaking of symmetries in the presence of dynamical gauge fields. Whereas many condensed matter systems demonstrate symmetry lowering phase transitions governed by spontaneous breaking of a global symmetry, gauge redundancy, being unphysical, cannot be broken. This fact, enshrined in Elitzur's theorem [3], belies a rich landscape of different phases separated by phase transitions whose study began systematically in the 1970s—confined, deconfined, Higgs, topologically ordered, etc. While Landau theory successfully accounts for a broad range of phase transitions in correlated many-body systems, the order parameters for theories with dynamical gauge fields are necessarily non-local, raising the question of how to understand the nature of the phase transitions in such theories. Recent advances generalizing notions of symmetries to higher-dimensional charged objects [4] allowed to extend the Landau paradigm to describe such phase transitions [5–7], for a review see [8].

The Anderson-Higgs mechanism [9–12], i.e. the condensation of charged scalar matter in gauge theories, is a cornerstone of physics. It plays a key role in our understanding of superconductivity phenomena and the nature of electroweak interactions within the Standard Model of particle physics. Two closely related questions about this mechanism have stood the test of time and continue to generate significant interest: First, what is the gauge-invariant order parameter characterizing Higgs phases [13]? Second, what is the distinction between the Higgs and confined phases [14]? Let us illustrate this with the 4D compact U(1) gauge theory, i.e. Maxwell theory with magnetic monopoles. In the pure gauge theory there is a deconfined phase at weak coupling in which static electric charges interact via a $1/r$ Coulomb potential mediated by a gapless photon. At strong coupling, the proliferation of magnetic monopoles drives the system into the gapped confined phase, where static electric charges interact via a linearly rising potential. On the other hand, if we couple the gauge field to a charged scalar Higgs field, then the Higgs field may condense, driving a transition from the gapless deconfined phase to a gapped Higgs phase via the Anderson-Higgs mechanism. Can these two gapped regimes be distinguished, and if so by what order parameter? For a Higgs field in the fundamental representation of the gauge group, i.e. one carrying elementary charge, it is generally understood that the Higgs and confined regimes are actually the same

phase, i.e. they are not separated by any thermodynamic bulk phase transition [14, 15]. This Higgs-confinement continuity is believed to be true in generic models with a gauge group $\mathcal{G}$ coupled to a scalar Higgs field in the fundamental representation, including both discrete and continuous Abelian and non-Abelian gauge groups [14].

The question of whether there is a qualitative difference between these two regimes has been revisited time after time from many different perspectives [16]. To survey briefly the history of the endeavor to delineate these two regimes, one approach has been to perform a partial gauge fixing and observe symmetry breaking in an unfixed global subgroup, which shows a phase transition separating them, though the location of the transition line is gauge-dependent and thus lacks a clear physical meaning [17]. Other proposals seek to delineate them in the presence of global symmetries (whose realization is unaltered between the two regimes), see e.g. [18–20]. Yet another approach, advanced partially by one of the authors, emphasizes that (in a certain limit) Abelian Higgs phases with fundamental matter exhibit symmetry-protected topological (SPT) order [21–23]. This observation motivates the investigation of the Higgs mechanism in open geometries, and zooms in on low-energy excitations localized near boundaries. In contrast to the confined regime, where the ground state is unique, in the Higgs regime previous studies uncovered energy spectrum degeneracies, see for example [21–23]. The robustness of these degeneracies originating from boundary-localized modes arises from the interplay of the protecting (generalized) symmetries that depends on the gauge group and dimensionality of the problem. In summary, the presence of a boundary introduces a criterion by which one can delineate the Higgs and confined regimes of Abelian gauge theories with fundamental matter—they are separated by a boundary phase transition.

In this paper we explore in detail boundary symmetry breaking in Wilson lattice gauge theories. We find that the Higgs-confinement boundary criticality mechanism is in fact ubiquitous. We begin in Section 2 by showing the presence of a boundary phase transition in the 4D $U(1)$ Abelian Higgs model, where the magnetic one-form symmetry is broken explicitly. We discuss how, in the presence of boundaries which allow flux but not charge to exit the system, there is a bulk U(1) global matter symmetry which, by the Gauss law, is equivalent to an electric flux symmetry acting on the boundary. We show that in a particular limit of the theory, in which the action reduces to a 3D XY model on the boundary, this boundary U(1) symmetry can be broken spontaneously. We provide numerical evidence that there is a corresponding boundary phase transition in the presence of bulk fluctuations, and we trace the phase boundary in the bulk phase diagram. Next we turn to non-Abelian Higgs theories in Section 3. We consider two types of Higgs models, those with group-valued Higgs fields and those with fundamental representation (vector-valued) Higgs fields, which coincide for gauge group SU(2) but differ for other gauge groups. We demonstrate that these models also have a global charge symmetry which is realized at the boundary. Using large-scale lattice simulation, we first show that 4D SU(2) Higgs theory has a boundary phase transition in the 3D O(4) universality class, verifying our theoretical prediction. We show that this boundary symmetry breaking is expected to be generic in group-valued Higgs models, and provide a general argument that fundamental-Higgs models with gauge group SU($N$) and SO($N$) exhibit O($2N$) and O($N$) boundary criticalities, respectively. We verify this prediction numerically for the case of the 4D SU(3) fundamental-Higgs. Lastly, in Section 4, we consider generalizing to higher-form Abelian-Higgs models, with a $k$-form gauge field coupled to a $(k-1)$-form Higgs field. We discuss how the higher-form matter symmetry is realized at the boundary through the Gauss law, and show that the action reduces in a limiting case to a boundary $(k-1)$-form gauge theory which may exhibit a confinement-deconfinement phase transition in which the matter $(k-1)$-form symmetry is spontaneously broken. In the same section, we perform a duality transformation and discuss how this symmetry breaking can be viewed from the perspective of magnetic defects which live at the boundary. Finally, we provide an overview of our find-

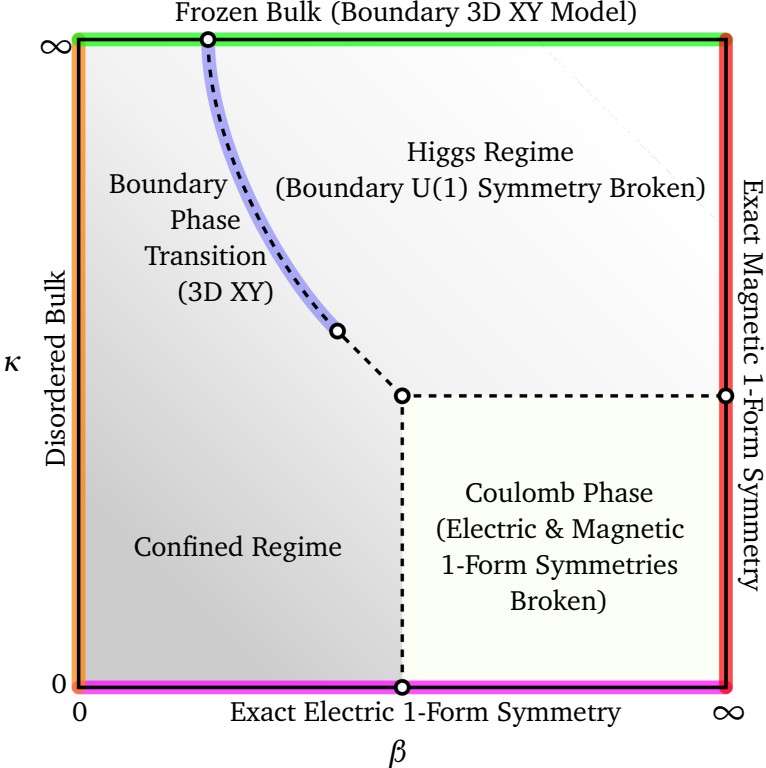

Figure 1: A sketch of the phase diagram of the 4D U(1) lattice Abelian-Higgs phase diagram discussed in Section 2. The confined and Higgs regimes belong to the same thermodynamic phase, though we demonstrate that in presence of symmetry-preserving boundary they are sharply separated by a second order boundary phase transition in the 3D XY universality class. Along the line $\kappa = 0$ the model has an exact electric 1-form symmetry, while along the line with $\beta = \infty$ the model has an exact magnetic 1-form symmetry. The confined regime is smoothly connected to the $\beta = 0$ limit where the gauge field is maximally disordered. Along the $\kappa = \infty$ line, the bulk is completely frozen, and the boundary reduces to a 3D XY model. The boundary $U(1)$ symmetry is spontaneously broken on the Higgs side of the transition line. A review of the structure of this phase diagram is provided in Appendix B.

ings and discuss how our work gives rise to new insights and challenges for the Higgs=SPT proposal.

## 2 Boundary Symmetry Breaking in Abelian Higgs Models

The prototypical theory for the interaction of charges with a gauge field is the U(1) Abelian-Higgs model—i.e. scalar QED, electrodynamics in (3+1)D Lorentzian or 4D Euclidean dimensions coupled to scalar matter. The continuum action for this theory is

$$S = \underbrace{-\frac{1}{4g^2} \int \mathrm{d}^4x \, F_{\mu\nu}F^{\mu\nu}}_{\text{Dynamical U(1) Gauge Field}} + \underbrace{\frac{1}{2} \int \mathrm{d}^4x \left( |D_\mu \phi|^2 - V(|\phi|^2) \right)}_{\text{Minimally Coupled Complex Scalar}}, \tag{1}$$

where $F = \mathrm{d}A$ is the field strength tensor, $A$ is the vector potential, $\phi$ is a complex scalar Higgs field, $D = \mathrm{d} - iA$ is the covariant derivative, and $V(\phi)$ is a potential for the Higgs

154 field. This theory is known to exhibit two phases: a deconfined or Coulomb phase, where
155 charged particles interact via a $1/r$ Coulomb potential mediated by massless photons and a
156 gapped confinement-Higgs phase. The latter phase has two distinct regimes, a confined regime
157 at strong coupling and a Higgs regime at weak coupling, which are continuously connected
158 without any thermodynamic phase transition between them [14].

159    This theory is invariant under local $U(1)$ gauge transformations of the form

$$\phi(x) \rightarrow \phi(x)e^{i\lambda(x)}, \qquad A \rightarrow A + \mathrm{d}\lambda, \tag{2}$$

160 where $\lambda$ is an arbitrary 0-form. By construction, gauge-non-invariant operators cannot exhibit
161 a non-zero vacuum expectation which is formalized by Elitzur's theorem [3]. Thus, while it is
162 commonly stated that this theory exhibits spontaneous breaking of the U(1) gauge symmetry
163 leading to the Higgs phase, $\langle \phi(x) \rangle = 0$ and thus cannot serve as a local order parameter for
164 such a phase transition. Rather, the gauge-invariant observables are non-local string operators,
165 such as an open Wilson line

$$W_C = \phi(x)^{\dagger} \exp\left( i \int_C A \right) \phi(y), \tag{3}$$

166 where $C$ is a curve from $y$ to $x$, which creates electric charges attached to their concomitant
167 electric field lines [2, 24]. When properly normalized, such observables allow one to quanti-
168 tatively distinguish the deconfined phase from the Higgs-confined phase [25–28].

169    We demonstrate that, in the presence of open boundaries of certain type, there is a second-
170 order *boundary phase transition* distinguishing the Higgs and confined phases. We derive an
171 explicit boundary theory in a limit where the bulk is completely frozen, which we show is a
172 3D XY model, and demonstrate with Monte Carlo that the critical exponents of the boundary
173 transition do not change when we restore bulk fluctuations. Surprisingly, the line of boundary
174 transitions appears to merge with the bulk critical endpoint, see Fig. 1. We then discuss
175 deformations of the model which tune the location of the boundary transition.

## 2.1    Preliminaries: Lattice Formulation

177 We begin our discussion with a quick summary of the formulation of the discretized lattice
178 theory, the importance of the Gauss law, the role of magnetic monopoles, before introducing
179 the open boundary problem and presenting our numerical results.

### 2.1.1    Action Formulation

181 To study this theory in more depth we consider regulating it by imposing a UV lattice cutoff.
182 We undertake our exploration of Higgs phases in gauge theories within the Wilson-Fradkin-
183 Shenker lattice formulation. We work on a 4D hypercubic lattice with linear dimension $L$ and
184 periodic boundaries, a discretization of four-dimensional Euclidean spacetime. We consider
185 a a complex 0-form Higgs field taking values $\phi_i$ at each vertex $i$. Expanding the Higgs field
186 at site $i$ as $\phi_i = \rho_i \exp(i\theta_i)$, we freeze the radial mode by fixing the radius $\rho_i$, which does
187 not affect the qualitative physics.[1] Thus we work with the compact $\mathbb{R}/2\pi\mathbb{Z}$-valued 0-form
188 $\theta$, i.e. the phase of the Higgs field, which is minimally coupled to the dynamical U(1) gauge
189 field. We consider a compact 1-form gauge potential $A$ taking values on each oriented link $\ell$,
190 $A_\ell \in \mathbb{R}/2\pi\mathbb{Z}$. Denoting the reversed orientation by $-\ell$, the gauge potential satisfies $A_{-\ell} = -A_\ell$.

---

[1]This freezing corresponds to the limit of infinite bare Higgs self-coupling [14,26], and the radial mode will be
restored upon coarse-graining. Equivalently it may be regarded as a Stückelberg field, and the model can be viewed
as a lattice discretization of a gauged nonlinear $\sigma$-model with target space U(1). We use these two perspectives to
give two different generalizations to non-Abelian gauge groups in Section 3.

We will often denote link variables by their endpoints, i.e. $A_{ij} = -A_{ji}$. See Appendix A for a more detailed explanation of lattice differential forms and the notation used here.

We demand the theory to be invariant under gauge transformations of the form of Eq. (2), which on the lattice become

$$\theta \to \theta + \lambda, \quad A \to A + d\lambda, \tag{4}$$

where $\lambda$ is an arbitrary $\mathbb{R}/2\pi\mathbb{Z}$-valued 0-form, and d is the discrete exterior derivative, defined so that $(d\lambda)_{ij} \equiv \lambda_j - \lambda_i$. The minimal gauge-invariant building blocks are the Wilson links $\Lambda_\ell$, defined on oriented links $\ell$,

$$\Lambda_\ell = \exp[i(d\theta - A)_\ell], \tag{5}$$

and the minimal Wilson loops $W_p$, defined on oriented plaquettes $p$,

$$W_p = \exp\big[i\,(dA)_p\big], \tag{6}$$

where $(dA)_p = \sum_{\ell \in \partial p} A_\ell$. The minimal gauge-invariant Euclidean lattice theory is the governed by what we will refer to as the Fradkin-Shenker action,

$$S_{\text{FS}} = -\beta \sum_p \text{Re}\, W_p - \kappa \sum_\ell \text{Re}\, \Lambda_\ell\,, \tag{7}$$

which reduces upon substituting in Eqs. (5) and (6) to the Abelian-Higgs model, the lattice equivalent of Eq. (1) in the limit where the radial mode of the Higgs field is frozen,

$$S_{\text{AH}} = -\beta \sum_p \cos(dA)_p - \kappa \sum_\ell \cos(d\theta - A)_\ell\,. \tag{8}$$

Note that $\kappa$ may be interpreted as the squared length of the Higgs field. The generating function of the model is

$$Z_{\text{AH}} = \int \prod_i d\theta_i \prod_\ell dA_\ell \, \exp[-S]. \tag{9}$$

### 2.1.2 Hamiltonian Formulation and Gauss Law

It will also serve us to consider the Hamiltonian formulation of the model on a 3D cubic lattice with continuous time. This may be obtained from the action by fixing to temporal gauge ($A_\ell = 0$ on all timelike links) and taking the continuum limit in the time direction, expressing the partition sum in terms of transfer matrices [29]. The Higgs field phase and gauge connection become operators, denoted $\hat{\theta}_i$ and $\hat{A}_\ell$ respectively, acting on a local Hilbert space on each vertex or link. They each have a canonically conjugate operator, denoted $\hat{n}_i$ and $\hat{E}_\ell$ respectively, both with integer eigenvalues, satisfying $[\hat{\theta}_i, \hat{n}_i] = i$ and $[\hat{A}_\ell, \hat{E}_\ell] = i$. Thus $\exp(\pm i\hat{\theta}_i)$ is the raising/lowering operator for $\hat{n}_i$, while $\exp(\pm i\hat{A}_\ell)$ is the raising/lowering operator for $\hat{E}_\ell$. The Hamiltonian may then be expressed as[2]

$$H_{\text{AH}} = \sum_\ell \hat{E}_\ell^2 - \beta \sum_p \cos\big(d\hat{A}\big)_p + \sum_i \hat{n}_i^2 - \kappa \sum_\ell \cos\big(d\hat{\theta} - \hat{A}\big)_\ell. \tag{10}$$

The operator $\hat{n}_i$ counts the amount of charge on site $i$, while $\hat{E}_\ell$ counts the number of electric field lines on oriented link $\ell$.

---

[2]Note that the $\beta$ and $\kappa$ couplings in the Hamiltonian formulation cannot be quantitatively compared to their values in the Lagrangian formulation, as they are renormalized when taking the continuum limit in the timelike direction [30].

217     Gauge transformations, Eq. (4), are implemented by the operators

$$
\hat{G}[\lambda] = \exp\left( i \sum_i \lambda_i \hat{n}_i + i \sum_\ell (\mathrm{d}\lambda)_\ell \hat{E}_\ell \right)
$$
$$
\equiv \exp\left( i \sum_i \lambda_i \hat{n}_i + i \sum_i \lambda_i (\mathrm{d}^\dagger \hat{E})_i \right). \tag{11}
$$

218     Here we have used the coexterior derivative (see Appendix A)

$$
(\mathrm{d}^\dagger \hat{E})_i = \sum_{\ell \in \partial^\dagger i} \hat{E}_\ell \tag{12}
$$

219     where $\partial^\dagger$ indicates the coboundary, the set of oriented links *ending* at site $i$. Demanding that
220     $\hat{G}[\lambda]$ acts as the identity on physical states for arbitrary $\lambda_i$ implies that physical states satisfy
221     the Gauss law constraint

$$
-(\mathrm{d}^\dagger \hat{E})_i = (\nabla \cdot \hat{E})_i = \hat{n}_i, \tag{13}
$$

222     at each site $i$, where we used $\hat{E}_{-\ell} = -\hat{E}_\ell$ to rewrite the constraint in terms of the lattice
223     divergence. Gauge invariant states satisfying the Gauss law are then created by Wilson line
224     operators,

$$
\hat{W}[\gamma] = \exp\left( i \sum_{\ell \in \gamma} (\mathrm{d}\hat{\theta} - \hat{A})_\ell \right), \tag{14}
$$

225     where $\gamma$ is a 1-dimensional contour in the lattice. Acting on the trivial vacuum state with
226     $n_i = E_\ell = 0$ everywhere, this operator creates a unit electric charge/anti-charge pair at the
227     ends of the contour connected by a string of unit electric flux. If $\gamma$ is a closed contour, this
228     inserts a closed string of electric flux.

### 229     2.1.3   Magnetic Monopoles

230     In addition to the electric sector, there is also the magnetic sector, though it is not readily seen in
231     this formulation, instead being exposed by duality transformations [31–35] (see Section 4.2.
232     In the Hamiltonian formulation, the magnetic excitations are sources of divergence of the
233     magnetic field, $\hat{B} = \mathrm{d}\hat{A}$, i.e. magnetic monopoles. In the action formulation, they may be
234     viewed as U(1) vortex defects of the gauge field, characterized by $\mathrm{d}^2 A \neq 0$, which are allowed
235     because the identity $\mathrm{d}^2 = 0$ is only enforced modulo $2\pi$. In 4D, these homotopy defects
236     form 1-dimensional closed strings in the dual lattice, which we refer to as 't Hooft loops, the
237     worldlines of magnetic monopoles. They are necessarily included in the Euclidean lattice
238     gauge theory partition sum due to the compactness of the U(1) link variables. In the limit
239     $\beta \to \infty$ of Eq. (8), fluctuations of the gauge field are completely suppressed and no monopoles
240     are present. Correspondingly, we may equate this absence of monopoles (equivalently, the
241     closure of magnetic flux lines) to the presence of an exact magnetic 1-form symmetry, indicated
242     in Fig. 1.

### 243     2.2   Open Boundaries and Global Symmetry

244     We address now a well-known, but subtle point: by taking $\lambda$ in Eq. (4) to be a constant
245     function, it appears at first sight that this theory has a *global* 0-form U(1) symmetry, shifting
246     $\theta_i \to \theta_i + \lambda$, leaving $A$ unchanged since $\mathrm{d}\lambda = 0$. In the Hamiltonian picture, this transformation
247     is generated by the total charge,

$$
\hat{Q}_{\mathrm{bulk}} = \sum_i \hat{n}_i, \tag{15}
$$

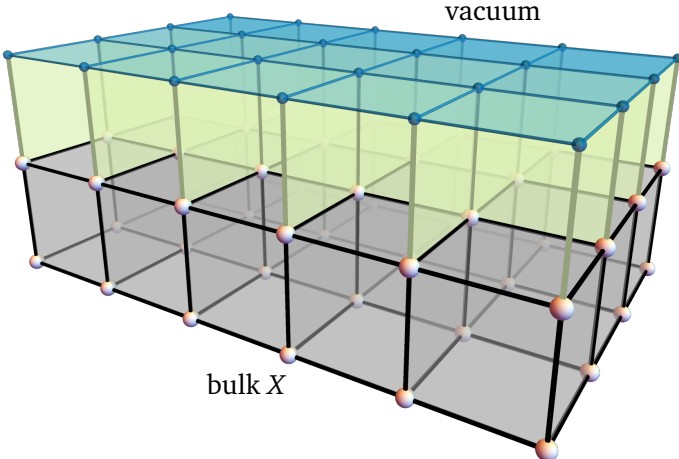

Figure 2: A demonstration of the boundary conditions considered in this work. The gauge field takes values on the bulk links (black) as well as a set of links extending out of the bulk (green). The holonomy of the gauge field is defined on all bulk plaquettes (gray), as well as on the set of plaquettes extending out of the bulk (green). The matter field takes values only in the bulk of the system (white spheres), with the Gauss law satisfied at all bulk vertices. The outside vacuum (blue sites, links, plaquettes) has no dynamical fields. With these boundary conditions electric flux is capable of passing through the boundary, allowing for non-trivial charge sectors in the bulk. We denote the bulk cells (white, black, gray) by $X$ and the boundary layer cells (green) by $\partial X$.

thus such a symmetry corresponds to global conservation of electric charge. Note, however, that in the absence of boundaries this "global symmetry" is pure gauge, because all physical quantum states belong to the zero-charge sector and thus carry the same quantum number. By the Gauss law, Eq. (13), $\hat{Q}_{\text{bulk}}$ is *exactly zero* for a system with periodic boundary conditions. In other words, since all electric flux lines must end somewhere inside the system, the system must be globally charge neutral. By construction, such a "symmetry" is therefore trivial and cannot be explicitly or spontaneously broken. However, in presence of specific boundary conditions, these global U(1) transformations actually generate a physical global symmetry that acts on the boundaries which can be spontaneously broken [22, 23].

We consider a lattice with open boundaries in the form illustrated in Fig. 2. The bulk of the system is a (hyper)cubic lattice with sites indicated by white spheres, links by black lines, and plaquettes by gray faces. At the boundary, we include a layer of cubic cells (green) which separate the bulk from the vacuum (blue sites, edges, and plaquettes). In particular, there is a set of links bridging between the bulk and the vacuum which carry dynamical gauge degrees of freedom and can therefore support electric flux lines which effectively exit the system. The vacuum side (blue) does not contain any dynamical degrees of freedom.

Key to our choice of boundary conditions is that we demand that the Gauss law, Eq. (13), is respected at every bulk site (white sphere), including those at the ends of the boundary links. No Gauss law constraints are imposed at vacuum sites. Let $A_i$ denote the gauge potential on the boundary link touching site $i$, oriented "in" from the vacuum to the bulk. For the Gauss law to be respected at $i$, we must have that under the gauge transformation Eq. (4), $A_i \rightarrow A_i + \lambda_i$, i.e. it is "uncompensated" at the vacuum end of the link.

With this choice of boundary conditions, the global part of the gauge symmetry becomes physical—charge can pass in and out of the system, meaning there are different gauge-invariant charge sectors. More precisely, there are gauge-invariant half-open Wilson string operators

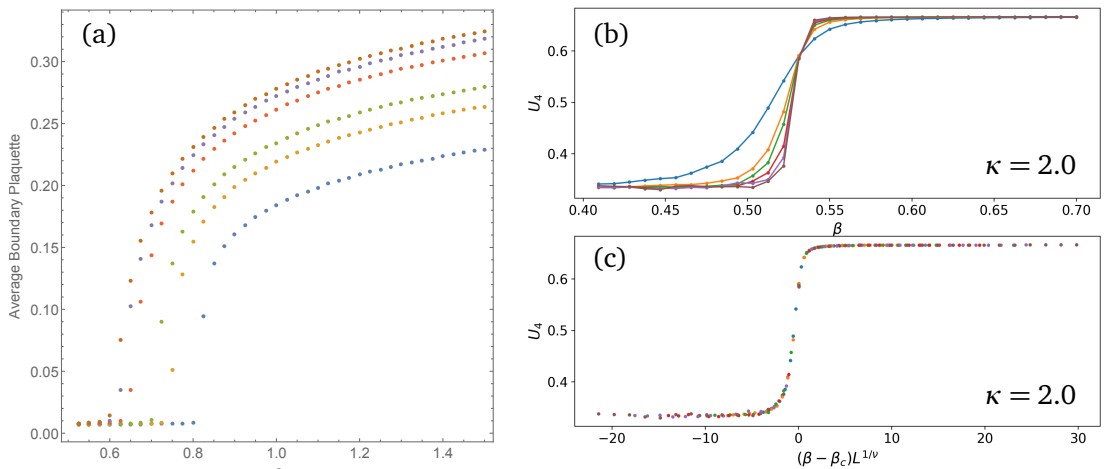

Figure 3: Boundary criticality for the 4D $U(1)$ Higgs phase: (a) Average boundary plaquette for $\kappa =$ 0.55, 0.64, 0.85, 0.94, 1.0 for $L = 16$. The transition shifts towards smaller $\beta$ for larger values of $\kappa$. (b) The Binder ratio $U_4$ for the magnetization at the boundary as a function of $\beta$ for $\kappa = 2.0$ and $L = 16, 20, 24, 28, 32$. (c) Rescaled Binder ratio for $\kappa = 2.0$ showing collapse for $\nu = 0.67$ corresponding to the 3D XY universality class.

with one end in the bulk and the other passing through the boundary,

$$\hat{W}[\gamma_{\text{open}}] = \exp\left( i \left[ \hat{\theta}_i - \sum_{\ell \in \gamma_{\text{open}}} \hat{A}_\ell \right] \right), \tag{16}$$

where $\gamma_{\text{open}}$ is a contour starting in the vacuum and ending at bulk site $i$. These operators create an isolated charge in the bulk, attached to an electric flux line that exits through the boundary, which is not possible in a closed system. These half-open string operators are *charged* under the global transformations generated by $\hat{Q}_{\text{bulk}}$, Eq. (15). By the Gauss law, $\hat{Q}_{\text{bulk}}$ is equivalent to the *net electric flux through the boundary*,

$$\hat{Q}_{\text{bulk}} = \sum_{\ell \in \partial X} \hat{E}_\ell \equiv \hat{Q}_{\text{bdry}}, \tag{17}$$

where the sum is over all boundary links (green in Fig. 2) oriented *out*. If the Hamiltonian contains no half-open string operators, then the bulk charge is conserved, and there is a global U(1) symmetry. Eq. (17) defines a "bulk-boundary correspondence" between charge and flux, and generates a global symmetry which may either be seen as acting on the bulk matter degrees of freedom or on the boundary gauge degrees of freedom.

## 2.3   Boundary Symmetry Breaking in the Abelian Higgs Model

The theory can now be chosen such that the Hamiltonian commutes with the charge $\hat{Q}_{\text{bulk}}$. This now-physical global $U(1)$ symmetry corresponds to global charge conservation or, equivalently, conservation of flux through the boundary. Since the open system has a global symmetry on the boundary, it may be spontaneously broken, a scenario we now study. With the boundary conditions shown in Fig. 2, the Euclidean action that we will study is given by

$$S = S_{\text{AH}}^{\text{bulk}} + S^{\text{bdry}}, \tag{18}$$

with the boundary portion given by the Wilson plaquette loops on the boundary plaquettes (light green in Fig. 2),

$$S^{\text{bdry}} = -\beta \sum_{p \in \partial X} \cos(\mathrm{d}A)_p. \tag{19}$$

The absent links on the vacuum side are excluded, so that

$$(\mathrm{d}A)_{p \in \partial X} = A_i + A_{ij} - A_j, \tag{20}$$

where $A_i$ indicates the value of $A$ on the boundary link touching site $i$, oriented inwards from the vacuum to the bulk.[3]

### 2.3.1 Boundary XY Model at Infinite $\kappa$

To begin, we consider the behavior of this theory in the $\kappa \to \infty$ limit, i.e. deep in the Higgs regime. From the bulk action, Eq. (8), in this limit the bulk satisfies the constraint $A = \mathrm{d}\theta$, i.e. the bulk gauge field is exact and thus pure gauge. Indeed there are no physical degrees of freedom left in the bulk—rotating to unitary gauge, $\theta = \text{const.}$, we end up with $A = 0$ on all bulk links and a vanishing matter field. However, no such constraint is enforced on the boundary links bridging between the bulk and vacuum, and the gauge field on these links is free to fluctuate. Thus we obtain a dynamical 3D theory on the boundary of the system governed by the boundary action in Eq. (19).

In this limit, referring to Fig. 2, each boundary plaquette (green) has one edge in the bulk (black) with $A_{ij} = \theta_j - \theta_i$, and two edges straddling between the bulk and vacuum (green) which remain dynamical degrees of freedom. Substituting this into Eq. (20), we can recombine terms into the gauge-invariant variables

$$\vartheta_i = A_i - \theta_i, \tag{21}$$

corresponding to a half-open Wilson line coming from the vacuum and ending at site $i$. The boundary action can then be written in the gauge-invariant form

$$S^{\text{bdry}}_{\kappa \to \infty} = -\beta \sum_{\langle ij \rangle \in \partial X} \cos\left(\vartheta_j - \vartheta_i\right), \tag{22}$$

which is a 3D XY model at inverse temperature $\beta$. This must exhibit a continuous phase transition from a paramagnet at small $\beta$ to a spontaneously broken phase at large $\beta$. Thus we infer that along the $\kappa = \infty$ line in the phase diagram there is a boundary phase transition in the 3D XY universality class indicated at the top of Fig. 1. We note that in the case of the $\mathbb{Z}_2$ Abelian-Higgs model, the same mechanism generates a boundary Ising model at $\kappa = \infty$ [22].

### 2.3.2 Boundary Phase Transition at Finite $\kappa$

Next we consider $\kappa$ to be large, $\kappa \gg 1$, but finite. The constraint $A = \mathrm{d}\theta$ is no longer enforced exactly, so we expand in small fluctuations as $A = \mathrm{d}\theta + \delta A$. Assuming that the bulk action can be expanded in terms of $\delta A \ll 1$ (i.e. that topological defects are negligible), the bulk action becomes a Proca-type action,

$$S^{\text{bulk}}_{\kappa \gg 1} \approx \frac{\beta}{2} \sum_p F_p^2 + \frac{\kappa}{2} \sum_\ell (\delta A)_\ell^2 \quad (\kappa \gg 1), \tag{23}$$

---

[3]We may equivalently consider the exterior vacuum (blue in Fig. 2) to have trivial gauge field $A_\ell = 0$ on all vacuum links and zero Higgs field $\phi_i = 0$ on all vacuum sites.

where $F_p = \mathrm{d}(\delta A)$, which describes a massive 1-form field. The boundary action is then

$$S_{\kappa \gg 1}^{\mathrm{bdry}} \sim -\beta \sum_{\langle ij \rangle \in \partial X} \cos\big(\vartheta_j - \vartheta_i + \delta A_{ij}\big). \tag{24}$$

While at infinite $\kappa$ the theory reduces to an XY model on the boundary, at finite $\kappa$ the XY model is minimally coupled to the weakly fluctuating massive bulk gauge field. While the bulk field lives in a higher dimension, the boundary remains quasi-3D, exponentially localized with a length scale determined by the mass of the bulk photon, $m^2 \sim \kappa$. We therefore expect the symmetry breaking phase transition at the boundary to persist at large but finite $\kappa$.

We may ask where the boundary transition line may run from $\kappa \to \infty$, finite $\beta$. As it is a spontaneously broken symmetry it must end either on a boundary of the phase diagram or on a bulk transition line. The former case is ruled out as follows: It cannot end on the $\beta = 0$ line because this is trivial from the point of view of both bulk and boundary variables. It also cannot end on the $\kappa = 0$ line because matter decouples on this line. The only remaining possibility is that the line ends at $\beta \to \infty$ but there we understand the bulk theory as being pure gauge and an XY model so the physical degrees of freedom on the boundary drop out. We conclude that the boundary transition line must end on a bulk transition line.

These arguments are suggestive of the picture illustrated in Fig. 1, with a boundary phase transition between the Higgs and confinement regimes of the bulk phase diagram. To test this assertion, we have carried out Monte Carlo simulations of the full 4D lattice gauge theory with boundary. We compute the local XY order parameter $\langle \vartheta_i \rangle$ on the boundary as well as gauge invariant bulk observables $\langle \Lambda_\ell \rangle$ and $\langle W_p \rangle$. The results are summarized in Fig. 3. We find clear signs of a boundary phase transition, with Fig. 3(a) showing the boundary order parameter behavior as a function of $\beta$ while holding $\kappa$ fixed showing behavior consistent with a continuous boundary phase transition. Fig. 3(b) shows the Binder cumulant for the order parameter taken along a cut at $\kappa = 2$ for different system sizes ranging from $L = 16$ to $L = 32$, showing crossing behavior consistent with a second-order transition. Lastly, Fig. 3(c) shows the Binder parameter with $\beta$ scaled by $L^{1/\nu}$ using the 3D XY critical exponent $\nu \approx 0.67$ [36], showing excellent scaling collapse, confirming a second-order phase transition on the boundary even for only moderately large $\kappa$.

Monte Carlo simulations of the 3D XY model in the literature put the critical point at $\beta_c = 0.45420(2)$ [36]. We find that $\beta_c$ tends towards this value in the large $\kappa$ limit. On general grounds we should expect that bulk fluctuations will serve to disorder the boundary. Therefore, by lowering $\kappa$ we expect the transition shifts to larger $\beta$ (lower effective temperature in the statistical model). This is indeed what we observe numerically. For even smaller $\kappa$ we find that the boundary transition line appears to intercept the bulk critical endpoint, as shown in Fig. 1.

### 2.3.3 Tuning the Boundary Coupling

We now consider tuning the boundary coupling in Eq. (19) relative to the bulk, parameterized by the dimensionless ratio

$$\alpha = \beta_{\mathrm{bdry}}/\beta_{\mathrm{bulk}}. \tag{25}$$

Such a change is allowed by gauge symmetry and does not affect either the electric or magnetic 1-form symmetries. The reason for this modification is that by tuning $\alpha$ we can shift the location of the critical $\beta$ in the $\kappa = \infty$ limit, as $\beta_{\mathrm{bdry},c}^{\kappa \to \infty}(\alpha) = \beta_{\mathrm{bdry},c}^{\kappa \to \infty}(\alpha = 1)/\alpha$. This implies that the location of the boundary transition line must shift in the phase diagram in order to meet the location of the transition in the $\kappa \to \infty$ limit.

Indeed, this is precisely what we find numerically with the resulting transition lines shown in Fig. 4. We find, for all $\alpha$, that the transition is present and that it drifts to larger $\beta$ as $\kappa$

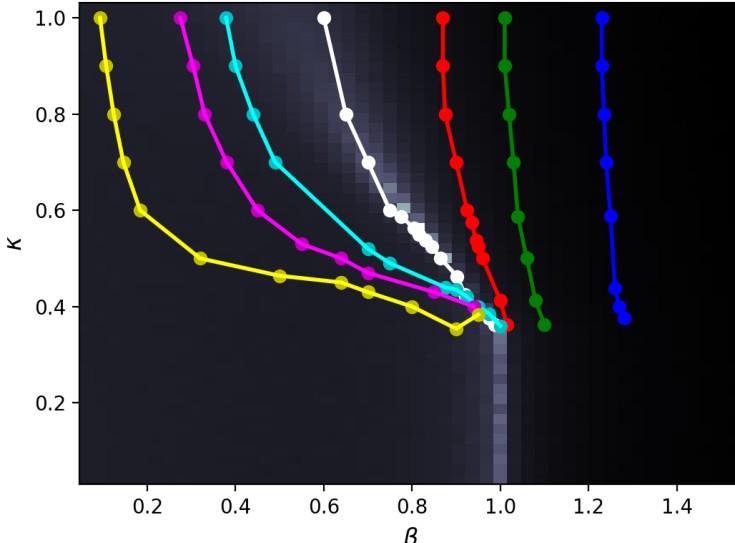

Figure 4: Boundary phase transition lines for the U(1) Abelian Higgs model, for different $\alpha = \beta_{\mathrm{bdry}}/\beta_{\mathrm{bulk}}$, overlaid on the bulk plaquette susceptibility. From right to left, $\alpha$= 0.4, 0.5, 0.6, 1.0, 2.0, 3.0, 10.0 for system size $L = 16$.

is reduced, consistent with having to lower the temperature more to suppress the enhanced matter field fluctuations. For $\alpha < 1$ the line shifts to larger values of $\beta$, and appears to separate from the tricritical point, instead terminating on the first order transition line separating the Higgs phase from the deconfined phase. For $\alpha > 1$, the boundary transition line shifts to smaller values of $\beta$, and appears to continue to terminate on the tricritical point.

## 2.4 Summary and Discussion: Abelian Case

In this section we have explored the case of the U(1) Abelian-Higgs model in $D$ spacetime dimensions with open boundary conditions of the form shown in Fig. 2, governed by the action defined by Eqs. (8), (18) and (19). In the case of open boundaries there is a global bulk charge symmetry for this model, because charge is not allowed to enter or leave the system. By the Gauss law, this is physically equivalent to a symmetry acting on the boundary of the system, yielding conservation of total electric flux through the boundary. We investigated the possibility of spontaneously breaking this symmetry.

In the $\kappa \to \infty$ limit, the bulk degrees of freedom are fully frozen while the boundary degrees of freedom remain fluctuating. The boundary theory in this limit can be written in the gauge-invariant form of a $(D-1)$-dimensional 0-form U(1) model, i.e. an XY model, Eq. (22), which therefore exhibits a boundary phase transition at a critical value of $\beta$ in the appropriate XY universality class. By performing explicit numerical simulation in $D = 4$ Euclidean space-time dimensions and measuring the gauge-invariant order parameter, Fig. 3, we found that this boundary phase transition persists away from the $\kappa \to \infty$ limit, and appears to stay in the 3D XY universality class. The transition line appears to terminate at a critical point in the bulk phase diagram and can be tuned by adjusting the boundary coupling relative to the bulk coupling, as shown in Fig. 4.

This boundary transition line appears to delineate between the Higgs and confining regimes of the phase diagram. The idea that the Higgs phase can be characterized by boundary symmetry breaking was first raised in [22,23]. This gives physical meaning to the non-gauge-invariant adage that the Higgs regime is a charge condensate by imposing open boundaries that make the symmetry physical. We note, however, that by tuning the boundary coupling relative to

the bulk coupling, the location of the transition line in the phase diagram can be moved. This raises questions as to what the boundary phase transition implies as a probe of bulk physics, which we defer to the discussion at the end of the paper (Section 5.2). First we broaden our perspective by studying more complex non-Abelian models exhibiting similar physics in Section 3.

A few comments on extensions and exceptional cases are in order before we proceed. These results naturally carry over to the discrete Abelian gauge groups $\mathbb{Z}_N$ by restricting the Higgs field $\theta_i$ and gauge field $A_{ij}$ to take discrete values in multiples of $2\pi/N$. The boundary theory Eq. (22) then becomes an $N$-state clock model. The $\mathbb{Z}_2$ case was discussed in [22] where the boundary theory was shown to be an Ising model. They also can be generalized to higher-form extensions of the Abelian-Higgs model, which we discuss further in Section 4.

The $D = 3$ U(1) Abelian-Higgs model is interesting because it reduces in the $\kappa \to \infty$ limit to the $D = 2$ XY model on the boundary, which exhibits a BKT transition. It would be interesting to know if this BKT transition persists to finite $\kappa$. We complete our discussion of special cases by highlighting here $U(1)$ gauge theory in four dimensions coupled to a charge $q = 2$ Higgs field. For this case, a sharp distinction can be made in the bulk between confining and Higgs phases with a transition between them. The distinguishing feature of this theory is the partial Higgsing of the $U(1)$ gauge group down to $\mathbb{Z}_2$ [37]. For sufficiently large $\kappa$ the bulk transition is the confinement-deconfinement transition of the residual $\mathbb{Z}_2$ gauge theory. On the boundary, however, one still expects the emergence of the XY model we identified for the $q = 1$ case. It would be quite interesting to investigate the interplay of boundary U(1) 0-form symmetry breaking with the bulk $\mathbb{Z}_2$ 1-form symmetry breaking and the resulting topological order present at large $\beta$ and $\kappa$.

## 3  Boundary Symmetry Breaking in Non-Abelian Higgs Models

We now turn to extend the results of the previous section regarding Abelian Higgs models and boundary criticality to non-Abelian Higgs models. We will show that the general picture of boundary symmetry breaking persists, albeit with a richer structure owing to a set of non-commuting gauge transformations. We discuss two types of non-Abelian Higgs models: those with group-valued Higgs fields and those with fundamental representation vector-valued Higgs fields with fixed length. The two classes of models are equivalent for gauge group SU(2), and are distinct for other gauge groups. The group-valued case is a relatively straightforward extension of the Abelian case, because the fixed-length XY-rotor Higgs field considered previously is naturally a U(1) group element. The vector-valued case is more subtle because the $\kappa \to \infty$ limit does not trivialize all bulk degrees of freedom. We present numerical results for both SU(2) and vector-valued SU(3) cases and extract corresponding boundary criticalities.

### 3.1  Preliminaries: Lattice Formulation

Let $\mathcal{G}$ be a compact connected Lie group, e.g. SU($N$) or SO($N$). The lattice action is formulated in terms of group-valued link variables $U_\ell \in \mathcal{G}$ satisfying $U_{-\ell} = U_\ell^{-1}$, which may be viewed as the exponentiated Lie-algebra-valued gauge field, $U_\ell = P \exp(i \int_\ell A)$, where $P$ indicates path-ordering. Under a gauge transformation, these transform as

$$U_{ij} \to g_i U_{ij} g_j^{-1} \tag{26}$$

where $g_i \in \mathcal{G}$ are arbitrary group elements associated to each site $i$. The minimal gauge-invariant quantity is the Wilson plaquette-loop (compare to Eq. (6)),

$$W_p = \frac{1}{\dim(r)} \text{Tr}_r \Big[ \prod_{\ell \in \partial p}^{P} U_\ell \Big] \tag{27}$$

where the superscript $P$ on the product indicates path-ordering, and the trace may be taken in a representation $r$. Normalizing by the dimension of the representation ensures that the trivial Wilson loop has unit magnitude. We focus primarily on the cases SU($N$) and SO($N$), taking the trace in the fundamental representation as $N \times N$ matrices.

### 3.1.1 Higgs Fields

For the Higgs field, many different models can be considered by putting the Higgs field in different representations of the gauge group. We consider two different types here for concreteness: vector-valued (fundamental representation) Higgs, and group-valued Higgs. These are both possible extensions of the Abelian U(1) rotor model considered in Section 2, since a rotor may be viewed either as a fixed-length vector, or as a U(1) group element. In either case, the action is given by the Fradkin-Shenker form, Eq. (7), the only difference being the definition of the Wilson link $\Lambda_\ell$. The generating function for the quantum theory is given by the Euclidean path integral, where the integration over the group-valued variables is performed with respect to the Haar measure.

The familiar model is the fundamental-Higgs, where the Higgs field is an $N$-component vector, as in the Standard Model and analogous to Eq. (1). Denoting the Higgs vector at site $i$ by $\phi_i \in V_i$, we freeze the radial mode as in the Abelian Fradkin-Shenker model. Gauge transformations rotate the Higgs field as $\phi_i \to g_i^{\text{f}} \phi_i$ where $g_i^{\text{f}}$ is a group element in the fundamental matrix representation. The group-valued link variables define parallel-transport maps for the Higgs field, $U_{ij} : V_j \to V_i$, i.e. they related the color frames at neighboring sites, and the generalization of the gauge-invariant Wilson link observable, Eq. (5), is

$$\Lambda_\ell = \langle \phi_i, U_{ij}^{\text{f}} \phi_j \rangle_i \equiv \sum_{\alpha, \beta=1}^{N} \phi_i^{\alpha *} (U_{ij}^{\text{f}})^{\alpha \beta} \phi_j^{\beta}, \tag{28}$$

where $\langle -, - \rangle_i$ is the canonical inner product on $V_i$, $*$ indicates complex conjugation, and we enforce the fixed-length constraint $\langle \phi_i, \phi_i \rangle = 1$.

The second type of model we consider takes the Higgs field to be *group-valued*, like the link variables, denoted $\varphi_i \in \mathcal{G}$. In this case, the Higgs field transforms as $\varphi_i \to g_i \varphi_i$ under gauge transformations, and we can define a gauge-invariant Wilson link by

$$\Lambda_\ell = \frac{1}{N} \text{Tr}_{\text{f}} [\varphi_i^{-1} U_{ij} \varphi_j], \tag{29}$$

where we take the trace in the fundamental representation. This is a lattice regularization of a gauged principal chiral model [38], a non-linear $\sigma$-model whose target space is the group manifold.

### 3.1.2 Hamiltonian Formulation and Gauss Law

The Hamiltonian formulation of the non-Abelian lattice gauge theory has a similar form to the Abelian case, but the electric field of the non-Abelian theory carries color indices and the different components do not commute. Fixing to temporal gauge and reformulating the partition function using transfer matrices, taking the continuum limit in the time direction one

obtains a Hamiltonian for the time evolution [30, 39]. The basic ingredients are the group-valued link operators $\hat{U}_\ell$ with eigenstates $|U\rangle$, such that $\hat{U}_\ell|U\rangle = U|U\rangle$ and $\hat{U}_{-\ell}|U\rangle = U^{-1}|U\rangle$, along with a set of translation operators

$$\hat{T}_\ell(g)|U\rangle = |gU\rangle, \quad \hat{T}_{-\ell}(g)|U\rangle = |Ug^{-1}\rangle, \tag{30}$$

where left and right translations correspond to the two orientations of the link. Each group element may be expressed as $U = \exp(i\theta^a t^a)$, where $\theta^a$ are real numbers and $t^a$ a basis for the Lie algebra of gauge group $\mathcal{G}$. The $\theta^a$ serve as coordinates on the group manifold, and may be thought of as generalized Euler angles. The link operators can then be expressed as

$$\hat{U}_\ell = \exp(i\hat{\theta}_\ell^a t^a), \qquad \hat{T}_\ell(e^{i\lambda^a t^a}) = \exp(i\lambda^a \hat{E}_\ell^a). \tag{31}$$

The operators $\hat{\theta}_\ell^a$ are position operators on the group manifold, while $\hat{E}_\ell^a$ are the color-electric fields, which serve as the conjugate momenta and can be expressed as derivatives with respect to the $\theta^a$. The electric fields satisfy the same commutation relations as the group generators,

$$[\hat{E}_\ell^a, \hat{E}_\ell^b] = if^{abc}\hat{E}_\ell^c, \tag{32}$$

where $f^{abc}$ are the structure constants of $\mathcal{G}$.

For the Higgs field in the group-valued representation, we define the group-valued operators $\hat{\varphi}_i$ (analogous to $\hat{U}$) and left- and right-translation generators $\hat{t}_{i,L}^a$ and $\hat{t}_{i,R}^a$ (analogous to $\hat{E}$). For the Higgs field in the fundamental vector representation with frozen radial mode, the classical configuration space is that of a rigid rotor, and we define corresponding angular momentum operators $\hat{J}_i^\mu$. The Hamiltonian is then given by the Kogut-Susskind form [39, 40]

$$H = \sum_\ell |\hat{E}_\ell|^2 - \beta \sum_p (\hat{W}_p + \hat{W}_p^\dagger) + \sum_i |\hat{Q}_i^{\mathrm{m}}|^2 - \kappa \sum_\ell (\hat{\Lambda}_\ell + \hat{\Lambda}_\ell^\dagger), \tag{33}$$

where all sites, links, and plaquettes are purely spatial. Here, $\hat{Q}_i^{\mathrm{m}}$ are the matter charge operators,

$$\hat{Q}_i^{\mathrm{m}} = \begin{cases} \hat{t}_{i,L} & \text{group-valued Higgs,} \\ \hat{J}_i & \text{fundamental Higgs,} \end{cases} \tag{34}$$

and $|\hat{E}_\ell|^2$ and $|\hat{Q}_i^{\mathrm{m}}|^2$ are the corresponding quadratic Casimir operators, which do not depend on whether we use left- or right-generators. The operator $\hat{W}_p$ is the operator analog of Eq. (27), the trace of the oriented product of $\hat{U}_\ell$ on the links of spatial plaquette $p$. Similarly, $\hat{\Lambda}_\ell$ is the operator analog of Eq. (29).

The eigenstates of $|\hat{E}_\ell|^2$ correspond to the irreducible representations of the gauge group, with the $\hat{U}_\ell$ acting as raising and lowering operators [39–41]. The same is true for the group-valued Higgs, with $\hat{\varphi}_i$ acting as the raising and lowering operators, while for the fundamental vector-valued Higgs, the charge eigenstates are angular momentum eigenstates of a rigid rotor.

Gauge transformations are performed by the operators

$$\hat{G}[\lambda] = \exp\left( i\sum_i \lambda_i^a \hat{Q}_i^{\mathrm{m},a} + i\sum_{\langle ij\rangle} (\lambda_i^a \hat{E}_{ij}^a + \lambda_j^a \hat{E}_{ji}^a) \right)$$
$$= \exp\left( i\sum_i \lambda_i^a \hat{Q}_i^{\mathrm{m},a} + i\sum_i \lambda_i^a (\nabla \cdot \hat{E}^a)_i \right), \tag{35}$$

where the lattice divergence is defined as

$$(\nabla \cdot \hat{E}^a)_i = \sum_{-\ell \in \partial^\dagger i} \hat{E}_\ell^a, \tag{36}$$

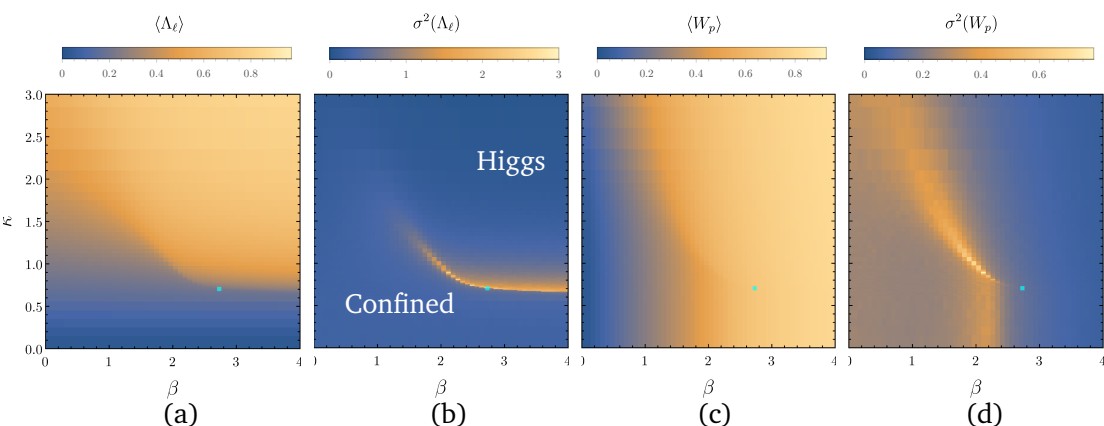

Figure 5: Bulk phase diagram of the SU(2) Higgs model, showing (a) the link expectation value $\langle \Lambda_\ell \rangle$, (b) the link variance $\sigma^2(\mathrm{Re}\,\Lambda_\ell)$, (c) the Wilson plaquette average $\langle \mathrm{Re}\,W_p \rangle$, and the plaquette variance $\sigma^2(\mathrm{Re}\,W_p)$. The small cyan-colored point is the location of the critical endpoint identified in Ref. [42], $(\beta_c, \kappa_c) \approx (2.73, 0.70)$. From the critical endpoint there is a first-order transition line extending to larger $\beta$ with nearly-constant $\kappa$, which is clearly seen in (b). A rapid-crossover region extends from the critical point to smaller $\beta$ and larger $\kappa$, signaled in both (b) and (d) by strong bulk fluctuations, indicating the "supercritical" region which roughly delineates the Higgs and confined regimes.

with the sum taken over the links emanating from site $i$ oriented *out*. $\hat{G}[\lambda]$ acts as the identity on physical, gauge-invariant states, which therefore satisfy the color-electric Gauss laws,

$$(\nabla \cdot \hat{E})^a_i = -\hat{Q}^{\mathrm{m},a}_i, \tag{37}$$

one for each color index.

Note that this is similar but subtly distinct from the Abelian case, Eqs. (11) to (13), where we used $\hat{E}^{\mathrm{Abelian}}_\ell = -\hat{E}^{\mathrm{Abelian}}_{-\ell}$. This relationship is not true in non-Abelian gauge theory. Instead, left- and right-translations of the gauge field are related by

$$T_{-\ell}(g)|U_\ell\rangle = |U_\ell g^{-1}\rangle = T_\ell(U_\ell g^{-1} U_\ell^{-1})|U_\ell\rangle, \tag{38}$$

which implies that the electric field in the two directions along a link are related by

$$\hat{E}^a_{-\ell}|U\rangle = -U^{\mathrm{a},ab}_\ell \hat{E}^b_\ell |U\rangle, \tag{39}$$

where $U^{\mathrm{a}}_\ell$ is the adjoint representation of $U_\ell$. As such the gauge field itself is charged in the adjoint representation with respect to color rotations, generated by the charge operators

$$\hat{Q}^{\mathrm{g},a}_\ell = \hat{E}^a_\ell + \hat{E}^a_{-\ell}, \tag{40}$$

which are manifestly orientation-independent. The classic (though heuristic) way to think of this is that the gauge bosons (gluons) carry a distinct charge and anti-charge in the two directions along the link. In the Abelian case, $U^{\mathrm{a}}_\ell = 1$ in Eq. (39), and the link charge Eq. (40) is exactly zero.

### 3.1.3  Open Boundaries and Global Symmetry

We introduce electric open boundary conditions as in Fig. 2, with dynamical links extending from the bulk to the vacuum which allow electric flux to pass through the boundary. We denote the link variables on the boundary link touching site $i$ by $U_i$, with the convention that the link

is oriented "in" from the vacuum to site $i$. We have minimal open Wilson strings going around the boundary plaquettes, which we can write as

$$W_{p \in \partial X} = \frac{1}{N} \operatorname{Tr}_f [U_i U_{ij} U_j^{-1}] \tag{41}$$

Using these, we define the boundary action for our theory directly analogous to the Abelian case as

$$S^{\text{bdry}} = -\beta \sum_{p \in \partial X} \operatorname{Re} W_p \tag{42}$$

Electric flux can thus enter and leave the system, but matter charges cannot. Under gauge transformations the fields transform as

$$\mathcal{G}_{\text{gauge}} : \begin{cases} \phi_i \to g_i^f \phi_i \quad \text{or} \quad \varphi_i \to g_i \varphi_i\,, \\ U_{ij} \to g_i U_{ij} g_j^{-1}\,, \\ U_i \to U_i g_i^{-1}. \end{cases} \tag{43}$$

Notice that each boundary link only receives a transformation from the *inside* end, where it terminates on a matter field. In addition to this gauge symmetry, the boundary action Eq. (42) has a *physical global $\mathcal{G}$ symmetry acting on the boundary*,

$$\mathcal{G}_{\text{bdry}} : U_i \to g U_i \tag{44}$$

where every boundary link is translated from the *outside* end.

This boundary symmetry is similar to the Abelian case, but with a subtle distinction. The total color charge of the system, including boundary links, is

$$\hat{Q}_{\text{total}}^a = \sum_i \hat{Q}_i^{\text{m},a} + \sum_\ell \hat{Q}_\ell^{\text{g},a}\,, \tag{45}$$

which rotates all matter fields in the fundamental representation and all gauge fields, including boundary links, in the adjoint representation. But this is *not* the generator of global gauge transformations, Eq. (43), under which the boundary links only rotate from the *inside*. The generator of global gauge transformations is the "bulk charge"

$$\hat{Q}_{\text{bulk}}^a = \sum_i \hat{Q}_i^{\text{m},a} + \sum_{\ell \in X} \hat{Q}_\ell^{\text{g},a} + \sum_{\ell \in \partial X} \hat{E}_{-\ell}^a\,, \tag{46}$$

where the second sum contains only the bulk links, and in the last sum the boundary links are oriented inwards. By the Gauss law, this operator must be *zero* on the physical gauge-invariant Hilbert space. On the other hand, the generator of the global symmetry is the "boundary charge"

$$\hat{Q}_{\text{bdry}}^a = \sum_{\ell \in \partial X} \hat{E}_\ell^a\,, \tag{47}$$

again with inward orientation. Together these make up the total charge of the system,

$$\hat{Q}_{\text{total}}^a = \hat{Q}_{\text{bulk}}^a + \hat{Q}_{\text{bdry}}^a = \hat{Q}_{\text{bdry}}^a\,, \tag{48}$$

where we have assumed global gauge invariance to identify $\hat{Q}_{\text{bulk}}^a = 0$. Thus the boundary symmetry Eq. (44) may be viewed, by the Gauss law constraint, as being generated by the total color charge of the system. This is analogous to the Abelian case, where the total charge of the system is just the matter charge, Eq. (15), since the links do not carry any charge. Note, however, that in the non-Abelian case the charge of the boundary links is "fractionalized" into a piece that contributes to the bulk charge and a piece that contributes to the boundary charge.

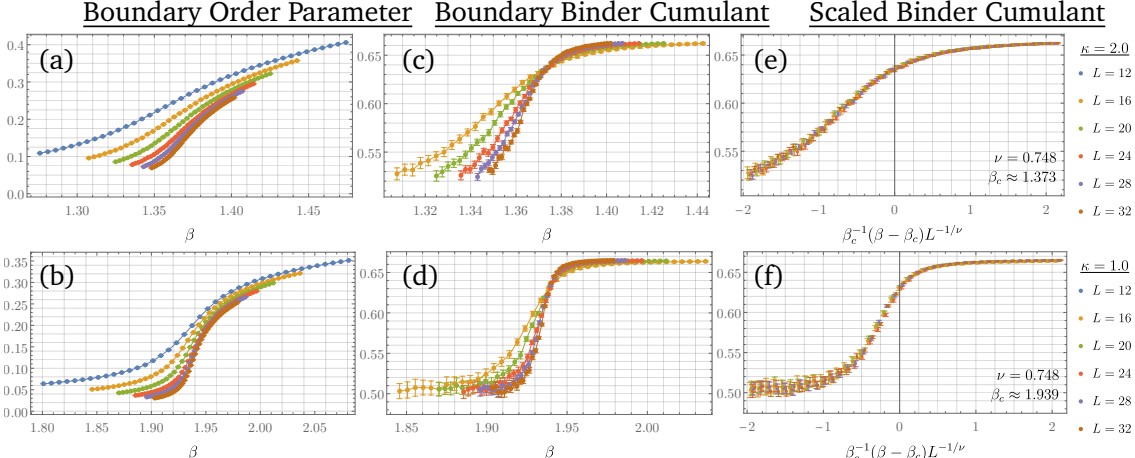

Figure 6: Boundary criticality in the SU(2) Higgs model. (a,b) Boundary order parameter, (c,d) Binder cumulant, and (e,f) scaled Binder cumulant for the SU(2) Higgs model at (a,c,e) $\kappa = 2.0$ and (b,d,f) $\kappa = 1.0$, each shown for system sizes ranging from $L = 12$ to $L = 32$. Each data point is averaged over $10^5$ samples. The Binder cumulants for different system sizes collapse when scaled with the 3D O(4) universality critical exponent $\nu \approx 0.748$ [43], indicating that the boundary phase transition remains second order all the way to the bulk critical endpoint (with $\kappa_c \approx 0.7$ [42]).

## 3.2 Boundary Symmetry Breaking in the SU(2) Higgs Model

We focus first on the case where the gauge group is SU(2). In this case the fundamental and group-valued representations are equivalent. In the fundamental representation, each $\phi_i$ is a $\mathbb{C}^2$ vector with unit length, and the configuration space is a 3-sphere. Notice that SU(2) is also topologically the 3-sphere, meaning that every configuration of the Higgs vector $\phi_i$ can be written as a unique fundamental-representation SU(2) matrix times a fixed vector, for example as

$$\phi_i = \begin{pmatrix} \phi_1 \\ \phi_2 \end{pmatrix} = \begin{pmatrix} \phi_1 & -\phi_2^* \\ \phi_2 & \phi_1^* \end{pmatrix} \begin{pmatrix} 1 \\ 0 \end{pmatrix} \equiv \varphi_i^{\mathrm{f}} \phi_0, \tag{49}$$

where $\phi_1$ and $\phi_2$ are complex numbers. Note that the determinant of $\varphi_i^{\mathrm{f}}$ is the length of the rotor. The Wilson link for the vector Higgs model, Eq. (28), can then be written as

$$\Lambda_{ij} = \langle \phi_0, (\varphi_i^{\mathrm{f}})^{-1} U_{ij}^{\mathrm{f}} \varphi_j^{\mathrm{f}} \phi_0 \rangle = \frac{1}{2} \mathrm{Tr}_{\mathrm{f}} [\varphi_i^{-1} U_{ij} \varphi_j] \tag{50}$$

which is exactly equivalent to the group-valued Higgs definition, Eq. (29). This makes the SU(2) Higgs rotors special, since they may be viewed as either vector-valued or group-valued (which is also the case for U(1) rotor).

### 3.2.1 Bulk Phase Diagram

To map the phase diagram, we perform classical Monte Carlo simulations for the 4D SU(2) model defined by the Fradkin-Shenker action, Eq. (7), with periodic boundary conditions on an $L^4$ hypercubic lattice [30,44,45]. The phase diagram can be mapped out by measuring the Wilson plaquette and link observables along with their variances (i.e. susceptibilities), which are shown in Fig. 5. There is a roughly horizontal first-order transition line extending from a critical endpoint at $(\beta_c, \kappa_c) \approx (2.73, 0.70)$ [42] (cyan box) towards $\beta \to \infty$. This line is clearly visible in the link susceptibility, Fig. 5(b).

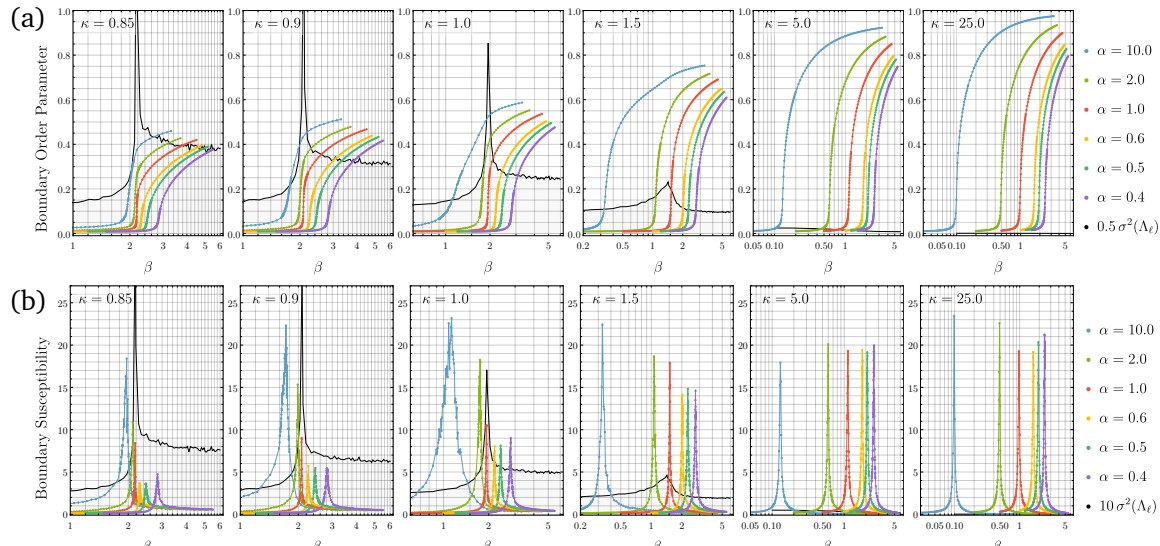

Figure 7: Results for varying $\alpha$ in the SU(2) Higgs model. Behavior of (a) the boundary order parameter and (b) the boundary order parameter variance, cut along $\beta$ with $\kappa$ fixed, for different values of $\alpha$. The variance of $\Lambda_\ell$ in the bulk is shown behind (black curves, scaled for visibility), as an indication of the magnitude of bulk fluctuations and how they influence the boundary transition in a finite-size system. Data taken at $L = 24$ with $10^4$ samples averaged for each data point.

The phase diagram exhibits only one clearly distinct bulk phase—the confined-Higgs phase— a deconfined phase being absent in non-Abelian theories. The phase diagram is, however, roughly separated into two regions indicated by the behavior of the Wilson link expectation value, shown in Fig. 5(a), with $\langle \mathrm{Re}\,\Lambda_\ell \rangle \sim 0$ indicating the confining regime and $\langle \mathrm{Re}\,\Lambda_\ell \rangle \sim 1$ indicating the Higgs regime. To the left of this critical endpoint is a supercritical region (a Widom line [46]) extending to smaller $\beta$ and larger $\kappa$, a rapid crossover from the Higgs to the confined regimes. The location of this supercritical region is most evident in the Wilson plaquette susceptibility, Fig. 5(d), which shows a pronounced intensity emanating from the critical endpoint. We expect this phase diagram to be qualitatively consistent with those for general non-Abelian Higgs models with either fundamental vector- or group-valued Higgs fields. In the group-valued case the continuity of the Higgs and confined regimes was proven by Fradkin and Shenker [14].

### 3.2.2 Boundary Symmetry Breaking

We now consider open boundaries, with boundary action Eq. (42) in a slab geometry. As in Section 2.3, we start by considering the limiting behavior when $\kappa \to \infty$. In this limit every bulk link satisfies the constraint $\mathrm{Re}\,\Lambda_\ell = 1$. In this section we will resolve this constraint by fixing a gauge. Gauge-invariant formulations for the group-valued representation are presented in Section 3.3, and for the fundamental representation in Section 3.4.

From Eq. (50), if we fix to unitary gauge where all the Higgs rotors are aligned globally, $\phi_i = \phi_0$ or $\varphi_i = \mathbb{1}$, the bulk constraint becomes $(U_\ell^{\mathrm{f}})^{11} = 1$, or $\mathrm{Tr}_{\mathrm{f}}[U_\ell]/2 = 1$, which can only be satisfied if $U_\ell = \mathbb{1}$ on every bulk link. Thus in the $\kappa \to \infty$ limit of the SU(2) Higgs model, the bulk is completely frozen and has no remaining degrees of freedom, as in the Abelian case. The boundary links, however, have no constraint and remain fluctuating. In unitary gauge

where the bulk links are set to the identity, the boundary action becomes

$$S_{\kappa \to \infty}^{\text{bdry}} = -\beta \sum_{\langle ij \rangle \in \partial X} \frac{1}{2} \text{Tr}_f[U_i U_j^\dagger] \quad \text{(unitary gauge).} \tag{51}$$

This boundary action may be viewed as a lattice discretization of a nonlinear $\sigma$-model with target space the SU(2) group manifold, i.e. a principal chiral model [38].

The boundary model, Eq. (51), has an SU(2)×SU(2)≃O(4) symmetry. To see this, we can re-express it as an O(4) Heisenberg model by representing the SU(2) group-valued link variables as unit quaternions,

$$U_i \equiv \sum_{\mu=1}^{4} S_i^\mu \sigma^\mu \quad \text{with} \quad \sum_{\mu=1}^{4} S_i^\mu S_i^\mu = 1 \quad \text{(unitary gauge),} \tag{52}$$

where the $S_i^\mu$ are real numbers, $\sigma^0 = \mathbb{1}$ and $\sigma^1$, $\sigma^2$, and $\sigma^3$ are Pauli matrices. The boundary action then becomes an O(4) Heisenberg model,

$$S_{\kappa \to \infty}^{\text{bdry}} = -\beta \sum_{\langle ij \rangle \in \partial X} S_i^\mu S_j^\mu \quad \text{(unitary gauge).} \tag{53}$$

Therefore, in the limit $\kappa \to \infty$, the system exhibits a boundary phase transition in the 3D O(4) universality class, at a critical coupling $\beta_{\text{bdry},c} \approx 0.9360$ [43], with order parameter,

$$\mathcal{O}_{\text{SU(2)}} = \left\langle \left| \frac{1}{L^3} \sum_{i \in \partial X} \boldsymbol{S}_i \right| \right\rangle \quad \text{(unitary gauge).} \tag{54}$$

The gauge-invariant object which reduces to $U_i$ in the unitary gauge is

$$U_i \varphi_i \quad \text{or} \quad U_i \phi_i, \tag{55}$$

where the former is an SU(2) matrix which decomposes according to Eq. (52), and the latter is explicitly a 4-component unit-length vector. For large but finite $\kappa$, the bulk fluctuations are strongly gapped and the boundary should behave as quasi-$(D-1)$-dimensional, and we expect the boundary phase transition to persist as in the Abelian case.

To test this prediction, we perform Monte Carlo simulations with open boundaries and measure the order parameter, Eq. (54) (defined in terms of gauge invariant observables Eq. (55)), at finite values of $\kappa$. In Fig. 6 we show the evolution of the boundary order parameter as a function of $\beta$, for $\kappa = 2.0$ in (a) and $\kappa = 1.0$ in (b), for different system sizes. These reveal a transition from a disordered, symmetric boundary on the confined side (small $\beta$) to an ordered, symmetry-broken boundary on the Higgs side (large $\beta$). This value of $\kappa$ is quite close to the bulk critical point ($\kappa_c \sim 0.7$), demonstrating that the boundary phase transition persists far into the phase diagram where the bulk is quite strongly fluctuating. In Fig. 6(c) and (d), we show the Binder cumulant for the same cuts, showing crossing behavior at a $\kappa$-dependent critical coupling. In (e) and (f) we have rescaled $\beta - \beta_c(\kappa)$ using the 3D O(4) critical exponent $\nu \approx 0.748$ [43], demonstrating a clean scaling collapse, thus verifying that the transition remains second order and in the same universality class even for relatively small values of $\kappa$. The transition line appears to terminate at the bulk critical point, which can be seen in the red line in Fig. 7, though verifying this numerically is difficult as the bulk correlation length grows larger than the finite width of the open boundaries as the system approaches bulk criticality.

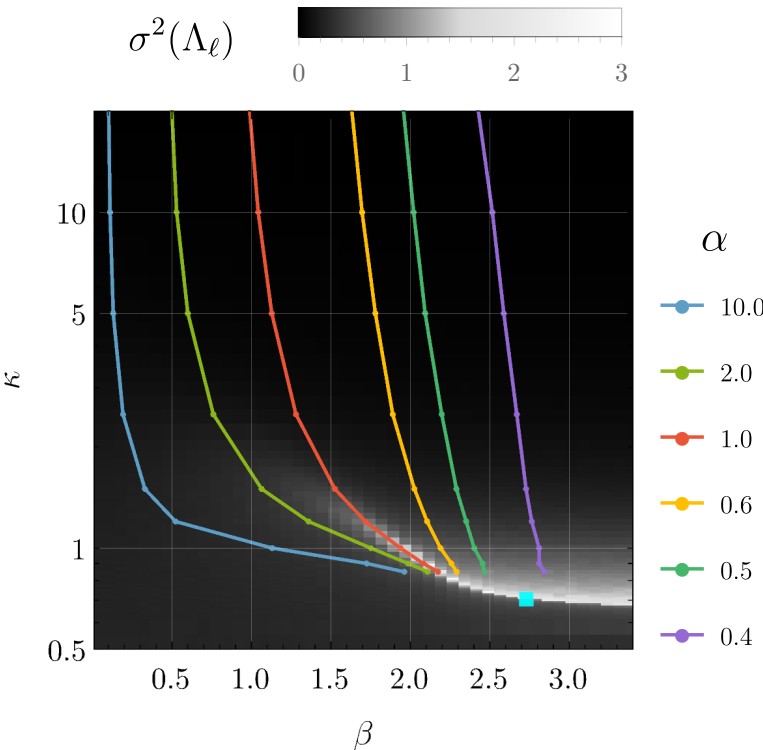

Figure 8: Location of the boundary phase transition for the SU(2) Higgs model for different values of $\alpha = \beta_{\text{bdry}}/\beta_{\text{bulk}}$, determined by the location of the peak in the boundary order parameter susceptibility (cf. Fig. 7) . Large values of $\alpha$ push the transition line to smaller $\beta_{\text{bulk}}$, while small values of $\alpha$ push the phase boundary to larger $\beta_{\text{bulk}}$. The cyan square indicates the location of the bulk critical endpoint. For $\alpha = 1$ (red), the boundary transition line appears to terminate at the bulk critical point, and closely follows the bulk rapid-crossover (super critical) region that extends beyond the critical endpoint. This remains true for $\alpha > 1$, while for sufficiently small $\alpha < 1$ the boundary transition line appears to terminate on the bulk first-order transition line.

### 3.2.3 Tuning the Boundary Coupling

We now consider varying the parameter $\alpha = \beta_{\text{bdry}}/\beta_{\text{bulk}}$, which shifts the location of the $\kappa \to \infty$ transition. Figure 7 shows the behavior of the boundary order parameter and its susceptibility for different values of $\alpha$, along different constant-$\kappa$ cuts at fixed system size. The corresponding behavior of the bulk link susceptibility is shown in black in the background for reference. The bulk transition line moves as $\alpha$ is varied, but appears to remain second-order throughout. Figure 8 summarizes the results by showing the approximate location of the boundary transition line for different values of $\alpha$, with results very similar to the Abelian case (Fig. 4). For $\alpha > 1$ the transition moves to smaller values of $\beta_{\text{bulk}}$ and appears to terminate at the bulk critical endpoint (cyan box). No boundary transition is detected for small values of $\kappa$ below the bulk first-order line. For $\alpha < 1$ the location of the boundary transition line moves to larger values of $\beta$, and appears to terminates on the line of bulk first-order transitions, at least for sufficiently small $\alpha$.

### 3.3 Boundary Symmetry Breaking in Non-Abelian Group-Valued Higgs Models

Having verified the existence of a boundary phase transition in the SU(2) Higgs model, which is both a fixed-length-rotor Higgs model and a group-valued Higgs model, we now consider how these results generalize to these two types of models separately for a general gauge group $\mathcal{G}$. From the point of view of the boundary action, the group-valued Higgs is the simpler case, so we consider it first. The action (up to an overall normalization convention for the trace) is

$$S_{\text{bulk}} = -\beta \sum_{p \in X} \text{Tr} \prod_{\ell \in \partial p} U_\ell - \kappa \sum_{\langle ij \rangle \in X} \text{Tr} \big[ \varphi_i^{-1} U_{ij} \varphi_j \big]. \tag{56}$$

The boundary action is given by Eq. (42). In addition to the global color charge symmetry acting on the boundary, Eq. (44), it also has a global $\mathcal{G}$ symmetry given by right multiplication of the Higgs field

$$\mathcal{G}_{\text{bulk}} : \varphi_i \rightarrow \varphi_i g. \tag{57}$$

The global symmetry group is therefore $\mathcal{G}_{\text{bdry}} \times \mathcal{G}_{\text{bulk}}$. We expect the bulk phase diagram to be qualitatively similar to the SU(2) case, Fig. 5, with a single thermodynamic phase, a first-order line terminating at a critical endpoint. These models were considered by Fradkin and Shenker [14], who showed that the Higgs and confining regimes are contiguous, as in the Abelian models.

We now consider taking the $\kappa \rightarrow \infty$ limit. Maximizing the trace in Eq. (56) yields the constraint $\varphi_i^{-1} U_{ij} \varphi_j = \mathbb{1}$, which implies that the bulk links can be expressed in terms of the matter field as

$$U_{ij} \xrightarrow{\kappa \rightarrow \infty} \varphi_i \varphi_j^{-1}. \tag{58}$$

The bulk of the system is completely frozen in this limit, which can be most easily seen in unitary gauge where $\varphi_i = \mathbb{1}$. Substituting Eq. (58) into Eq. (41), the boundary action can then be expressed in terms of the gauge-invariant observables

$$\Theta_i = U_i \varphi_i, \tag{59}$$

where $U_i$ was defined with the boundary link oriented "in", which are short half-open Wilson strings coming from the vacuum and ending at site $i$. The boundary action becomes

$$S_{\kappa \rightarrow \infty}^{\text{bdry}} = -\beta \sum_{\langle ij \rangle \in \partial X} \text{Re} \, \text{Tr} \big[ \Theta_i \Theta_j^{-1} \big], \tag{60}$$

which is a lattice chiral model. Compare this to the equivalent results for the Abelian case, Eq. (22) and Eq. (51), to which it reduces when $\mathcal{G} = \text{U}(1)$. The $\mathcal{G}_{\text{bdry}} \times \mathcal{G}_{\text{bulk}}$ symmetry acts on this chiral model by left and right multiplication of the $\Theta_i$, respectively. Such a model is known to exhibit chiral symmetry breaking, i.e. breaking to the diagonal $\mathcal{G}$ subgroup [47]. We discuss boundary phase transitions in these models further in Section 3.5

Note that if $\mathcal{G}$ is Abelian there is no distinction between left and right group multiplication, therefore $\mathcal{G}_{\text{bulk}}$ and $\mathcal{G}_{\text{bdry}}$ are not independent symmetries of the system. $\mathcal{G}_{\text{bulk}}$ corresponds to global rotation of the Higgs field generated by the total matter charge, Eq. (15), while $\mathcal{G}_{\text{bdry}}$ is generated by the net electric flux through the boundary links, Eq. (17). These two generators are the same operator on the physical gauge-invariant Hilbert space by the Gauss law. In contrast, in the group-valued non-Abelian Higgs models these are really distinct symmetries generated by different physical operators.

### 3.4  Boundary Symmetry Breaking in Non-Abelian Fundamental-Higgs Models

We now consider non-Abelian gauge groups $SU(N)$ (and by trivial generalization $SO(N)$) with fundamental Higgs fields. As a convenient shorthand, in this section we will represent the Higgs vectors as kets, $\phi_i \equiv |\phi_i\rangle$, which should not be confused with quantum states. Furthermore, we suppress the subscript "f" on the fundamental representation matrices $U_\ell$. The action we write as

$$S_{\text{bulk}} = -\beta \sum_{p \in X} \frac{1}{N} \text{Tr} \prod_{\ell \in \partial p} U_\ell - \kappa \sum_{\langle ij \rangle \in X} \text{Re} \langle \phi_i | U_{ij} | \phi_j \rangle. \tag{61}$$

The boundary action is given by Eq. (42). For $N > 2$ the Higgs rotors can no longer be identified as group elements. Because of this, the $\kappa \to \infty$ constraint,

$$\text{Re} \langle \phi_i | U_{ij} | \phi_j \rangle = 1, \tag{62}$$

does not completely trivialize the bulk. To see this, note that the constraint enforces that nearest-neighbor Higgs rotors are parallel relative to the gauge field. However, since there is still freedom to perform rotations about the colinear axis of the remaining $N - 1$ components of the Higgs field, the gauge field can continue to fluctuate so long as it does not rotate the Higgs field away from the parallel axis. This is explicitly seen by rotation to unitary gauge where the Higgs field is parallel in a global frame, which then fixes one diagonal element of the gauge field to unity, i.e.

$$\phi_i = \begin{pmatrix} 1 \\ 0 \\ \vdots \end{pmatrix} \quad \xrightarrow{\kappa \to \infty} \quad (U_{ij})^{11} = 1 \qquad \text{(unitary gauge)}. \tag{63}$$

This constraint forces the link matrices to take the form

$$U_{ij}^{SU(N)} \xrightarrow{\kappa \to \infty} \begin{pmatrix} 1 & 0 \\ 0 & U_{ij}^{SU(N-1)} \end{pmatrix} \qquad \text{(unitary gauge)}. \tag{64}$$

Thus in this limit the bulk theory becomes (gauge-equivalent to) an $SU(N-1)$ gauge theory. Note that $SU(1)$ is trivial, making the $SU(2)$ case special, as discussed in Section 3.2.

### 3.4.1  Gauge-Invariant Resolution of the Infinite Kappa Limit

To resolve the constraint in a fully gauge-invariant fashion analogous to Eqs. (22) and (60), we first note that the constraint Eq. (62) actually implies that

$$\langle \phi_i | U_{ij} | \phi_j \rangle = 1, \tag{65}$$

which follows from the fact that real part of a Hermitian inner product on $\mathbb{C}^N$ is the Euclidean product when the vector space is viewed as $\mathbb{R}^{2N}$. In other words, $|\phi_i\rangle$ and $U_{ij}|\phi_j\rangle$ have the same real and imaginary components, and so are the same complex vector. We can think of $|\phi_i\rangle$ and $|\phi_j\rangle$ as unit vectors spanning a two-dimensional complex vector space, in which case $U_{ij}$ must act within this subspace as the unique $SU(2)$ rotation $\tilde{U}_{ij}$ rotating $|\phi_j\rangle$ to $|\phi_i\rangle$.[4] Therefore we can resolve the constraint as

$$U_{ij} = u_i \tilde{U}_{ij} u_j, \tag{66}$$

---

[4]Within the two-dimensional subspace this rotation is given by $\varphi_i^{\text{f}}(\varphi_j^{\text{f}})^{-1}$, using the notation of Eq. (49). That this rotation is unique follows from the fact that the two vectors have unit norm within this $\mathbb{C}^2$ subspace, thus they live on a 3-sphere, and $SU(2)$ is isomorphic to the 3-sphere, so each point on the 3-sphere corresponds to a unique $SU(2)$ rotation.

687  where $u_i$ is an SU($N$) matrix which preserves $|\phi_i\rangle$, i.e. an SU($N-1$) rotation in the subspace
688  orthogonal to $|\phi_i\rangle$. Note that the $u_i$'s are independent for every link, i.e. they are associated to
689  the ends of the links and are independent on different links touching the same site. Further-
690  more, there is only one independent SU($N-1$) degree of freedom on each link, because $\tilde{U}_\ell u$
691  is equivalent to $(\tilde{U}_\ell u \tilde{U}_{-\ell}) \tilde{U}_\ell$. Thus the constraint reduces each link variable to an SU($N-1$)
692  degree of freedom, and the whole theory reduces to an SU($N-1$) gauge theory.

693     While Eq. (66) demonstrates the reduction of the gauge group, it is not that useful for
694  formulating the boundary theory. A more useful way is to decompose each $U_{ij}$ by sandwiching
695  it between two resolutions of the identity decomposed into the parallel and perpendicular
696  subspace of $|\phi_i\rangle$ and $|\phi_j\rangle$. Namely, for site $i$

$$\mathbb{1} = P_i + |\phi_i\rangle\langle\phi_i|, \tag{67}$$

697  where $P_i$ is the projector to the orthogonal complement of $|\phi_i\rangle$. Note that under a gauge trans-
698  formation $P_i \to g_i P_i g_i^{-1}$. Inserting this identity on either side of a link variable decomposes it
699  into four pieces,

$$(P_i + |\phi_i\rangle\langle\phi_i|) U_{ij} (P_j + |\phi_j\rangle\langle\phi_j|) =$$
$$P_i U_{ij} P_j + |\phi_i\rangle\langle\phi_i|U_{ij}|\phi_j\rangle\langle\phi_j|$$
$$+ |\phi_i\rangle\langle\phi_i|U_{ij}P_j + P_i U_{ij}|\phi_j\rangle\langle\phi_j|. \tag{68}$$

700  In the limit $\kappa \to \infty$, this simplifies significantly. Firstly, the two cross terms (the last line)
701  are exactly zero by Eq. (66), i.e. because $P_i U_{ij}|\phi_j\rangle = P_i|\phi_i\rangle = 0$. Second, in the second term,
702  Eq. (65) reduces it to $|\phi_i\rangle\langle\phi_j|$. In summary, in the limit $\kappa \to \infty$ every link variable in the bulk
703  can be expressed as

$$U_{ij} \to P_i U_{ij} P_j + |\phi_i\rangle\langle\phi_j|. \tag{69}$$

704  It follows that a product of two consecutive link variables is

$$U_{ij} U_{jk} \to P_i U_{ij} P_j U_{jk} P_k + |\phi_i\rangle\langle\phi_k| \tag{70}$$

705  where we used that $P_i^2 = P_i$ and $\langle\phi_j|\phi_j\rangle = 1$. Therefore gauge-invariant closed Wilson loops
706  have a projector to the orthogonal subspace inserted between each consecutive link,

$$\mathrm{Tr}\big[U_{ij} U_{jk} U_{kl} U_{li}\big] \to 1 + \mathrm{Tr}\big[P_i U_{ij} P_j U_{jk} P_k U_{kl} P_l U_{li}\big], \tag{71}$$

707  which is another manifestation of the Higgsing down to an SU($N-1$) gauge theory.

708     Now consider the three-legged plaquettes appearing in the boundary action, Eq. (42).
709  Inserting the identities at the two bulk sites, we obtain

$$\mathrm{Tr}\Big[U_i U_{ij} U_j^{-1}\Big] \xrightarrow{\kappa \to \infty} \mathrm{Tr}\Big[U_i \big(P_i U_{ij} P_j + |\phi_i\rangle\langle\phi_j|\big) U_j^{-1}\Big]$$
$$= \mathrm{Tr}\Big[U_i P_i U_{ij} P_j U_j^{-1}\Big] + \langle\Phi_i|\Phi_j\rangle \tag{72}$$

710  where we have defined the gauge-invariant variables

$$|\Phi_i\rangle \equiv U_i|\phi_i\rangle, \tag{73}$$

711  which are $\mathbb{C}^N$ unit vectors corresponding to the short half-open Wilson strings at the boundary.
712  This highlights an important distinction for fundamental Higgs compared to group-valued
713  Higgs models—here the half-open Wilson string is a vector degree of freedom, not group-
714  valued. The $\kappa \to \infty$ theory then can be expressed in the gauge-invariant form

$$S_{\mathrm{bulk}} \to -\frac{\beta}{N} \sum_{p \in X} \Big(1 + \mathrm{Re}\,\mathrm{Tr}\big[U_{ij} P_j U_{jk} P_k U_{kl} P_l U_{li} P_i\big]\Big),$$

$$S_{\mathrm{bdry}} \to -\frac{\beta}{N} \sum_{\langle ij\rangle \in \partial X} \Big(\mathrm{Re}\,\mathrm{Tr}\big[U_i P_i U_{ij} P_j U_j^{-1}\big] + \mathrm{Re}\,\langle\Phi_i|\Phi_j\rangle\Big). \tag{74}$$

Thus the bulk Higgses down to an $SU(N-1)$ gauge theory while the boundary decomposes into a pure $SU(N-1)$ part plus a part which may be viewed as an $SU(N)$ ferromagnet. The global $SU(N)$ boundary symmetry acts as $U_i \to gU_i$, under which $|\Phi_i\rangle \to g|\Phi_i\rangle$. The three-leg projected Wilson loop which appears in the boundary action in Eq. (74) is not charged under this symmetry since the trace cancels the contribution from the two ends. We therefore generically expect the boundary $SU(N)$ symmetry to spontaneously break above a critical $\beta$ down to $SU(N-1)$, and the short Wilson string rotors $|\Phi_i\rangle$ on the boundary to exhibit long range order and Goldstone modes. We discuss the nature of this phase transition further and provide numerical support for this statement in Section 3.5.

### 3.4.2 Formulation for General Gauge Groups

For a general gauge group $\mathcal{G}$ with fundamental Higgs, the large-$\kappa$ limit Higgses it down to a subgroup $\mathcal{H}$. We consider here cases where the residual gauge group $\mathcal{H}$ is non-Abelian. In the limit $\kappa \to \infty$ the bulk fluctuates as a pure gauge theory with gauge group $\mathcal{H}$ governed by the Wilson action, while the boundary links continue to explore the full gauge group $\mathcal{G}$. The generic picture (which can be obtained in unitary gauge) is that the action Higgses down to

$$S_{\kappa\to\infty} = -\beta \sum_{p\in X} \mathrm{Re\,Tr}\, W_p^{\mathcal{H}} - \beta \sum_{\langle ij\rangle \in \partial X} \mathrm{Re\,Tr}\Big[U_i^{\mathcal{G}} U_{ij}^{\mathcal{H}} (U_j^{\mathcal{G}})^{-1}\Big], \tag{75}$$

where $U^{\mathcal{H}}$ is the $\mathcal{G}$ link variable restricted to the $\mathcal{H}$ subgroup, and $W^{\mathcal{H}}$ is the corresponding Wilson plaquette for these $\mathcal{H}$-valued bulk links. Taking the traces in any faithful representation of the group should yield the same physics.

Assuming the bulk is gapped with a finite correlation length, as it must be if $\mathcal{H}$ is non-Abelian, the boundary is quasi-$(D-1)$-dimensional with some finite correlation length extending into the bulk. The system retains a $\mathcal{G}$ global symmetry rotating all of the boundary links, together with the bulk $\mathcal{H}$ gauge symmetry. The appropriate boundary theory is therefore expected to be a gauged nonlinear $\sigma$-model with target space $\mathcal{G}$ with subgroup $\mathcal{H}$ gauged, or equivalently, a nonlinear $\sigma$-model with target space the quotient space $\mathcal{G}/\mathcal{H}$. For example, $SU(N)$ Higgses down to $SU(N-1)$ and the quotient space is $SU(N-1)/SU(N) \simeq S^{2N-1}$, which agrees with the finding in Eq. (74) of a boundary theory of $SU(N)$ rotors, whose configuration space is a sphere in $2N$ dimensions. Similarly, $SO(N)$ Higgses to $SO(N-1)$, with quotient space $S^{N-1}$.

### 3.4.3 Hamiltonian Perspective

Similar considerations apply to the Hamiltonian formulation of the non-Abelian Higgs theory described in Section 3.1.2. The analog of the large $\kappa$ limit in the Hamiltonian formulation (Eq. (33)) is first of all to drop the conjugate variables $\hat{Q}_i^m$ leaving only the $\hat{\Lambda}$ variables in the matter sector. This essentially renders the Higgs fields classical and they may be gauge fixed without loss of generality along some fixed direction $\phi^\alpha = \delta^{\alpha 1}$. There is now a Hamiltonian constraint

$$-\kappa \sum_\ell \hat{U}_\ell^{11} \tag{76}$$

that breaks gauge fluctuations from $SU(N)$ down to $SU(N-1)$.

Now if we consider only the boundary plaquettes, this constraint acts only on the bulk links parallel to the boundary while those extending out of the boundary have no such constraint. Therefore the boundary theory of the four dimensional bulk in the large $\kappa$ limit is a three dimensional $SU(N)$ chiral model that is partially gauged by an $SU(N-1)$ gauge group where the $SU(N-1)$ gauge theory permeates the bulk.

## 3.5 Boundary Phase Transition Order and Universality Class

So far we have demonstrated that Higgs models with group-valued or fundamental Higgs fields have large-$\kappa$ limits with well-defined boundary degrees of freedom that may exhibit symmetry breaking. Having discussed the $\kappa \to \infty$ boundary actions for the family of fundamental Higgs SU($N$) models we are in a position to say something about the phase transitions at the boundary. As the bulk is gapped for finite $\kappa$ we should expect that the boundary theory has a finite correlation length into the bulk making it a quasi-$(D-1)$-dimensional boundary theory so that statements at $\kappa \to \infty$ hold also for finite $\kappa$.

In the light of Refs. [22, 23] and our analysis above, the quasi-$(D-1)$-dimensional boundary theory typically should have a symmetry breaking phase transition as the boundary coupling is tuned. But what is the nature of the phase transition? When there are only global symmetries, the symmetry group and the spacetime dimension determine the type of the transition. When some symmetries are gauged, the full set of global symmetries (including higher-form symmetries originating from the gauging [4]) are expected to determine the nature of the phase transition, while the gauge redundancy only serves to reduce to the quotient space. For the coupled bulk-boundary models considered here, the bulk gap ensures the integrity of the boundary model across the phase diagram well away from bulk critical points. Having identified the physical gauge-invariant boundary variables on the lattice that can go critical, in this part we write down the corresponding Landau boundary theories.

### 3.5.1 Group-Valued Higgs

The boundary degrees of freedom, Eq. (59), are gauge-invariant composites transforming under the global $\mathcal{G}_L \times \mathcal{G}_R$ symmetry. Considering the case $\mathcal{G} = \mathrm{SU}(N)$, the coarse-grained (generally complex) matrix-valued fields are denoted $X_i^{ab}$ [48], where $a, b$ are color indices, which transform as $\mathbf{X} \to \mathbf{g}_L \mathbf{X} \mathbf{g}_R^\dagger$ under the symmetry. The Landau theory is

$$\mathcal{L} = \mathrm{Re}\,\mathrm{Tr}\left[\partial^\mu \mathbf{X} \partial_\mu \mathbf{X}^\dagger\right] + a\,\mathrm{Re}\,\mathrm{Tr}\left[\mathbf{X}^\dagger \mathbf{X}\right] + b\,\mathrm{Re}\,\mathrm{Det}\left[\mathbf{X}\right] + \cdots \tag{77}$$

The SU($N$) × SU($N$) symmetry is susceptible to breaking down to the diagonal subgroup. This set of models has been studied in Ref. [48] to which we refer for more details. One finds, in the case $N = 2$, that the determinant contributes to the quadratic term. Numerically one finds a continuous transition consistent with a quartic term stabilizing the free energy. In the case $N = 3$, the determinant is cubic implying that the transition is first order. In the case $N = 4$, the determinant contributes a quartic term but with a negative sign that is expected to drive the transition first order. We therefore expect a continuous transition for $N = 2$ and first order for $N = 3$ and for $N = 4$.

Chiral models on a lattice, such as the one in Eq. (60), have been studied for many years especially in two dimensions at large $N$, where they are integrable [49]. If one is interested in boundaries of four dimensional gauge theories, the three dimensional analogs of such models are of interest. One early work on $\mathcal{G} = SU(N)$ in three dimensions, Ref. [48], contains Monte Carlo results for $N = 2, 3, 4$. For $N = 2$ (Section 3.2) the symmetry group is SU(2) × SU(2) ≃ O(4) and the transition is continuous, consistent with O(4) criticality. For $N = 3, 4$ the numerical results reveal the transition to be first order in agreement with mean field theory predictions.

### 3.5.2 Fundamental Higgs

The boundary model in the case of a fundamental Higgs has gauge-invariant degrees of freedom of the form $\Phi_i^a \equiv U_i^{ab} \phi_i^b$ where $a, b$ are the color indices, Eq. (73). There is a single global $\mathcal{G}$ symmetry that acts from the left $\Phi \to g\Phi$. This will be broken spontaneously for sufficiently

large $\beta$ (modulo Mermin-Wagner restrictions). The coarse-grained version of $\Phi^a$ is denoted $\Psi^a$ and the Landau theory for SU($N$) is

$$\mathcal{L} = \partial^\mu \Psi^\dagger \partial_\mu \Psi + a\Psi^\dagger \Psi + b(\Psi^\dagger \Psi)^2 + \dots \tag{78}$$

which is invariant under U($N$) transformations. This may instead be viewed as a theory of $2N$ real variables invariant under the enlarged O($2N$) symmetry group. This Landau theory gives the impression that, for the three dimensional boundary of a four dimensional SU($N$) gauge theory, there is a phase transition in the O($2N$) universality class. In the case of SU(2) this is O(4) criticality as shown above at the level of the microscopic model both analytically and numerically. Note that the coupling of the $\Phi_i$ rotors in Eq. (74) is invariant under O($2N$) rotations, even though the microscopic action manifestly only has a global SU($N$) symmetry.

It may seem surprising that the SU($N$) invariant model exhibits O($2N$) criticality. One simple check is that SU($N$) has $N^2 - 1$ generators and that SU($N$) $\to$ SU($N-1$) symmetry breaking therefore has $2N - 1$ broken generators (corresponding to the number of Goldstone modes) which matches the count for O($2N$) $\to$ O($2N-1$) symmetry breaking where O($N$) has $N(N-1)/2$ generators. But what about the terms that are SU($N$) invariant but not O($2N$) invariant? These terms are certainly present. An example is given by taking an operator $\Xi_{ab} \equiv \Psi_a \Psi_b^\dagger$ which transforms as $g\Xi g^\dagger$ and considering its determinant. This is only U($N$) invariant, but it is also an irrelevant operator for $N > 2$ in $D = 4$. More precisely, the couplings in the action originating from the determinant have mass dimension $[g_{\text{det}}] = D + N(2 - D)$ which, in dimensions higher than two, is negative for all but $N = 2, 3$ in three dimensions and $N = 2$ in four dimensions. As we have seen, the case of SU(2) is special because it is identical to a problem with a group valued Higgs. So we have treated it separately. Therefore the remaining puzzle relates to SU(3) in three dimensions where the determinant coupling goes like $|\Psi^\dagger \Psi|^3$ and is marginal by power counting. As with the problem of scalar field theory in three dimensions [50], among other cases, we expect this sixth order term to be marginally irrelevant. Taking this together with our results for SU(2) we surmise that the SU($N$) boundary theory phase transition is in the O($2N$) universality class since terms breaking O($2N$) down to SU($N$) are irrelevant or marginally irrelevant.

### 3.5.3 Fate of the Un-Higgsed Subgroup: SU(3) Numerical Results

The discussion above for the fundamental Higgs assumed that the relevant critical degrees of freedom on the boundary are the $\Phi_i$, Eq. (73). One may be concerned, however, about the residual fluctuations from the un-Higgsed SU($N-1$) part of the gauge group. In particular, the $\Phi_i$ are composite degrees of freedom between a boundary link and its attached Higgs field, and that the boundary link also appears in the second term in the boundary action, Eq. (74), a three-leg Wilson loop with projectors at the two bulk sites. However, since this term is not charged under the global SU($N$) symmetry acting on the boundary links from the outside, we expect it to be irrelevant to the boundary criticality.

To verify this and our prediction of O($2N$) criticality, we have performed simulations of 4D SU(3) gauge theory with fundamental Higgs. The Higgs field $\phi_i$ is a 3-component complex unit vector at each site, and we measure the average of the Higgs field measured from the vacuum end of the boundary links, Eq. (73), along with the boundary Wilson loops,

$$\Phi = \frac{1}{L^3} \sum_{i \in \partial X} (U_i \phi_i) \in \mathbb{C}^3 \tag{79}$$

$$W_P = \frac{1}{3L^3} \sum_{\langle ij \rangle \in \partial X} \frac{1}{3} \text{Re Tr}\left[ U_i P_i U_{ij} P_j U_j^{-1} \right]. \tag{80}$$

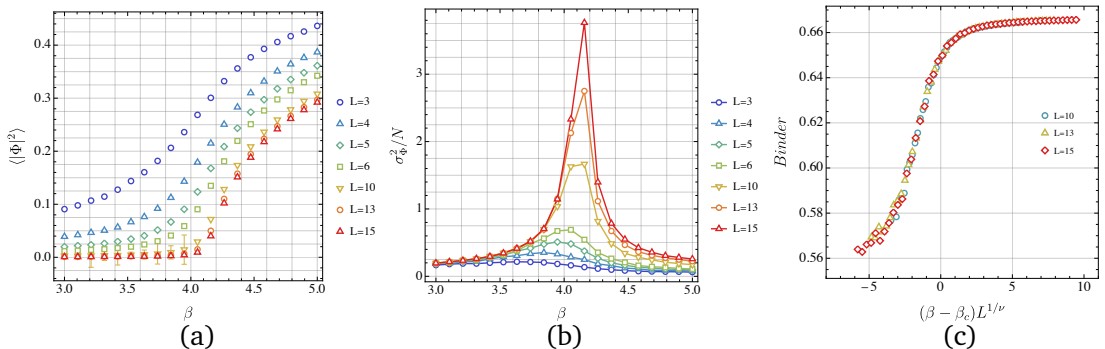

Figure 9: Monte Carlo results for the boundary phase transition for the SU(3) fundamental-Higgs model. (a) Boundary order parameter as a function of $\beta$ at $\kappa = 3.0$. (b) Boundary susceptibility for the same $\kappa$ showing a critical $\beta$ of $\beta_c \approx 4.19$. (c) We use the previous $\beta_c$ together with the 3D O(6) critical exponent $\nu = 0.789$ from Ref. [51] to plot the Binder cumulant, showing a clear scaling collapse.

The numerical results for the average Higgs field are presented in Fig. 9 for $\kappa = 3.0$. Figure 9(a) shows its behavior for various system sizes, while Fig. 9(b) shows the behavior of its variance. The behavior is consistent with that of an order parameter at a second order phase transition, approaching zero at small $\beta$ and continuously increasing from $\beta > \beta_c \approx 4.19$, with a diverging susceptibility. Figure 9 shows the Binder cumulant scaled by the 3D O(6) critical exponent $\nu \approx 0.789$ [51], which shows a clear collapse to a universal scaling function. Furthermore, Fig. 10 shows the behavior of $W_P$, which shows no system-size dependence and smooth behavior, verifying that this term is irrelevant at the transition. We therefore conclude that the SU(3) Higgs model in $D = 4$ indeed demonstrates a boundary phase transition in the 3D O(6) universality class, even though a single fundamental Higgs field does not remove all bulk degrees of freedom when $\kappa \to \infty$, in agreement to the general picture of boundary O(2N) criticality on the boundary of SU(N) fundamental Higgs models.

## 3.6 Summary and Discussion: Non-Abelian Case

In this section we have discussed global boundary symmetries in non-Abelian Higgs models and their spontaneous symmetry breaking with both fundamental Higgs fields and group-valued Higgs fields. For general non-Abelian gauge group $\mathcal{G}$ there is a global $\mathcal{G}$ symmetry acting on the boundary of the system, which is equivalent by the Gauss law to the total color charge of the system, Eq. (48). For the group-valued case, there is an additional bulk matter symmetry given by right multiplication of the matter field. If $\mathcal{G}$ is an Abelian group, these two symmetries exactly coincide, since there is no distinction between left and right multiplication. For $\mathcal{G} = $ SU(2), the two types of models are equivalent due to fact that SU(2) is equivalent to the unit quaternions. For other gauge groups the two types of models are inequivalent. In all cases, the location of the boundary transition that we identify can be shifted by tuning the boundary coupling.

In the group-valued case, the infinite $\kappa$ limit freezes the bulk of the system. The resulting boundary theory can be expressed in terms of gauge-invariant half-open Wilson strings at the boundary, Eq. (60). This may be viewed as a lattice discretization of a principal chiral model, with $\mathcal{G} \times \mathcal{G}$ symmetry coming from the boundary color flux and bulk matter symmetries. This global symmetry is expected to be broken to the diagonal subgroup at large $\beta$. Since the boundary is frozen in the $\kappa \to \infty$ limit and should remains strongly gapped at large $\kappa$, the boundary transition is expected to persist also for finite $\kappa$. Our SU(2) Monte Carlo simulations verify this, and the boundary transition line appears to terminate at the bulk critical endpoint.

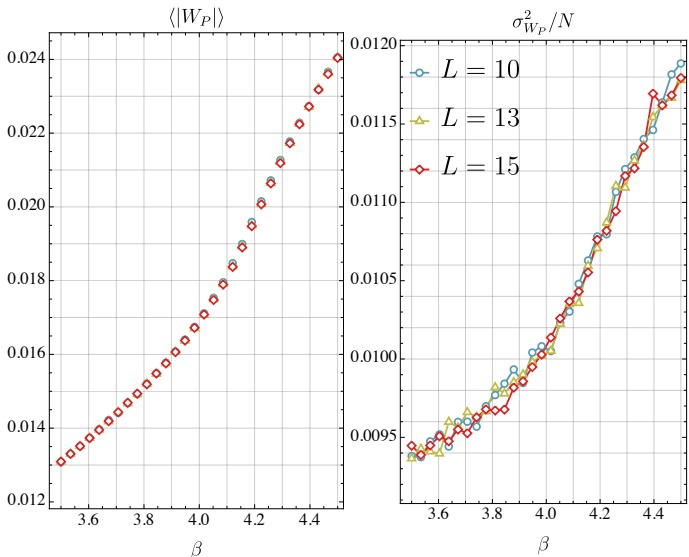

Figure 10: The expectation and variance of the boundary SU($N − 1$) Wilson loop, Eq. (80), which appears as one of the two terms in the $\kappa \to \infty$ limit of the fundamental-Higgs SU($N$) boundary action in Eq. (74). This quantity shows no system-size dependence, indicating it plays no role in the boundary phase transition (cf. the boundary order parameter in Fig. 9).

For Higgs fields in the fundamental representation, there is only a single global $\mathcal{G}$ symmetry, generated by the total color charge of the system and acting on the boundary. The general picture is that the bulk $\mathcal{G}$ gauge symmetry is Higgsed down to a subgroup $\mathcal{H}$. If $\mathcal{H}$ is trivial, then the bulk is completely frozen in the $\kappa \to \infty$ limit. If $\mathcal{H}$ is Abelian, there may be a bulk phase transition even at infinite $\kappa$ (though that does not preclude a boundary symmetry breaking). If $\mathcal{H}$ is non-Abelian then the bulk reduces to a gapped $\mathcal{H}$ gauge theory as $\kappa \to \infty$. The boundary theory is expected to be a gauged nonlinear $\sigma$-model with target space $\mathcal{G}/\mathcal{H}$, Eq. (75). Note that by placing the Higgs in other representations (e.g. adjoint) or adding addition Higgs fields, one may target different subgroups $\mathcal{H}$ [52] and thus obtain different boundary theories. For example, by starting from an SO($N$) gauge theory with $M$ Higgs fields in the fundamental representation, the general expectation is that it reduces to an SO($N − M$) gauge theory in the bulk, and the boundary target space is the Stiefel manifold SO($N$)/SO($N − M$), which have recently attracted significant interest [53]

We considered in particular the case SU($N$) (and by direct extension SO($N$)). In the $\kappa \to \infty$ limit, the bulk Higgses down to a SU($N − 1$) gauge theory, while the boundary decomposes into a pure SU($N − 1$) part and an SU($N$) rotor part. The global symmetry, under most circumstances, will be broken spontaneously at large $\beta$. We argued that this SU($N$) fundamental Higgs boundary phase transition lies in the O($2N$) universality class, since there are no relevant operators differentiating SU($N$) from O($2N$). We tested this prediction numerically for the SU(3) lattice gauge theory with fundamental Higgs, demonstrating a clean scaling collapse with O(6) critical exponents.

One interesting exceptional case is SO(3) gauge theory with fundamental Higgs. This is equivalent to the Georgi-Glashow electroweak theory, with gauge group SU(2) and the Higgs field in the adjoint representation. In the $\kappa \to \infty$ limit we can fix to unitary gauge which Higgses the bulk gauge group down to SO(2) $\simeq$ U(1). In $D = 4$ and $\kappa \to \infty$, the bulk has a confinement-deconfinement transition as a function of $\beta$, while we predict an additional O(3) boundary critical point in the Higgs regime. This may have implications for the physics of

899   domain walls or cosmic strings formed in the early universe.

## 4   Boundary Symmetry Breaking in Higher-Form Abelian Higgs Models

902   Thus far we have discussed boundary symmetry breaking in Higgs phases of both Abelian and
903   non-Abelian gauge theories with 1-form gauge fields, drawing a general picture of a global
904   charge symmetry realized on the boundary due to the Gauss law constraint of a gauge-invariant
905   system. Here we consider a further extension, to higher-form gauge fields. A $k$-form gauge
906   field describes the parallel transport of charged $(k-1)$-dimensional objects [54–57]. Two-
907   form gauge fields are often called Kalb-Ramond fields in string theory literature [58], they
908   appear in the dual descriptions of superfluids and superconductors [59,60] and may be real-
909   ized in certain spin models on frustrated lattices [61]. The gauge group for $k > 1$ is generically
910   Abelian, because a unique path ordering only exists on 1-dimensional contours [54,55].

### 4.1   Higher-Form Abelian Higgs Models

912   Here we consider the case of $k$-form U(1) gauge theory, though restriction to $\mathbb{Z}_N$ subgroups
913   follows. Let $A$ be a $k$-form field and $\theta$ a $(k-1)$-form field, each taking values in $\mathbb{R}/2\pi\mathbb{Z}$ in
914   $D$ Euclidean spacetime dimensions, with $1 \le k \le D-2$.[5] By a $k$-form we mean a function $\omega$
915   on oriented $k$-dimensional cells $c$ of the lattice, such that $\omega(-c) = -\omega(c)$, where $-c$ denotes
916   the cell $c$ with the opposite orientation. See Appendix A for a more detailed discussion of the
917   discrete differential forms notation used throughout this section. Let $X_k$ denote the collection
918   of $k$-cells of the lattice, each with a fixed orientation. The fields are governed by the generalized
919   Fradkin-Shenker action

$$S_{\text{bulk}} = -\beta \sum_{c \in X_{k+1}} \cos(\mathrm{d}A)_c - \kappa \sum_{c' \in X_k} \cos(\mathrm{d}\theta - A)_{c'}. \tag{81}$$

920   The exterior derivative of a $k$-form $\omega$ is a $(k+1)$-form $\mathrm{d}\omega$ whose value is defined by the discrete
921   Stoke's theorem,

$$\mathrm{d}\omega_c = \sum_{c' \in \partial c} \omega_{c'}, \tag{82}$$

922   where the sum is over the $k$-cells forming the oriented boundary of the $(k+1)$-cell $c$. This is
923   a straightforward generalization of the Abelian Higgs model, Eq. (8), to a theory of extended
924   $(k-1)$-dimensional charged objects attached to $k$-dimensional electric flux branes [54].
925       The action is invariant under higher-form gauge transformations,

$$\theta \to \theta + \lambda, \quad A \to A + \mathrm{d}\lambda, \tag{83}$$

926   for an arbitrary $(k-1)$-form $\lambda$. We will see that this enforces the Gauss law attaching electric
927   branes to the charged objects. When $k > 1$, this gauge invariance includes "gauge-of-gauge"
928   transformations

$$\theta \to \theta + \mathrm{d}\alpha, \tag{84}$$

929   for arbitrary $(k-2)$-form $\alpha$. This yields an additional Gauss-type constraint which enforces
930   that the $(k-1)$-dimensional electrically charged objects are closed.[6]

---

[5]If $k = 0$ there is no gauge field. If $k = D-1$ then the gauge field is not dynamical and can be completely
integrated out using the Gauss law, yielding long-range interactions for the Higgs field. If $k = D$ then $\mathrm{d}A = 0$
identically.

[6]In the continuum formulation these extra constraints are the pure-spatial components of the conserved higher-
form Noether current [23].

When $\kappa = 0$, Eq. (81) reduces to a pure $k$-form U(1) gauge theory, which has a global electric $k$-form symmetry given by

$$A \to A + \lambda \quad \text{with} \quad \mathrm{d}\lambda = 0, \tag{85}$$

corresponding to conservation of global electric flux. In $D$ spacetime dimensions, it also admits magnetic homotopy defects, whose cores trace out $k_{\mathrm{m}}$-dimensional worldsheets in space-time [54, 55, 62], where

$$k_{\mathrm{m}} = D - (k + 2). \tag{86}$$

In the limit $\beta \to \infty$ it has a $k_{\mathrm{m}}$-form magnetic symmetry, discussed further in Section 4.2.

The phase diagram of this model is expected to be similar to that of the 1-form U(1) gauge theory, sketched in Fig. 1, as long as $D > k + 2$ ($k_{\mathrm{m}} > 0$). In the marginal case $D = k + 2$ ($k_{\mathrm{m}} = 0$), the magnetic defect is an instanton (zero dimensional in spacetime) and is expected to destabilize the deconfined phase [63–65]—a generalization of the Polyakov mechanism for 1-form U(1) gauge theory in $D = 3$ [66], which itself may be viewed as a higher-form generalization of the Mermin-Wagner theorem for 0-form symmetries [4, 5]. In those marginal cases, the bulk phase diagram should be qualitatively similar to that of the non-Abelian models as in Fig. 5 [63, 67, 68].

### 4.1.1 Gauss Law and Matter Symmetry

Before we introduce the Hamiltonian formulation and discuss matter symmetry, we point out that the Higgs charges for $k > 1$ behave slightly differently than for $k = 1$. When $k = 1$, the zero-dimensional point charges must come in pairs at the two ends of oriented electric strings, and are either positive or negative depending on which end of the string they sit. When $k > 1$ the charges are extended oriented objects (e.g. strings when $k = 2$) living on the edges of electric $k$-branes. Such a brane for $k > 1$ can have a single edge, meaning that it is perfectly valid to have a single charged object in the system. As such the extended charged objects are net-charge-neutral if they are contractible [54, 55]. It is only if they wrap around periodic boundaries that they have non-trivial global charge.

The Hamiltonian formulation follows the exact arguments laid out in Section 2.1, with conjugate operators $[\hat{\theta}, \hat{n}] = i$ on all $(k-1)$-cells and $[\hat{A}, \hat{E}] = i$ on all $k$-cells. Invariance of physical states under the gauge transformations, Eq. (83), enforces the Gauss law

$$-(\mathrm{d}^\dagger \hat{E})_c \equiv -\sum_{c' \in \partial^\dagger c} \hat{E}_{c'} = \hat{n}_c \quad (c \in X_{k-1}), \tag{87}$$

where $c$ is a $(k-1)$-cell and the sum is over its coboundary, the set of $k$-cells $c'$ containing $c$ in their positively-oriented boundary. This simply says that the number of electric $k$-branes emanating from $c$ is equal to the amount of electric charge on $c$. For $k > 1$, the charges carry a sense of orientation. For example, when $k = 2$ the charges are nothing but the electric strings of a 1-form gauge field. The "gauge-of-gauge" symmetry, Eq. (84), enforces the constraint

$$\mathrm{d}^\dagger \hat{n} = 0, \tag{88}$$

which says that the charged objects are closed.[7]

In the $k = 1$ theory discussed in Section 2 with 0-form Higgs field, the net charge of the Higgs field generates a global 0-form symmetry, which is pure gauge with periodic boundaries

---

[7] This constraint obviously follows from the Gauss law Eq. (87) (following from $\mathrm{d}^2 = 0$), just as Eq. (84) is already implied by Eq. (83), but it is worth spelling out for those unfamiliar with this point.

but becomes physical with the choice of electric boundary conditions. The natural extension of this global matter symmetry for general $k$ is a $(k-1)$-form symmetry,

$$\theta \to \theta + \lambda \quad \text{with} \quad d\lambda = 0. \tag{89}$$

Letting $d = D - 1$ be the dimension of space, the generators of these transformations are Gukov-Witten operators [4] supported on $(d-(k-1))$-dimensional closed surfaces $\tilde{\Sigma}$ in the dual lattice

$$\hat{Q}(\tilde{\Sigma}) = \sum_{\tilde{c} \in \tilde{\Sigma}} \hat{n}_c \equiv \langle \hat{n}, \delta_{\tilde{\Sigma}} \rangle, \tag{90}$$

where $c$ is the $(k-1)$-cell in the direct lattice corresponding to $\tilde{c}$ in the dual lattice, and $\delta_{\tilde{\Sigma}}$ is a $(k-1)$-form Poincaré dual to $\tilde{\Sigma}$.[8] These operators generate the symmetry, Eq. (89),

$$e^{-i\alpha \hat{Q}(\tilde{\Sigma})}|\theta\rangle = |\theta + \alpha \delta_{\tilde{\Sigma}}\rangle, \tag{91}$$

where $\lambda \equiv \alpha \delta_{\tilde{\Sigma}}$ with $\alpha$ a constant. Because $\tilde{\Sigma}$ is a closed surface its Poincaré dual is a closed form, $d\delta_{\tilde{\Sigma}} = \delta_{\partial \tilde{\Sigma}} = 0$.

These operators are topological, i.e. they only depend on the homology class of $\tilde{\Sigma}$, owing to the matter Gauss law Eq. (88). Consider replacing it with another surface such that $\tilde{\Sigma}' - \tilde{\Sigma} = \partial \tilde{V}$ for some $(d-(k-2))$-volume $\tilde{V}$. Then $\delta_{\tilde{\Sigma}'} = \delta_{\tilde{\Sigma}} + d\delta_{\tilde{V}}$. Plugged into Eq. (90), we have

$$\hat{Q}(\tilde{\Sigma}') = \langle \hat{n}, \delta_{\tilde{\Sigma}} \rangle + \langle d^\dagger \hat{n}, \delta_{\tilde{V}} \rangle = \hat{Q}(\tilde{\Sigma}), \tag{92}$$

where we used the matter Gauss law, Eq. (88). Therefore there is one $(k-1)$-form charge generator for each homology class in $H_{d-(k-1)}$. Deforming $\tilde{\Sigma}$ without changing its homology class corresponds to shifting $\theta$ by an exact form, which are just the gauge transformations of Eq. (84).

The operators charged under $\hat{Q}(\tilde{\Sigma})$ are the charge creation/annihilation operators, i.e. Wilson branes supported on open $k$-dimensional surfaces $M$ which insert an electric membrane with charge on its boundary,

$$e^{i\alpha \hat{Q}(\tilde{\Sigma})} e^{i(d\hat{\theta} - A)(M)} e^{-i\alpha \hat{Q}(\tilde{\Sigma})} = e^{-i\hat{A}(M)} e^{i\alpha \hat{Q}(\tilde{\Sigma})} e^{i\hat{\theta}(\partial M)} e^{-i\alpha \hat{Q}(\tilde{\Sigma})}$$

$$= e^{-i\hat{A}(M)} e^{i(\hat{\theta} + \alpha \delta_{\tilde{\Sigma}})(\partial M)}$$

$$= e^{i\alpha \#(\partial M, \tilde{\Sigma})} e^{i(d\hat{\theta} - \hat{A})(M)}, \tag{93}$$

where we have defined the intersection number between the $(k-1)$-dimensional $\partial M$ in the direct lattice and the $(d-(k-1))$-dimensional $\tilde{\Sigma}$ in the dual lattice,

$$\#(\partial M, \tilde{\Sigma}) = \delta_{\tilde{\Sigma}}(\partial M). \tag{94}$$

Thus the operator $\hat{Q}(\tilde{\Sigma})$ simply counts the intersection number of the closed charged objects with $\tilde{\Sigma}$.

In the Maxwell case, $k = 1$, the matter charges are point particles, $\tilde{\Sigma}$ is a closed $d$-dimensional volume in the dual lattice, and the only non-trivial choice is to take it to be all of space. The associated charge then simply counts the number of positive minus the number of negative charges. For a less trivial example, consider the case $k = 2$ and $d = 3$, so that the matter charges are 1-dimensional strings and $\tilde{\Sigma}$ is a closed two-dimensional surface in the dual lattice, intersecting a collection of links in the direct lattice. A choice of its Poincaré dual is a

---

[8]For our purposes, the defining property of the Poincaré dual of a $(d-k)$-dimensional closed surface $\tilde{\Sigma}$ in the dual lattice is that it is a $k$-form in the direct lattice acting as a generalized delta function, for example the unit $k$-form supported on the direct lattice $k$-cells piercing $\tilde{\Sigma}$ with the appropriate orientation. Note that $\delta_{\tilde{\Sigma}}$ is only defined up to an exact form because $\tilde{\Sigma}$ is closed, i.e. it is a cohomology class.

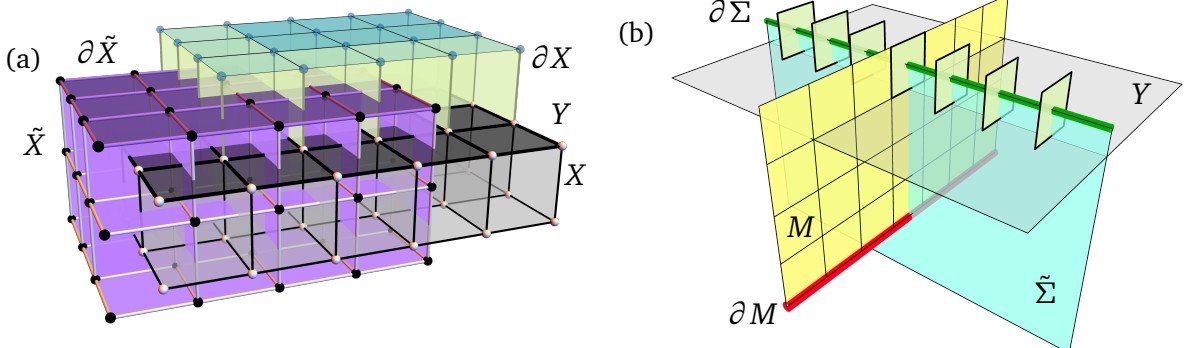

Figure 11: (a) An illustration of the relation between the direct lattice and dual lattice at the boundary of the system in $d = 3$ spatial dimensions. The direct lattice is colored the same as in Fig. 2. The dual lattice is indicated by black dual sites, white dual edges, and purple dual plaquettes. Each $k$-cell of the direct lattice corresponds to a $(d-k)$-cell in the dual lattice. Because the boundary $(\partial X)$ of the direct lattice $(X)$ is open with cells "sticking out" (green), the dual boundary $(\partial \tilde{X})$ is closed and has no protruding cells. We also color the $(D-1)$-dimensional edge layer $(Y)$ dark gray, which is where the $\kappa = \infty$ boundary action is defined, Eq. (102). (b) An illustration of a termination of $\tilde{\Sigma}$ on the boundary for a 1-form Higgs field coupled to a 2-form gauge field in $d = 3$ spatial dimensions. $\tilde{\Sigma}$ is a two-dimensional surface (cyan) which terminates at the boundary along a 1-dimensional contour (green). The Gauss law makes $\hat{Q}(\tilde{\Sigma})$ equal to the sum of the electric fluxes $\hat{E}_p$ on the boundary plaquettes pierced by the edge of $\tilde{\Sigma}$ (green squares with solid edges), Eq. (96). The edge $\partial \tilde{\Sigma}$ then intersects the half-open Wilson membranes attached to a charged string (red curve) attached to an electric membrane (yellow) exiting the system through the open boundary.

1-form which is zero everywhere except on the intersected links, on which it has unit value in the direction normal to the surface. The charge operator, Eq. (90), then counts (with signs, because $\hat{n}_{-c} = -\hat{n}_c$) the number of strings piercing the surface $\tilde{\Sigma}$, i.e. the intersection number between closed strings and the surface. Because the charges are closed strings, this intersection number must be zero unless $\tilde{\Sigma}$ winds around a periodic boundary and intersects a charge which also winds around the periodic boundaries in the transverse direction.

In a closed system, gauge invariance guarantees that the matter charge operators are exactly zero because of the Gauss law, Eq. (87), which when inserted into Eq. (90) yields $\langle \hat{E}, \mathrm{d}\delta_{\tilde{\Sigma}} \rangle = 0$. Equivalently, the intersection numbers, Eq. (94), are exactly zero because $\partial M$ has trivial homology class, i.e. $\delta_{\tilde{\Sigma}}(\partial M) = \mathrm{d}\delta_{\tilde{\Sigma}}(M) = 0$ since $\tilde{\Sigma}$ is closed. Consider for example the case $k = 2$, where the charge operators count the number of charged strings wrapped around a periodic boundary. We cannot, in a closed system, have a single non-contractible charged string, because it is attached to an electric membrane which must end somewhere inside the system. The only possibility is that it ends on another non-contractible charged string going the other direction, such that the two strings constitute the boundary of the electric membrane. This is the sense in which a closed system is charge-neutral when $k > 1$.

### 4.1.2 Open Boundaries and Global Matter Symmetry

To make the charge operators non-trivial, we must impose boundary conditions such that either $\tilde{\Sigma}$ can terminate on the boundary, and $\delta_{\tilde{\Sigma}}$ is not a closed form, or such that the contour $\partial M$ can terminate at the surface and thus not be closed. The latter case, however, means that the charged objects can pass through the boundary, meaning charge is not conserved. Therefore,

extending our results from Section 2, we take the former case. We consider open boundaries in the same form as Fig. 2. To be precise, the boundary consists of a layer of $D$-dimensional hypercubes, with all cells in the boundary layer which do *not* touch a bulk cell removed (or, equivalently, $A = 0$ and $\theta = 0$ on those cells). In other words, we remove the cells on the vacuum side, creating links missing one end, plaquettes missing one edge, cubic cells missing one face, etc. The boundary action is given by

$$S_{\text{bdry}} = -\beta \sum_{c \in \partial X_{k+1}} \cos(\text{d}A)_c, \tag{95}$$

where $\partial X_{k+1}$ denotes the set of $(k+1)$-cells in the boundary layer. As in Section 2, this boundary action does not allow the matter field $\theta$ to tunnel through the boundary meaning that all the charged objects are closed and contained inside the bulk.

In the presence of these boundary conditions, the matter charge operators, Eq. (90), can generate a physical symmetry. For $k > 1$ we must take care to consider how the surfaces $\tilde{\Sigma}$, which live in the dual lattice, can terminate at the boundary. Because we chose a "rough" boundary, as in Fig. 2, with a layer of cells sticking out from the edge of the system, the dual lattice boundary is flat. This is illustrated in Fig. 11(a) which shows the direct lattice bulk and boundary as in Fig. 2, along with the dual lattice in red and purple. The bulk part of the dual lattice is colored purple, while the boundary layer of the dual lattice is colored red and can be seen to form a flat surface without any protruding cells. The charge operators $\hat{Q}(\tilde{\Sigma})$ can terminate on this flat dual boundary layer.

For concreteness, consider the case $k = 2$ in $d = 3$, in which case the matter charges are strings and $\tilde{\Sigma}$ is a two-dimensional membrane in the dual lattice. Due to our boundary conditions the charge strings cannot terminate at the boundary, but the electric membranes can exit the system through the boundary. Referring to Fig. 11(b), consider a state with a single charged string wrapping around a periodic direction, as shown in Fig. 11(b) by the red line, attached to an electric membrane which exits through the boundary, illustrated by the yellow surface. This charged string is detected by taking the membrane $\tilde{\Sigma}$ to intersect it transversely, as illustrated by the cyan surface. The charge associated to the surface, Eq. (90), is then related to the electric flux through the boundary via the Gauss law,

$$\hat{Q}(\tilde{\Sigma}) = \langle -\text{d}^\dagger \hat{E}, \delta_{\tilde{\Sigma}} \rangle = -\langle \hat{E}, \text{d}\delta_{\tilde{\Sigma}} \rangle = -\langle \hat{E}, \delta_{\partial \tilde{\Sigma}} \rangle \equiv \hat{Q}_{\text{bdry}}(\partial \tilde{\Sigma}). \tag{96}$$

In the figure, $\partial \tilde{\Sigma}$ is shown as a dark green line which pierces a collection of boundary plaquettes (green squares with black borders). According to Eq. (96), the amount of charge measured by $\hat{Q}(\tilde{\Sigma})$ is equal to the number of electric branes exiting through the boundary measured by the set of plaquettes pierced by $\partial \tilde{\Sigma}$. In summary, the physical matter symmetry is generated by charge operators supported on $\tilde{\Sigma}$ which terminate at the boundary and, by the Gauss law, acts on the gauge field $A$ on the boundary elements pierced by $\partial \tilde{\Sigma}$.

### 4.1.3 Boundary Symmetry Breaking

Returning now to the higher-form gauge-Higgs action, Eq. (81), let us now see how the physical charge symmetry is spontaneously broken at the boundary. In the limit $\kappa \to \infty$ we have the constraint $A = \text{d}\theta$ on every bulk $k$-cell. This completely freezes the bulk degrees of freedom, as is seen by rotating to unitary gauge, $\theta = 0$, resulting in $\text{d}A = 0$. A covariant way to see this is that the gauge field operators $A(M)$ for $k$-dimensional closed surfaces $M$ trivialize—if $M$ is contained entirely within the bulk, then $A(M) = \text{d}\theta(M) = \theta(\partial M) = 0$ since $M$ is closed. This constraint is not imposed on the field variables on the cells touching the vacuum, however. As a result, if $M$ exist through the boundary, then we can decompose it into two pieces, $M = M|_X + M|_{\partial X}$, where $M|_X$ is the part of $M$ supported on bulk cells and $M|_{\partial X}$ is the part

$k = 1$, 0-Form Higgs Field

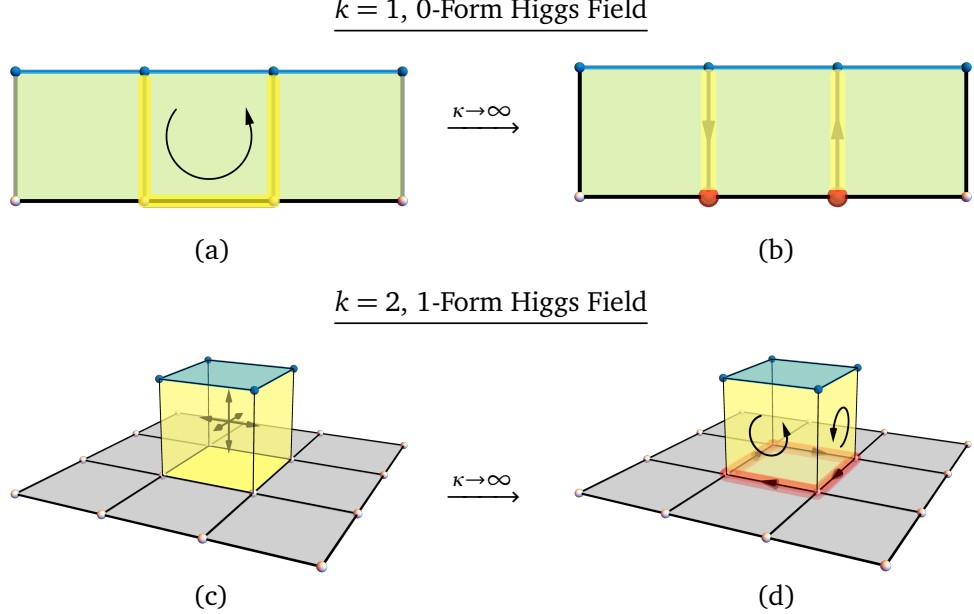

(a) (b)

$k = 2$, 1-Form Higgs Field

(c) (d)

Figure 12: An illustration of how the $\kappa \to \infty$ constraint $A = \mathrm{d}\theta$ turns closed membrane operators exiting through the boundary into open membrane operators terminating on a Higgs operator at the boundary. Each boundary $(k+1)$-cell $c$ has one bulk $k$-cell $c'$ in its boundary. The constraint turns $A(c')$ on this $k$-cell into $\mathrm{d}\theta(c') = \theta(\partial c')$. The end result is that all operators of the form $A(M)$ where $M$ exits the system through the boundary and is closed in the bulk are turned into open operators terminating on Higgs operators at the boundary, Eq. (98). We show (a,b) the case $k = 1$ with a 0-form Higgs field, where (a) a string operator exiting the system at both ends (yellow), (b) turns into a pair of half-open string operators terminating on Higgs operators (red); and (c,d) the case $k = 2$ with a 1-form Higgs field, where (c) a closed membrane operator exiting the system (yellow), (d) turns into a "half-open" membrane terminating on a Higgs string (red).

supported on boundary cells. Since $\partial M = 0$, we have $\partial M|_{\partial X} = -\partial M|_X$. As a result, the gauge field operators decompose as

$$
\begin{aligned}
A(M) \xrightarrow{\kappa \to \infty} &A(M|_{\partial X}) + \mathrm{d}\theta(M|_X) \\
=&A(M|_{\partial X}) + \theta(\partial M|_X) \\
=&A(M|_{\partial X}) - \theta(\partial M|_{\partial X}) \tag{97} \\
=&(A - \mathrm{d}\theta)(M|_{\partial X}). \tag{98}
\end{aligned}
$$

In other words, for closed surfaces $M$ exiting through the boundary, the $\kappa \to \infty$ constraint $A = \mathrm{d}\theta$ reduces $A(M)$ to a half-open Wilson operator terminating on Higgs operators as soon as it touches the bulk. Similarly, longer half-open Wilson surfaces ending on the Higgs field in the bulk, e.g. Eq. (16), are reduced to short half-open Wilson surfaces ending on the Higgs field at the boundary.

This is depicted in Fig. 12 for the cases $k = 1$ and $k = 2$, where $M$ is taken to be the smallest Wilson surfaces which are closed in the bulk, $M = \partial c$ for $c \in \partial X_{k+1}$. In the case $k = 1$, depicted in Fig. 12(a), $c$ is an oriented boundary plaquette (green), and $\partial c$ consists of three links (yellow), one of which is in the bulk. We decompose $\partial c$ into the two links in the boundary, denoted $\partial c|_{\partial X}$, and the one link in the bulk, denoted $\partial c|_X$. The gauge field on the

1072 link in the bulk trivializes by the constraint $A = d\theta$ into $\theta$ evaluated at its two endpoints, i.e.
1073 $\theta(\partial(\partial c|_X))$, depicted as red spheres in Fig. 12(b). These are the degrees of freedom appearing
1074 in the boundary action in Eq. (22). In the case $k = 2$, Fig. 12(c), $c$ is a three-dimensional cube
1075 and $\partial c$ is a set of five plaquettes (yellow). In Fig. 12(d), it decomposes into four plaquettes
1076 in the boundary (yellow), $\partial c|_{\partial X}$, and one plaquette in the bulk, $\partial c|_X$. The constraint turns
1077 the gauge field on this bulk plaquette into a Higgs string operator on its boundary (red). The
1078 extension to larger $k$ is obvious, but can't be illustrated.

1079     Using this, we can see that the boundary action Eq. (95) contains precisely these minimal
1080 operators. In the general case, the action reduces in this limit to

$$
\begin{aligned}
S_{\kappa\to\infty}^{\text{bdry}} &= -\beta \sum_{c \in \partial X_{k+1}} \cos(dA)_c \quad (A_{\text{bulk}} = d\theta) \\
&= -\beta \sum_{c \in \partial X_{k+1}} \cos(A - d\theta)_{\partial c|_{\partial X}}.
\end{aligned}
\tag{99}
$$

1081 We have already seen in Section 2 how in the case $k = 1$ this can be recast as a 0-form XY
1082 model, Eq. (22), which we can now extend to the case $k > 1$. Note that every boundary
1083 $k$-cell is associated to a unique bulk $(k-1)$-cell (the one which trivializes under the $A = d\theta$
1084 constraint). We define composite degrees of freedom for each such pair,

$$
\vartheta(c) = A_{\partial X}(c) - \theta(c) \quad (c \in \partial X_{k-1}),
\tag{100}
$$

1085 where $A_{\partial X}(c)$ is $A$ evaluated on the unique boundary cell corresponding to $c$, which we treat as
1086 a $(k-1)$-form rather than a $k$-form. Note that $\vartheta$ are not gauge invariant when $k > 1$, because
1087 they would create open charged objects, but we can combine them to construct the gauge-
1088 invariant degrees of freedom appearing in Eq. (99). For example, in the case $k = 2$, each $\vartheta$
1089 consists of $A$ on a boundary plaquette and $-\theta$ on the bulk link it touches, four of which combine
1090 to form the gauge-invariant composite object in Fig. 12(d). Let us denote the layer of bulk cells
1091 which touch the boundary layer, consisting of cells up to dimension $d-1$, as $Y$, illustrated in
1092 Fig. 11 by the dark layer between $X$ and $\partial X$. Each $(k+1)$-cell $c \in \partial X$ is associated to a unique
1093 $k$-cell $c_Y \in Y$. We treat $\vartheta$ as a $(k-1)$-form gauge field in $Y$, so that its exterior derivative is
1094 given by

$$
d\vartheta(c_Y \in Y_k) = (A_{\partial X} - \theta)(\partial c_Y) = (A - d\theta)(\partial c|_{\partial X}),
\tag{101}
$$

1095 where we identified $\partial c_Y$ with $\partial c|_{\partial X}$. We can then write the boundary action Eq. (99) as

$$
S_{\kappa\to\infty}^{\text{bdry}} = -\beta \sum_{c \in Y_{k-1}} \cos(d\vartheta).
\tag{102}
$$

1096 Equation (102) is precisely the action of a $(k-1)$-form U(1) gauge theory defined at the
1097 boundary of the system in terms of composite half-open Wilson operators. In the case $k = 1$ it
1098 reduces to the 0-form XY model identified in Section 2.

1099     We conclude that a $k$-form Abelian-Higgs model in $D$ dimensions Higgses in the $\kappa \to \infty$
1100 limit down to a $(k-1)$-form gauge theory on the $D-1$ dimensional boundary. If $D > k+2$
1101 ($k_m > 0$), this implies there will be a boundary phase transition. For $k = 1$ this will be in the
1102 $(D-1)$-dimensional XY universality class, with a symmetry-broken phase at large $\beta$. For $k > 1$
1103 it will be a confinement-deconfinement transition (expected to be first-order [7]), with a con-
1104 fined phase at small $\beta$ and a deconfined phase at large $\beta$. This reduces to the results reported
1105 in Section 2 for $k = 1$ and $D = 4$. The symmetry that is spontaneously broken at large $\beta$ is
1106 the $(k-1)$-form matter symmetry, corresponding to the electric symmetry of the boundary
1107 gauge theory when $k > 1$, under which $d\vartheta$ (as a half-open Wilson operator) is charged. A
1108 summary of this result is given in the table Table 1 for $k \leq 3$ and $2 \leq D \leq 6$, which may

1109 be extended straightforwardly to all $k$ and $D$. In the cases where $D = k + 2$ ($k_{\rm m} = 0$), a gen-
1110 eralized Mermin-Wagner theorem [5, 8] (equivalently, the Polyakov mechanism or magnetic
1111 instanton proliferation [66]) will prevent the symmetry from breaking. This may imply that
1112 the Higgs regimes of these marginal cases, which do not exhibit bulk higher-form symmetry
1113 breaking (deconfinement), are qualitatively distinct from the cases exhibiting boundary sym-
1114 metry breaking.

## 4.2 Electric-Magnetic Dual Picture

1116 We can gain further insight into these Abelian models and the boundary symmetry breaking
1117 by reformulating them in terms of dual magnetic variables. We will do so for gauge group
1118 U(1). The duality transformation is well-established in a variety of forms [31–34]. We derive
1119 it here in the presence of our open boundary conditions, which provides a concrete picture of
1120 both the electric and magnetic sectors of the theory near the boundary.

### 4.2.1 Open Boundary Duality Transformation

1122 The partition function of the original action for the $k$-form gauge theory is

$$Z = \int \mathcal{D}A \mathcal{D}\theta \, e^{-\beta \sum_{(X+\partial X)_{k+1}} \cos(\mathrm{d}A) - \kappa \sum_{X_k} \cos(\mathrm{d}\theta - A)}. \tag{103}$$

1123 This can be turned into a theory of electric strings and particles by utilizing the identity

$$e^{-x \cos(y)} = \sum_{n \in \mathbb{Z}} I_n(x) e^{iny},$$

1124 where $I_n$ are Bessel functions. We introduce integer $(k+1)$-form $e$ coupled to $\mathrm{d}A$ and $k$-form
1125 $j_{\rm e}$ coupled to $\mathrm{d}\theta - A$, to rewrite the partition function exactly as

$$Z = \int \mathcal{D}A \mathcal{D}\theta \sum_e \sum_{j_{\rm e}|_{\partial X}=0} I_e(\beta) I_{j_{\rm e}}(\kappa) e^{i\langle e, \mathrm{d}A \rangle + i\langle j_{\rm e}, \mathrm{d}\theta - A \rangle}. \tag{104}$$

1126 The restriction $j_{\rm e}|_{\partial X} = 0$ arises from the fact that we did not include half-open Wilson strings
1127 coupling the bulk to the vacuum. Utilizing the adjointness relation $\langle x, \mathrm{d}y \rangle = \langle \mathrm{d}^\dagger x, y \rangle$, this
1128 reduces to

$$\begin{aligned} Z &= \sum_e \sum_{j_{\rm e}|_\partial=0} I_e(\beta) I_{j_{\rm e}}(\kappa) \int \mathcal{D}A \, e^{i\langle \mathrm{d}^\dagger e - j_{\rm e}, A \rangle} \int \mathcal{D}\theta \, e^{i\langle \mathrm{d}^\dagger j_{\rm e}, \theta \rangle} \\ &= \sum_e \sum_{j_{\rm e}|_\partial=0} I_e(\beta) I_{j_{\rm e}}(\kappa) \, \delta(\mathrm{d}^\dagger e - j_{\rm e}) \, \delta(\mathrm{d}^\dagger j_{\rm e}). \end{aligned} \tag{105}$$

1129 We interpret the $k$-form $j_{\rm e}$ as the worldlines swept out by the $(k-1)$-dimensional Higgs ex-
1130 citations (carrying integer electric charge), and $e$ as the worldsheets swept out by the $k$-form
1131 electric field. The first delta function is just the Gauss law—it tells us that the worldsheets of
1132 electric flux must terminate on the worldlines of the Higgs charges. The second delta function
1133 enforces that the the worldlines $j_{\rm e}$ are "divergence-free," i.e. they form closed $k$-dimensional
1134 surfaces, corresponding to global conservation of charge. Note that, because the electric world-
1135 sheets can end, the electric 1-form symmetry $\mathrm{d}^\dagger e = 0$, present when $\kappa = 0$, is explicitly broken,
1136 though it can be restored as an emergent symmetry at energies below the charge gap.

1137 With the choice of "electric" boundary conditions (Fig. 2), we have the constraint $j_{\rm e} = 0$
1138 on all links extending from the bulk to the vacuum in Eq. (105), because there is no $\mathrm{d}\theta$ term
1139 for these links, while $e$ can be non-zero on the boundary plaquettes. This means that electric

charge cannot enter or exit the system, but electric flux can. The fluxes in the bulk form closed surfaces unless they terminate on charges. In the presence of the boundary these fluxes may be cut off at the boundary without terminating on charges.

We resolve the two constraints by writing[9]

$$j_e = d^\dagger h, \quad e = h + d^\dagger a_m, \tag{106}$$

for integer $(k+1)$-form $h$ and $(k+2)$-forms $a_m$, respectively. Note that these are not gauge-invariant and can be shifted by co-exact forms. We utilize the large-argument expansion of the Bessel functions,

$$I_n(z) \approx \frac{e^z}{\sqrt{2\pi z}} \left( 1 - \frac{4n^2 - 1}{8z} \right) \approx \frac{e^{z+1/8z}}{\sqrt{2\pi z}} e^{-n^2/2z},$$

to approximate the partition function by[10]

$$Z \approx \sum_{a_m} \sum_h e^{-\frac{(d^\dagger a_m + h)^2}{2\beta}} e^{-\frac{(d^\dagger h)^2}{2\kappa}} \delta(d^\dagger h|_{\partial X}), \tag{107}$$

where the delta function comes from the constraint $j_e|_\partial = 0$ and we dropped the prefactor. Finally, we apply Poisson resummation to turn these into real-valued fields,

$$\sum_{n \in \mathbb{Z}} f(n) = \int_{\mathbb{R}} dx \sum_{m \in \mathbb{Z}} f(x) e^{-2\pi i m x}.$$

We promote the integer fields to real fields,

$$h \to H \quad \text{and} \quad a_m \to A_m, \tag{108}$$

coupled respectively to integer currents $b$ and $j_m$ via Poisson resummation. Lastly, we move to the dual lattice, replacing $d^\dagger \alpha \leftrightarrow d\tilde{\alpha}$ for each $p$-form $\alpha$, where $\tilde{\alpha}$ is a $(D-p)$-form in the dual lattice, the discrete Hodge dual of $\alpha$ [34]. We thus obtain the dual partition function

$$Z_{\text{dual}} = \sum_{\tilde{j}_m, \tilde{b}} \int_{d\tilde{H}|_{\partial \tilde{X}} = 0} \mathcal{D}\tilde{H} \mathcal{D}\tilde{A}_m e^{-\frac{(d\tilde{A}_m + \tilde{H})^2}{2\beta} - \frac{(d\tilde{H})^2}{2\kappa}} e^{-i2\pi[\langle \tilde{A}_m, \tilde{j}_m \rangle + \langle \tilde{H}, \tilde{b} \rangle]}, \tag{109}$$

where $\tilde{j}_m$ and $\tilde{A}_m$ are $(D-(k+2))$-forms, while $\tilde{b}$ and $\tilde{H}$ are $(D-(k+1))$-forms, with the constraint that $d\tilde{H}$ is zero within the dual boundary layer coming from the delta function in Eq. (107). Finally, let us rescale the fields by absorbing the $2\pi$ into the definition of $\tilde{A}_m$ and $\tilde{H}$, to obtain the dual action

$$S_{\text{dual}} = -\frac{\beta'}{2}(d\tilde{A}_m + \tilde{H})^2 - \frac{\kappa'}{2}(d\tilde{H})^2 - i[\langle \tilde{A}_m, \tilde{j}_m \rangle + \langle \tilde{H}, \tilde{b} \rangle], \tag{110}$$

with dual couplings $\beta' = 1/4\pi^2\beta$ and $\kappa' = 1/4\pi^2\kappa$.

The currents $\tilde{j}_m$ and $\tilde{b}$ correspond to the winding defects of the U(1) gauge field $A$ and the Higgs field $\theta$, respectively. The former are the magnetic monopole worldlines, while the latter are the worldsheets of the vortices of the Higgs field. Under a gauge transformation of the $\mathbb{R}$-valued gauge fields $\tilde{A}_m \to \tilde{A}_m + \tilde{\lambda}$, $\tilde{H} \to \tilde{H} - d\tilde{\lambda}$, the action is shifted by

$$\delta S_{\text{dual}}^{\text{bulk}} = -i[\langle \tilde{\lambda}, \tilde{j}_m \rangle - \langle d\tilde{\lambda}, \tilde{b} \rangle] = -i\langle \tilde{\lambda}, \tilde{j}_m - d^\dagger \tilde{b} \rangle. \tag{111}$$

---

[9]Note that $j_e$ and $e$ can have a harmonic component corresponding to the electric winding sectors which we neglect. These give rise to ground state degeneracies in the topological Coulomb phase.

[10]We use the shorthand $(\omega)^2$ to denote $\langle \omega, \omega \rangle$.

|  | 1-Form U(1) Abelian-Higgs | 2-Form U(1) Abelian-Higgs | 3-form U(1) Abelian-Higgs |
|---|---|---|---|
| $D = 2+1$ | 2D 0-Form U(1) BKT transition $k_\mathrm{m} = 0$ | $k_\mathrm{m} = -1$ | $k_\mathrm{m} = -2$ |
| $D = 3+1$ | 3D 0-Form U(1) continuous transition $k_\mathrm{m} = 1$ | 3D 1-Form U(1) permanently confined $k_\mathrm{m} = 0$ | $k_\mathrm{m} = -1$ |
| $D = 4+1$ | 4D 0-Form U(1) continuous transition $k_\mathrm{m} = 2$ | 4D 1-Form U(1) (de)confinement transition $k_\mathrm{m} = 1$ | 4D 2-Form U(1) permanently confined $k_\mathrm{m} = 0$ |
| $D = 5+1$ | 5D 0-Form U(1) continuous transition $k_\mathrm{m} = 3$ | 5D 1-Form U(1) (de)confinement transition $k_\mathrm{m} = 2$ | 5D 2-Form U(1) (de)confinement transition $k_\mathrm{m} = 1$ |

Table 1: The boundary theories of $k$-form U(1) Abelian-Higgs models in $D$ spacetime dimensions in the $\kappa \to \infty$ limit and their phase transitions. The pattern is that a $k$-form Abelian Higgs model Higgses down to a $(k-1)$-form gauge theory on the boundary without matter. The number $k_\mathrm{m} = D - (k+2)$ indicates the dimension of the magnetic worldlines of the bulk theory, $\tilde{j}_\mathrm{m}$ in Eq. (109). Note that in the $\beta \to \infty$ limit the bulk has a corresponding $k_\mathrm{m}$-form symmetry. Gray boxes indicates cases where the gauge field has no dynamics. The top non-trivial box of each column, which has $D = k+2$ ($k_\mathrm{m} = 0$), are affected by magnetic instanton proliferation (the Polyakov mechanism or a generalized Mermin-Wagner theorem forbidding boundary symmetry breaking) and do not exhibit a boundary phase transition, except in the case $k = 1$ in $D = 3$, which can exhibit a BKT transition.

If we integrate over all gauges, i.e. over all the generators $\tilde{\lambda}$, we obtain delta functions that yield the constraint

$$\tilde{j}_\mathrm{m} = \mathrm{d}^\dagger \tilde{b}. \tag{112}$$

This says that the magnetic monopole worldlines form the boundaries of the Higgs vortex worldsheets. This is precisely the magnetic Gauss law, i.e. it says that magnetic monopoles are the sources of magnetic strings (compare to the electric Gauss law Eq. (105)). This is familiar from superconductor phenomenology—vortex cores carry magnetic flux. It also follows from this that $\mathrm{d}^\dagger \tilde{j}_\mathrm{m} = 0$, i.e. that the magnetic charge worldlines are closed and magnetic charge is conserved, which follows from the "gauge-of-gauge" invariance $\tilde{A}_\mathrm{m} \to \tilde{A}_\mathrm{m} + \mathrm{d}\tilde{\alpha}$ for arbitrary $\tilde{\alpha}$.

This theory can be recast as a U(1) gauge theory as follows. Summing over $\tilde{b}$ undoes one of the Poisson resummations and forces $\tilde{H} = 2\pi\tilde{h}$, where the integer field $\tilde{h}$ is the Hodge dual of $h$ in Eq. (107). The residual gauge symmetry is $\tilde{A}_\mathrm{m} \to \tilde{A}_\mathrm{m} + 2\pi\tilde{l}$ and $\tilde{H} \to \tilde{H} - 2\pi\mathrm{d}\tilde{l}$, where $\tilde{l}$ is an integer shift, meaning that the gauge-invariant configuration space for $\tilde{A}_\mathrm{m}$ is actually $\mathbb{R}/2\pi\mathbb{Z}$ and for $\tilde{H}$ is $2\pi\mathbb{Z}$. The resulting theory is therefore a Villainized $k_\mathrm{m}$-form U(1) gauge theory [23,69] (Eq. (86)) coupled to magnetic currents, the dual of the original Abelian Higgs model model, Eq. (103), which was coupled to electric currents. The integer gauge field $\tilde{h}$ measures the winding numbers of the compact $\tilde{A}_\mathrm{m}$, and its fluxes $\mathrm{d}\tilde{H}$ are the homotopy defects of $\tilde{A}_\mathrm{m}$, which are the electric charges of the original theory. They act as sources for the fluxes $\mathrm{d}\tilde{A}_\mathrm{m}$, which are the electric strings in the direct lattice.

Let us briefly review how the dual bulk behaves in the various limits in the Maxwell case,

1183  $D = 4$ and $k = 1$. First, consider the $\beta \to \infty$ ($\beta' \to 0$) limit: in the electric formulation the
1184  gauge field is turned off, $dA = 0$, and the remaining Higgs sector is a gauged 4$D$ XY model.
1185  In the dual theory the magnetic charges are turned off (integrating $\tilde{A}_m$ sets $\tilde{j}_m = 0$) and the
1186  magnetic 1-form symmetry is restored, resulting in a gas of closed membranes interacting via
1187  their coupling to the 2-form gauge field. Performing the Gaussian integration over $\tilde{H}$ we obtain
1188  the action $(\kappa'/2)\langle \omega, (d^\dagger d)^{-1} \omega \rangle$, i.e. the gauge field $\tilde{H}$ generates Coulomb interactions among
1189  the membranes.

1190      Next, consider the $\kappa \to 0$ ($\kappa' \to \infty$) limit. Electric charges are turned off, restoring the
1191  electric 1-form symmetry and reducing to a pure U(1) gauge theory. In the dual theory
1192  the gauge field $\tilde{H}$ is turned off, $d\tilde{H} = 0$. The theory reduces to a Coulomb gas of magnetic
1193  monopole worldlines [31, 66], which has a phase transition separating the deconfined phase
1194  (low temperature condensate, 1-form symmetry spontaneously broken) and the confined phase
1195  (high temperature gas, 1-form symmetry unbroken).

1196      Lastly, consider the $\beta \to 0$ ($\beta' \to \infty$) limit, the strong coupling limit of the original theory.
1197  In the electric theory, we can fix unitary gauge to remove $\theta$ and obtain the action $-\kappa \cos(A)$ on
1198  every link independently, so the system is fully disordered. Equivalently, if we use Eq. (105),
1199  setting $\beta = 0$ forces all of the Bessel functions to vanish except when $e = j_e = 0$, which reduces
1200  the partition function to $\prod_\ell I_0(\kappa)$. In the dual theory, $\beta' \to \infty$, the weak coupling limit, and
1201  we have the constraint $\tilde{H} = -d\tilde{A}_m$, which further implies $d\tilde{H} = 0$. The resulting theory then
1202  just has Lagrange multipliers that force the charge loops and membranes to vanish, so the
1203  theory trivializes completely.

1204  **4.2.2  Dual Boundary Symmetry Breaking**

1205  We now consider setting $\kappa' = 0$ in Eq. (110) ($\kappa \to \infty$). The variables in play are $\tilde{A}_m$ and $\tilde{j}_m$,
1206  defined on all $k_m$-cells of the dual lattice, and $\tilde{H}$ and $\tilde{b}$ on all $(k_m + 1)$-cells. In the bulk, the
1207  $\kappa' = 0$ means that the kinetic term for $\tilde{H}$ drops out and so the fluxes of $\tilde{H}$ (corresponding to
1208  electric charges of the original action) are completely unconstrained. In other words, the bulk
1209  $\tilde{H}$ is in the strong coupling limit. We may integrate out $\tilde{H}$, to obtain

$$S_{\text{dual}}^{\text{bulk}} = -\frac{1}{2\beta'}\langle \tilde{b}, \tilde{b} \rangle - i\langle \tilde{A}_m, (\tilde{j}_m - d^\dagger \tilde{b})|_{\tilde{X}} \rangle. \tag{113}$$

1210  For a closed system, we can integrate out $\tilde{A}_m$ and express the bulk partition function in the
1211  form

$$Z_{\text{dual}}^{\text{bulk}} \xrightarrow{\kappa \to \infty} \sum_{\tilde{b}} \sum_{\tilde{j}_m} e^{-\langle \tilde{b}, \tilde{b} \rangle/2\beta'} \delta(d^\dagger \tilde{b} - \tilde{j}_m). \tag{114}$$

1212  The sum is over all possible configurations of the (open or closed) worldsheets $\tilde{b}$ with a bare
1213  surface tension $1/\beta' \propto \beta$. An intuitive picture in the Maxwell case, $k = 1$ and $D = 4$, is that
1214  in a time slice this corresponds to magnetic monopole pairs attached by a magnetic string with
1215  linearly rising potential, i.e. the magnetic charges are confined in this limit, as expected for a
1216  bulk electric condensate which collimates the magnetic field into flux tubes. The characteristic
1217  size ("Debye" screening length) of the neutral monopole pairs tends to zero as $\beta$ tends to $\infty$.
1218  Alternatively, we may view this as monopole strings (worldlines) $\tilde{j}_m$ interacting electrostati-
1219  cally through the membranes of the $b$ field. For large $\beta$ the membranes are short, meaning
1220  that the strings are bound into charge-neutral pairs. This phase persists to all $\beta$ because the
1221  entropic gain of dipole strings outweighs their energetic cost at all effective temperatures.

1222      Now consider an open boundary. Recall that the boundary of the dual lattice is "flat", as
1223  shown in Fig. 11, i.e. it has no cells extending into the vacuum. This means that no magnetic
1224  charge or magnetic flux can exit the system. More concretely, the constraint $j_e|_{\partial X} = 0$, in
1225  Eq. (104), in the electric variables is reflected in the dual constraint $d\tilde{H}|_{\partial \tilde{X}} = 0$, which enforces

that $\tilde{H}$ is pure gauge in the dual boundary layer. This means that the boundary action has no $\kappa$ dependence (it effectively has $\kappa = 0$). We resolve the boundary constraint as $\tilde{H} = \mathrm{d}\tilde{f}$, so that

$$S_{\mathrm{dual}}^{\mathrm{bdry}} = -\frac{\beta'}{2}(\mathrm{d}(\tilde{A}_{\mathrm{m}} + \tilde{f}))^2 - i\langle\tilde{A}_{\mathrm{m}}, \tilde{j}_{\mathrm{m}}|_{\partial\tilde{X}}\rangle - i\langle\mathrm{d}\tilde{f}, b|_{\partial\tilde{X}}\rangle \tag{115}$$

We then define a composite field $\tilde{\chi} = \tilde{A}_{\mathrm{m}} + \tilde{f}$, which is gauge invariant under the gauge transformations $\tilde{A}_{\mathrm{m}} \to \tilde{A}_{\mathrm{m}} + \tilde{\lambda}$ and $f \to \tilde{f} - \tilde{\lambda}$. This allows us to rewrite the boundary action as

$$S_{\mathrm{dual}}^{\mathrm{bdry}} = -\frac{\beta'}{2}(\mathrm{d}\tilde{\chi})^2 - i\langle\tilde{\chi}, \mathrm{d}^\dagger\tilde{b}|_{\partial\tilde{X}}\rangle - i\langle\tilde{A}_{\mathrm{m}}, (\tilde{j}_{\mathrm{m}} - \mathrm{d}^\dagger\tilde{b})|_{\partial\tilde{X}}\rangle. \tag{116}$$

To proceed from here we need to be careful about how the $\tilde{j}_{\mathrm{m}}$ and $\tilde{b}$ can move between the bulk and boundary layers. We do so in the $\kappa \to \infty$ limit, by combining this with the bulk action Eq. (113). We can then integrate out $\tilde{A}_{\mathrm{m}}$ to generate the magnetic Gauss law, which is enforced on every link. This leaves us with the total action

$$S_{\mathrm{dual}} \xrightarrow{\kappa\to\infty} -\frac{1}{2\beta'}\langle\tilde{b}, \tilde{b}\rangle_{\tilde{X}} - \frac{\beta'}{2}(\mathrm{d}\tilde{\chi})^2_{\partial\tilde{X}} - i\langle\tilde{\chi}, \tilde{j}_{\mathrm{m}}|_{\partial\tilde{X}}\rangle. \tag{117}$$

The boundary portion describes monopoles moving in the boundary layer interacting via a non-compact gauge field.

One should be concerned here as to how the $\tilde{j}_{\mathrm{m}}$ from the bulk (edges of $\tilde{b}$) couple to the boundary. The key to understand what happens here is that (i) the $\tilde{b}$ membranes can lie in the boundary layer where they cost zero action, and (ii) for this action to be gauge-invariant under shifts of $\tilde{\chi}$, we must have an additional boundary Gauss law,

$$\mathrm{d}^\dagger\tilde{j}_{\mathrm{m}}|_{\partial\tilde{X}} = 0 \tag{118}$$

This implies that, in the limit $\kappa \to \infty$, *magnetic monopoles cannot move between the boundary and the bulk*, i.e. there is a $(k_{\mathrm{m}} - 1)$-form symmetry on the boundary (0-form in the Maxwell case) corresponding to conservation of boundary magnetic charge. The action Eq. (117) is precisely the dual of the $\kappa = \infty$ boundary $(k-1)$-form U(1) gauge theory Eq. (102), a 3D XY model in the Maxwell case.

The $\tilde{j}_{\mathrm{m}}$ in the boundary layer must be coupled to $\tilde{b}$ membranes by the magnetic Gauss law, but these have zero tension if they lie entirely within the boundary layer. This means that the Higgs vortices (magnetic field lines) are effectively not present within the boundary layer. This was by construction, since there were no $\mathrm{d}\theta$ terms in the original action which would allow for vortices of the Higgs field within the boundary layer. As a result, in the partition function to leading order the boundary and bulk are effectively decoupled from each other. Given any configuration of monopole worldlines $\tilde{j}_{\mathrm{m}}$, which are closed in the boundary and closed in the bulk, the dominant contribution to the partition function will be for the boundary $\tilde{j}_{\mathrm{m}}$ to be connected to tensionless membranes in the boundary, rather than to be connected to a bulk monopole by a tensionful one. As a result, the monopoles on the boundary can condense at small $\beta$, which is the dual to the boundary symmetry breaking transition we found in the electric formulation.

One can then consider turning on small $\kappa'$ and doing a strong coupling expansion. This will have the effect of renormalizing the bulk length scale enabling boundary monopoles to extend further into the bulk by extending a magnetic flux tube while preserving the quasi-$(D-1)$-dimensional nature of the boundary.

## 4.3  Summary and Discussion: Higher-Form Case

In this section, we have generalized our results from Section 2 on 1-form Abelian-Higgs models to higher form Abelian-Higgs models in $D$ spacetime dimensions. In particular, we found

that $k$-form gauge field coupled to a $(k-1)$-form Higgs field reduces at infinite $\kappa$ to a $(k-1)$-form gauge theory on the boundary whose dynamical degrees of freedom are half-open Wilson branes terminating on the Higgs field as soon as they enter the bulk. This boundary theory exhibits a confinement-deconfinement transition when $D > k + 2$ ($k_{\rm m} > 0$), which spontaneously breaks the $(k-1)$-form global matter symmetry at large $\beta$. Table 1 summarizes this pattern of boundary symmetry breaking. In the marginal cases, $D = k + 2$ ($k_{\rm m} = 0$), a generalized Mermin-Wagner theorem prevents the symmetry from breaking, except in the case $D = 3$ and $k = 1$, where we predict a BKT boundary transition. There are also precisely the cases where the same mechanism destabilizes the deconfined phase in the bulk [63, 64]. As in the 1-form Abelian and non-Abelian cases studied numerically in this paper, we expect that this boundary phase transition extends into the phase diagram when bulk fluctuations are restored and will end at a bulk critical point, demarcating a boundary between the Higgs and confining regimes.

The general mechanism for this emergent boundary theory at large $\kappa$ identified by considering higher-form Abelian-Higgs models revolves around the constraint $A = {\rm d}\theta$ enforced exactly at infinite $\kappa$. This constraint implies that the Wilson operators which create electric charges attached to electric membranes, $\exp[i({\rm d}\hat{\theta} - \hat{A})(M)]$ for open surfaces $M$, act as the identity. This naïvely indicates that the system is a condensate of electric charge in this limit, i.e. the ground state is a coherent state of the charge annihilation operators. As a consequence, the electric-brane insertion operators $\exp[i\hat{A}(M)]$ for closed surfaces $M$ trivialize in the bulk. In the presence of electric-flux-permeable open boundaries, however, operators inserting electric flux through the boundary are immediately screened as soon as they enter the bulk, terminating on the Higgs field, as shown in Fig. 12, and become charged under the matter symmetry. These operators are the dynamical degrees of freedom at play at the boundary which exhibit the matter symmetry breaking. Presumably when $\kappa$ is reduced the electric flux can penetrate further into the system before being screened by the electric charge condensate, forming a quasi-$(D-1)$-dimensional boundary. It would be interesting to explore whether this mechanism can be extended to higher-form non-Abelian theories described by higher-categorical gauge groups [70–74].

In the second part of this section, we studied the dualized version of these theories, identifying the boundary degrees of freedom and accounting, in the 1-form case and $D = 4$, for the existence of the dual to the 3D XY model that we found in terms of direct variables. Magnetic charge moves in the boundary layer, and there is an extra Gauss law in the $\kappa \to \infty$ limit which originates from the constraint that electric charge cannot leave the system. This enforces that magnetic charge cannot leave the boundary into the bulk, and thus the charges on the boundary can condense, leading to the dual phase transition. It would be of interest to study the dual theory in more detail, since it gives a clearer picture of the boundary symmetry breaking in the large-$\kappa$ (small $\kappa'$) limit. In particular, it would be worthwhile to pursue Monte Carlo simulations in the dual representation, for which efficient algorithms have been developed [75–77]. It is also worth re-emphasising the importance of our choice of boundary conditions, which prevented electric charge from leaving the system while allowing electric flux to leave. In the dual theory this led to a flat dual boundary, meaning magnetic charge and flux is always contained in the system and cannot leave. It follows that if one started with flat boundaries, which keep all electric charge and flux inside the system, the dual boundaries would be open, i.e. magnetic charges are kept in the system but magnetic flux can leave. This implies a physical magnetic charge symmetry which can spontaneously break in the confined regime rather than the Higgs regime, with a phase transition on the $\beta = 0$ axis instead of the $\kappa' = 0$ axis.[11]

---

[11]This was also discussed in the $\mathbb{Z}_2$ 1-form case in [22].

# 5  Discussion and Conclusion

## 5.1  Summary and Outlook

In this work we have explored a variety of models of charged Higgs fields coupled to gauge fields in the fundamental representation, under the imposition of boundary conditions which allow electric flux, but not charge, to exit the system. In a closed system, the gauge field does not have a physical global charge symmetry which can spontaneously break, because a charge is always attached to electric flux, which must end inside the system on another charge. With the "electric-flux-permeable" boundary conditions we consider, the charge sectors and global symmetry become physical, as non-zero bulk charge can be compensated by non-zero boundary flux, and can in principle spontaneously break. This work, focusing on the boundary degrees of freedom, complements work exploring the interplay of global and gauge symmetries in the bulk. See for example recent work in Refs. [78–82]. More broadly, the boundary perspective offers novel insight into the gauge-invariant description of Higgs phases and how they might be distinguished from confined phases, issues with a long history [13–15, 25, 26] which continue to generate interest to the present day [17–23, 37, 83–85].

We have considered the Abelian-Higgs model, two types of non-Abelian Higgs models (with fundamental representation and group-valued Higgs fields), and higher-form Abelian-Higgs models. In terms of the inverse gauge coupling $\beta$ and the matter coupling $\kappa$, all of these models share the common feature of a continuity between an electric-charge confining regime at small $\beta, \kappa$, and a Higgs regime at large $\beta, \kappa$, which are two ends of one continuous thermodynamic phase, as proven by Fradkin and Shenker [14]. We have demonstrated through a combination of analytical argument and numerical investigation that, under all but a few marginal cases, there is a boundary phase transition which indicates the spontaneous breaking of the matter symmetry at large $\beta, \kappa$, i.e. in the Higgs regime, as illustrated in Fig. 1. Already in the seminal work of Fradkin and Shenker [14] it was understood that in gauge theories with fundamental Higgs matter, under most circumstances thermodynamic quantities exhibit no singularities along paths between Higgs and confined regimes, meaning that they form the same bulk phase of matter. The boundary spontaneous symmetry breaking investigated in this paper does not contradict this result, because it does not define a precise bulk phase boundary between the Higgs and confined regimes. Indeed, as we observed, one may tune the boundary coupling while preserving all the bulk properties to shift the location of the boundary phase transition.

Ref. [22] predicted a boundary phase transition in the case of gauge group $\mathbb{Z}_2$, and verified it using DMRG in $D = 2+1$ dimensions. Ref. [23] predicted the boundary transition for a magnetic monopole-free Abelian U(1) Higgs model without numerics. Using lattice gauge theory Monte Carlo simulations, we have explored a wider range of models, both for Abelian and non-Abelian gauge groups. In Section 2, we have numerically verified and studied the boundary transition for gauge group U(1) (with monopoles) in $D = 3+1$ dimensions, which exhibits a boundary XY transition. In Section 3, we extended this to non-Abelian gauge groups, performing numerical simulations for both SU(2) and SU(3) gauge theories coupled to fundamental Higgs fields in $D = 3+1$, and considered generalizations to SU($N$), SO($N$), and general gauge groups. Lastly, in Section 4 we studied higher-form generalizations of Abelian-Higgs models, and demonstrated that the corresponding higher-form matter symmetry can spontaneously break at the boundary. In all of our numerical simulations, the nature of the boundary phase transitions deduced from the numerical data conforms to predictions obtained by studying the $\kappa \to \infty$ limit.

We expect the considerations in this work to extend further to all gauge groups and different Higgs representations [52]. Notably we have not discussed higher-rank gauge theories coupled to scalar matter that are connected to fractonic excitations [86–91], but here too one

may preserve the Gauss law for tensor fields on the boundary and expect a U(1) global symmetry that, for certain classes of such models, can be broken spontaneously. It would also be of interest to extend these results to discrete non-Abelian gauge groups, and non-Abelian higher-form gauge theories with higher-categorical gauge groups [71–74].

Before continuing, we comment briefly on some interesting connections between our work and two other active areas of research: boundary criticality and asymptotic symmetries. "Traditional" boundary criticality has a long history [92,93], and has recently received some new insights [94,95]. The central idea is that the boundary of a system may undergo a phase transition in a novel "extraordinary" universality class when coupled to a critical bulk, by tuning the boundary coupling relative to the bulk. In this work we have only considered the behavior of the boundary when the bulk is gapped, but it would be interesting to explore the boundary criticality in more detail in the vicinity of the bulk critical point where the transition appears to end for $\alpha \geq 1$ (cf. Figs. 4 and 8), especially at the lower critical dimension [94]. Prior boundary criticality studies have focused on criticality from the spontaneously breaking of 0-form symmetries, but our work raises the possibility of novel boundary physics arising from dynamical gauge fields and higher-form symmetries. On the one hand, such exploration may yield novel boundary universality classes, and on the other it may yield some new insights into the behavior of these enigmatic bulk critical points [96,97].

Another active area with some overlap with this work is the exploration of asymptotic symmetries in gauge theories and gravity on the boundary of compactified spacetime [98–107]. For example, in pure electromagnetism in Minkowski spacetime at null infinity it is well known that there is an infinite dimensional symmetry generated by large gauge transformations with conserved charges living on the boundary [100]. Indeed, Ref. [103] has already made some direct connections between asymptotic (or "long-range gauge") symmetries and the boundary symmetries studied in this work. It would be interesting to deepen the connections between Higgs boundary criticality on the lattice and asymptotic symmetries in the continuum.

## 5.2 Higgs = SPT

It would be remiss of us to conclude this paper without a description of some of the work that motivated this study—namely the papers [21–23] discussing the relationship of Higgs phases to symmetry-protected topological (SPT) phases. We briefly summarize the main findings therein and comment on how our results bear on the generality of this relationship. For more recent works related to this topic see Refs. [108–110]. An SPT phase is a state of matter that cannot be adiabatically connected to a trivial phase under local symmetry-preserving perturbations without closing a gap. In general, a trivial phase can be reached without gap closure only if those symmetries are broken. A classic example of an SPT phase in an interacting lattice model is the spin one-half chain—the cluster model—whose nontrivial topology is protected by $\mathbb{Z}_2 \times \mathbb{Z}_2$ symmetry [111,112]. A consequence of the topological nature of the phase is the presence of gapless states localized at the boundary of an open chain.

### 5.2.1 Review of Higgs=SPT

The "Higgs=SPT" connection was first raised in Ref. [21], which considered gauging the $\mathbb{Z}_2$ fermion parity symmetry of the Kitaev chain, which hosts an SPT phase. Gauging trivializes the symmetry in the bulk, but by the same arguments we have discussed throughout this paper, it is instead realized non-trivially on the boundary. In the Higgs regime a fermionic SPT order emerges (that belongs to the same class as a stack of two Kitaev chains) protected by the $\mathbb{Z}_2$ boundary fermion parity symmetry and the $\mathbb{Z}_2$ 0-form magnetic symmetry of the gauge field. Allowing instantons (magnetic tunneling events) explicitly breaks the 0-form magnetic symmetry and kills the SPT. In the similar spirit, it was argued that by gauging the $\mathbb{Z}_2$ parity

symmetry of the transverse field Ising model, in the Higgs phase one ends up with the cluster SPT order.

These 1D results were extended in Ref. [22] to the Higgs phases of $\mathbb{Z}_2$ gauge theory with matter in general dimensions with electric boundary conditions (Fig. 2). The phase diagram is qualitatively the same as Fig. 1, with (within our notation) the deconfined phase at large $\beta$ and small $\kappa$ exhibiting $\mathbb{Z}_2$ toric code topological order (i.e. 1-form symmetry breaking) [22,28,97]. The $\mathbb{Z}_2$ magnetic symmetry is a $(d-1)$-form symmetry.[12] The global $\mathbb{Z}_2$ 0-form Ising matter symmetry is trivial in the bulk but is realized at the boundary. It was shown that on the line $\beta = \infty$ and in the large-$\kappa$ Higgs limit there is a cluster SPT order, protected by the $\mathbb{Z}_2$ higher-form magnetic symmetry and the matter symmetry, hosting gapless surface states and a boundary ground state degeneracy.[13] Going to finite $\beta$ explicitly breaks the magnetic 1-form symmetry, but as long as the magnetic symmetry is higher-form it is expected to survive in the infrared as a low-energy emergent symmetry, because the magnetic defects are gapped at large $\beta$.[14] Moreover, it was shown that in the $\kappa = \infty$ limit the boundary degrees of freedom are governed by an Ising model charged under the global matter symmetry, which is in the spontaneously broken phase for large $\beta$. The boundary ground state degeneracy of the putative $\beta = \infty$ SPT were directly identified with the Ising degeneracy of this phase. The boundary symmetry breaking was verified numerically with a phase diagram qualitatively the same as Fig. 1, and within the resolution of the DMRG numerics of Ref. [22] the phase boundary to the boundary gapless phase extends from the critical endpoint. However, one cannot simply identify the boundary-symmetry-broken regime with a bulk SPT, indeed Ref. [22] observed that the position of the transition depends on microscopic details near the boundary (such as the deformation $\alpha$ which we have considered). Instead it was argued that certain properties of SPT (such as edge modes and degeneracies of the entanglement spectrum) should be stable in the Higgs regime in an open region of the phase diagram in the vicinity of the $\beta \to \infty$ limit.[15] In particular, the mutual anomaly between the matter and magnetic symmetries at the boundary is robust, and correspondingly the edge mode degeneracies should be as well until a gap is closed.

These ideas were extended in Ref. [23], which considers U(1) Higgs models with *exact* magnetic symmetry, i.e. *without* magnetic monopoles. Starting from the continuum theory, Eq. (1), the mutual anomaly can be exposed by coupling to background gauge fields for both symmetries—1-form $A_{\text{mat}}$ for the matter symmetry and $(d-1)$-form $B_{\text{mag}}$ for the magnetic symmetry. Deep in the Higgs regime, the bulk topological response was found to be

$$S_{\text{SPT}} = \frac{1}{2\pi} \int B_{\text{mag}} \wedge dA_{\text{mat}}. \tag{119}$$

For a system with open boundaries this SPT response is not gauge invariant without adding appropriate boundary degrees of freedom which cancel the anomaly. For the $3+1$D Higgs phase, the boundary theory takes the form

$$S_{\text{Boundary}} = \frac{1}{2\pi} \int d\varphi \wedge d\vartheta \tag{120}$$

---

[12]Note that the magnetic symmetry is $(d-1)$-form for discrete gauge group and $(d-2)$-form for U(1), where $d$ is the dimension of space.

[13]For earlier work on SPTs protected by generalized symmetries see for example Refs. [74,113]

[14]The magnetic symmetry is equivalent to the closure of magnetic flux surfaces. The presence of magnetic defects allows the fluxes to end without closing, and thus explicitly breaks the symmetry. At large $\beta$ the magnetic defects will have a large gap, and therefore at low energies magnetic fluxes will again be closed. Thus the symmetry is expected to be emergent in the infrared [114]. Indeed this argument explains why the deconfined phase, which spontaneously breaks the higher-form electric and magnetic symmetries, is stable in the presence of gapped magnetic and electric charges [4].

[15]The entanglement spectrum in this model was investigated recently in [115].

for the conjugate pair: a compact scalar field $\varphi$ and a compact $U(1)$ gauge field $\vartheta$. This describes a boundary $U(1)$ superfluid phase, which has the correct mixed anomaly to cure the bulk one [6]. This continuum discussion was supplemented with a lattice Villain formulation, where monopoles are under control and the magnetic 1-form symmetry is preserved.

As it stands, we can summarize the Higgs=SPT situation as follows. For Abelian gauge fields coupled to fundamental matter with *exact* magnetic symmetry, the matter symmetry is realized at the boundary and shares a mutual anomaly with the bulk magnetic symmetry. The bulk of the system can be identified as an SPT protected by these two symmetries, and the SPT boundary modes originate from the spontaneously broken matter symmetry: for $\mathbb{Z}_N$ gauge group these are robust boundary discrete degeneracies, and for U(1) gauge group these are gapless edge modes identified with the boundary Goldstone modes. The authors of both Ref. [22] and Ref. [23] claim that some aspects of the SPT, in particular the edge modes and entanglement spectrum degeneracies, should be robust to explicit breaking of magnetic symmetry (when it is a higher-form symmetry), even if the bulk is no longer strictly an SPT (and can indeed be continuously connected to the trivial confined phase).

### 5.2.2 Insights, Challenges, and Future Directions

Our work potentially extends the Higgs=SPT story, but also provides new challenges for it. Firstly, the general Higgs=SPT arguments generalize directly to the higher-form Abelian models we considered in Section 4 if magnetic symmetry is enforced exactly. Secondly, we can say that spontaneous breaking of global charge conservation symmetry at the boundary is a universal feature of the Higgs regime, and that this statement applies to both Abelian and non-Abelian models. This could indicate that the Higgs-SPT relation holds in a much broader set of cases than originally envisioned. However, it may also be that the boundary symmetry breaking is a broader phenomenon than the Higgs=SPT relation, which happens to coincide with the SPT edge modes when the relation holds.

We wish to point out an open issue for the Higgs=SPT connection with continuous gauge groups in the models we have considered here. In the Abelian-Higgs model, when the gauge group is discrete the gapped Higgs phase is continuously connected to the $\beta = \infty$ limit where the magnetic symmetry is exact and the ground state is an SPT [22]. However, there is a subtlety about the large-$\beta$ limit in the U(1) case: the $\beta = \infty$ limit of the Higgs regime is gapless, namely it is gauge-equivalent to a bulk superfluid. Therefore, if there is an SPT in the U(1) Fradkin-Shenker model at $\beta = \infty$ where the magnetic symmetry is exact it must be gapless, and is therefore not adiabatically connected to the gapped Higgs phase when the gauge coupling is turned on.[16] Furthermore, the limiting point $\beta = \kappa = \infty$ in the phase diagram is problematic, because the gap scales as $\kappa/\beta$ as one approaches it [14]: approaching from $\beta = \infty$ the bulk is gapless, while approaching from $\kappa = \infty$ the bulk is gapped.[17] It is therefore unclear that the boundary edge modes of the gapped Higgs regime discussed in Section 2 and Section 4, particularly in the $\kappa = \infty$ limit, can be identified with those of a putative $\beta = \infty$ SPT. Clearly this same issue holds for any continuous gauge group, in particular to the non-Abelian cases studied in Section 3.

The physics of the non-Abelian gauge groups gives rise to a further puzzle. In Section 3 we have argued that in the limit $\beta, \kappa \gg 1$ the boundary is generically expected to be gapless since it breaks a continuous symmetry. If the Higgs=SPT scenario is general enough to encompass the non-Abelian cases too, one should be able to identify the protecting (higher-form) bulk "magnetic" symmetry. Pure SU($N$) gauge theory has no magnetic symmetry since the funda-

---

[16]This same problem with the $\beta = \infty$ limit is also present in the Villainized model studied by Ref. [23] (in contrast to the Wilson model we have studied), but in that model one can effectively tune the magnetic monopole mass to infinity while holding $\beta$ finite to obtain an exact magnetic symmetry.

[17]This limit was key to establishing the fixed-point SPT Hamiltonian in the $\mathbb{Z}_2$ case in Ref. [22].

mental group is trivial, and it is unclear whether there is a generalized symmetry present at $\beta = \infty$ for non-Abelian gauge groups that is an analog of the Abelian magnetic symmetry.

Let us take a broader perspective and focus on the mutual anomaly which is key to the Higgs=SPT phenomenology. The core idea in the Abelian theories is as follows. Given an Abelian $k$-form symmetry $\mathcal{G}$ there is a corresponding "dual" $(d-k-n)$-form $\mathcal{G}$ symmetry corresponding to suppressing defects in the ordered phase, where $n = 0$ (1) if $\mathcal{G}$ is discrete (continuous), and these share a mutual anomaly [22, 23, 114].[18] When the symmetry is gauged, it is realized as a $k$-form symmetry on the $(d-1)$-dimensional boundary, while the bulk has a new $((d-1)-k-n)$-form magnetic symmetry, which we have emphasized has exactly the right degree to play the role of the dual symmetry on the boundary, and indeed shares the correct mutual anomaly [22, 23]. Some generalization of this mechanism should likely also hold true for gauging 0-form non-Abelian symmetries and spontaneously breaking them at the boundary. In general the emergent symmetries arising from spontaneously breaking a 0-form $\mathcal{G}$ symmetry do not form a group but rather a symmetry category, and these symmetries share a mixed anomaly with $\mathcal{G}$ [114, 116]. Upon gauging the 0-form $\mathcal{G}$ symmetry, forcing it to the boundary, one expects that the resulting gauge field has a "magnetic" symmetry category which preserves the correct anomaly structure at the boundary. The precise mathematical structure of the resulting mutual anomaly after gauging is likely complicated due to the fact that the boundary symmetry is generated by the *total* charge (gauge plus matter, cf. Section 3.1.3), rather than just the charge of the Higgs field. However, we believe that identifying the relevant generalized symmetries of the non-Abelian gauge field and matching the boundary anomaly are the required ingredients to extend the Higgs-SPT connection to the non-Abelian setting.

# Acknowledgements

PM and PR acknowledge useful discussions with Pedro Bicudo and Nuno Cardoso as a part of a related collaboration. KTKC acknowledges Chris Hooley for useful discussion.

**Funding information** S.M. is supported by Vetenskapsrådet (grant number 2021-03685), Nordita and STINT. This work was in part supported by the Deutsche Forschungsgemeinschaft under the cluster of excellence ct.qmat (EXC-2147, project number 390858490).

# A Discrete Differential Calculus for Abelian Fields

We utilize notation which mimics continuum differential forms on a lattice, borrowed from algebraic topology [117] and used extensively in Section 4. Pedagogical treatments can be found in [34, 56]. Fields are described (locally) as differential forms (i.e. anti-symmetric tensors), whose primary property is that they can be integrated over surfaces. For our purposes we can think of them simply as functions of such surfaces, i.e. if $\omega$ is a differential $k$-form and $U$ an oriented $k$-dimensional surface, $\omega$ acts on $U$ by integration

$$\omega(U) := \int_U \omega, \tag{A.1}$$

---

[18]For example: a 0-form $\mathbb{Z}_n$ symmetry has a corresponding $d$-form $\mathbb{Z}_n$ symmetry associated to suppression of domains walls (conservation of symmetry breaking sector); a 0-form U(1) symmetry has a corresponding $(d-1)$-form U(1) symmetry associated to suppression of vortices (conservation of winding number); and a 1-form U(1) symmetry has a corresponding $(d-2)$-form U(1) symmetry associated to suppression of monopoles (i.e. magnetic symmetry, conservation of magnetic flux).

which results in some number. There are two main properties that we wish to preserve on the lattice: reversing the orientation of $U$ changes the sign of the integral, and if $U$ is divided into a collection of smaller parts the total integral is the sum of the integrals over the parts, i.e.

$$\omega(-U) = -\omega(U),$$
$$\omega(U_1 + U_2) = \omega(U_1) + \omega(U_2), \tag{A.2}$$

where $-U$ denotes the reversed orientation. Lastly, we can naturally define the derivative of a $k$-form $\omega$ to be a $(k+1)$-form $\mathrm{d}\omega$ whose value on a $(k+1)$-dimensional surface $U$ is defined by Stoke's theorem,

$$\int_U \mathrm{d}\omega := \int_{\partial U} \omega, \tag{A.3}$$

where $\partial U$ denotes the boundary of $U$.

In this paper, our space(time) is a $D$-dimensional cubical cell complex—a collection of $k$-cells for $k = 0, \ldots, D$, i.e. vertices, links, plaquettes, cubes, hypercubes, etc., where $k$-cells are glued along their $(k-1)$-dimensional boundary cells. We denote the collection of all cells by $X$ and the collection of all $k$-cells by $X_k$. Because we also consider open boundaries in the form of Fig. 2, we let $X_k$ denote just the bulk $k$-cells and $\partial X_k$ denote the $k$-cells in the boundary layer which touch the vacuum.

The cells naturally provide the integration surfaces once equipped with an orientation. It is natural to define the possible integration surfaces therefore as integer weighted linear combinations of oriented $k$-cells, call $k$-chains. The signs of the integer coefficients determine the orientations and their magnitudes determine how many times to integrate over each cell. Since we can formally add such chains together, they form an Abelian group, $\mathcal{C}_k$. The structure of the cell complex is contained in the boundary relation,

$$\partial : \mathcal{C}_k \to \mathcal{C}_{k-1}, \tag{A.4}$$

which distributes over the linear combinations on $k$-cells, and sends each oriented $k$-cell to the linear combination of its oriented boundary $(k-1)$-cells.

The sensible lattice analog of a differential form, i.e. a discrete $k$-form, also called a $k$-cochain, is a function which (i) maps chains to numbers, Eq. (A.1), and (ii) disributes over linear combinations, Eq. (A.2). In full generality, a discrete $k$-form $\omega$ is a linear map from chains to elements of any Abelian group $\mathcal{G}$,

$$\omega : \mathcal{C}_k \to \mathcal{G}. \tag{A.5}$$

In practice what this means is that a disrete $k$-form is defined by its value on each oriented $k$-cell $c$, and satsfies $\omega(-c) = -\omega(c)$, where $-\omega$ is understood as the inverse operation in the Abelian group $\mathcal{G}$. The space of discrete $k$-forms is denoted $\mathcal{C}^k(\mathcal{G})$. Since we already have a natural notion of the boundary operation on chains, we can define a natural exterior derivative operation on cochains, $\mathrm{d} : \mathcal{C}^k \to \mathcal{C}^{k+1}$, according to Stoke's theorem, i.e.

$$\mathrm{d}\omega(U) := \omega(\partial U), \tag{A.6}$$

where $\omega$ is a $k$-form, $\mathrm{d}\omega$ is a $(k+1)$-form, and $U$ is a $(k+1)$-chain.

Denoting the coefficients in a $k$-chain $u$ by

$$u = \sum_{c \in X_k} u_k c \quad (u_k \in \mathbb{Z}), \tag{A.7}$$

where each $k$-cell is summed once with a fixed orientation, we can define a natural inner product on $k$-chains as

$$(u, w) = \sum_c u_c w_c. \tag{A.8}$$

1558  Using this, we can define an adjoint of the boundary operator, which we call the coboundary,[19]

$$(u, \partial v) = (\partial^\dagger u, v). \tag{A.9}$$

1559  In particular, the coboundary of a single $k$-cell is

$$\partial^\dagger c = \sum_{c' \in X_{k+1}} (\partial^\dagger c)_{c'} c' = \sum_{c' \in X_{k+1}} (\partial^\dagger c, c') c' = \sum_{c' \in X_{k+1}} (c, \partial c') c' \tag{A.10}$$

1560  By choosing to orient all the $c'$ in the sum such that $(c, \partial c') = +1$ or 0, we can read this to
1561  say that *the coboundary of an oriented $k$-cell $c$ is the sum of all oriented $(k+1)$-cells containing*
1562  $+c$ *in their positively oriented boundary*. For example, the coboundary of a point $i$ is the set
1563  of oriented links which terminate at it, the coboundary of a link $\ell$ is the set of plaquettes $p$
1564  touching it, oriented so that $\partial p$ circulates in the same direction as $\ell$ is oriented, etc.

1565    The couboundary defines a co-exterior derivative, via a "co-Stoke's theorem",

$$d^\dagger \omega(c) := \omega(\partial^\dagger c), \tag{A.11}$$

1566  which reduces the degree of forms. In the case that $\mathcal{G}$ is $\mathbb{Z}$, $\mathbb{R}$, or $\mathbb{C}$, we can also define an
1567  inner product for $k$-forms,

$$\langle \alpha, \beta \rangle = \sum_{c \in X_k} \alpha(c)^* \beta(c) \tag{A.12}$$

1568  where $*$ denotes complex conjugation. It is easy to see that the codifferential is adjoint to the
1569  differential,

$$\begin{aligned}
\langle \alpha, d\beta \rangle &= \sum_{c \in X_k} \alpha(c)^* \beta(\partial c) = \sum_c \alpha(c)^* \sum_{c' \in \partial c} \beta(c') \\
&= \sum_{\substack{c \in X_k \\ c' \in X_{k-1}}} \alpha(c)^* \beta(c')(\partial c, c') = \sum_{\substack{c \in X_k \\ c' \in X_{k-1}}} \alpha(c)^* \beta(c')(c, \partial^\dagger c) \\
&= \sum_{c' \in X_{k-1}} \alpha(\partial^\dagger c')^* \beta(c') = \langle d^\dagger \alpha, \beta \rangle. 
\end{aligned} \tag{A.13}$$

1570  Qualitatively, one may think of the differential as a generalized gradient, describing how a $k$-
1571  form varies across $(k+1)$-cells, and the codifferential as a generalized divergence, describing
1572  how a $k$-form "flows into" $(k-1)$-cells.

1573    Lastly, we review a bit of basic algebraic topology terminology used in Section 4. The
1574  boundary operation is nilpotent, $\partial^2 = 0$, and thus defines an exact sequence of maps,

$$0 \to \mathcal{C}_D \xrightarrow{\partial_D} \cdots \mathcal{C}_1 \xrightarrow{\partial_1} \mathcal{C}_0 \to 0. \tag{A.14}$$

1575  We can define the homology groups, $H_k := \ker \partial_k / \operatorname{im} \partial_{k+1}$. The interpretation of these quotient
1576  groups is that they classify the non-contractible $k$-dimensional surfaces. Here, $\ker \partial$ is gener-
1577  ated by the set of surfaces without boundaries (called cycles), while $\operatorname{im} \partial$ is generated by the
1578  set of surfaces which are boundaries of a $(k+1)$-dimensional volume, and are therefore con-
1579  tractible. The elements of $H_k$ are equivalence classes of surfaces which differ by a boundary,
1580  i.e. homology classes.

1581    The dual of the homology classes on the differential form side are the cohomology classes,
1582  which are defined by the discrete equivalent of the de Rham complex:

$$0 \to \mathcal{C}^0 \xrightarrow{d_0} \mathcal{C}^1 \xrightarrow{d_1} \cdots \xrightarrow{d_{D-1}} \mathcal{C}^D \to 0, \tag{A.15}$$

---

[19]Note that this differs from the algebraic topology terminology, where the discrete exterior derivative is often called the coboundary.

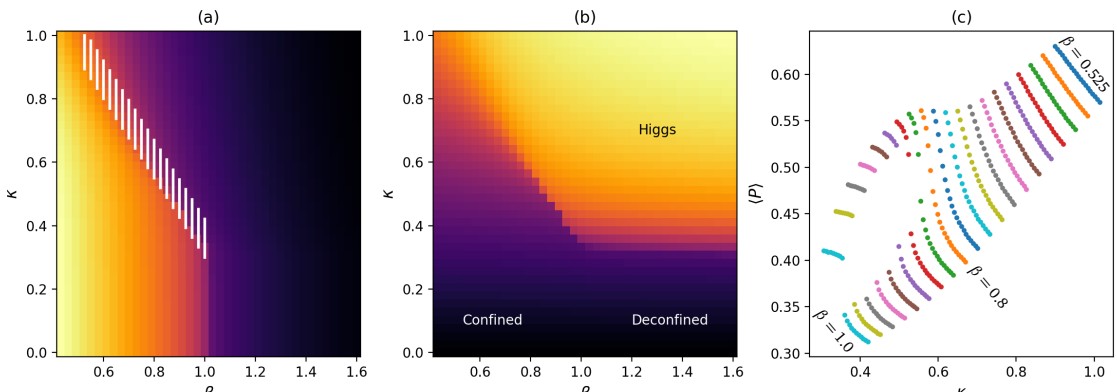

Figure 13: (a,b) Phase diagram of the 4D $U(1)$ Abelian Higgs model for $L = 16$ revealed through (a) expectation of the minimal Wilson loop, $\langle 1 - \text{Re}\, W_p \rangle$, and (b) expectation of the Wilson link, $\langle \text{Re}\, \Lambda_\ell \rangle$, averaged over all plaquettes and links, respectively. The color ranges from 0.1 (black) to 0.8 (yellow). (c) Cuts of $\langle 1 - \text{Re}\, W_p \rangle$, at constant $\beta$ for different values of $\beta$ from 0.525 to 1.000 in steps of 0.025 from right to left. The system is isotropic with $L = 16$ and 50000 sweeps. The data shows a clear sign of a phase transition for larger $\beta$ and a crossover for smaller $\beta$. The data is consistent with the presence of a first order transition with a critical endpoint at about $\beta = 0.85$.

as $H^k := \ker \mathrm{d}_k / \operatorname{im} \mathrm{d}_{k-1}$. Here $\ker \mathrm{d}_k$ is the set of locally-constant $k$-forms (called closed forms), and $\operatorname{im} \mathrm{d}_{k-1}$ is the set of $k$-forms which are gradients of $(k-1)$-forms (called exact forms). Every exact form is closed, but there can exist closed forms which are not exact, which are classified by cohomology. Cohomology classes are equivalence classes of closed forms which differ by an exact form, just as homology classes are equivalence classes of closed surfaces which differ by a boundary. Note that exact forms integrate to zero by Stoke's theorem on any closed surface, while closed forms need only integrate to zero on closed surfaces that are boundaries. Forms with non-trivial cohomology class integrate non-trivially on non-contractible surfaces, defining a pairing between cohomology and homology classes.

# B  U(1) Bulk Phase Diagram: Limits, Symmetries and Monte Carlo

Here we provide a brief review of the bulk phase diagram of the 4D $U(1)$ Abelian-Higgs model, described by the action Eq. (8), the Hamiltonian Eq. (10), and dual action Eq. (110), by considering the various limits and undertaking numerical Monte Carlo simulation in the action formulation. The phase diagram is sketched in Fig. 1, which conveniently summarizes the discussion that follows. In particular, it is well-known that there are only two distinct phases [14], the Coulomb phase and the Higgs-confined phase. According to Eq. (104), this is a theory of electric strings terminating on electric point charges. The dual magnetic description is in terms of magnetic strings terminating on magnetic point charges (vortices and monopoles of the Higgs and gauge fields, respectively). The basic structure of the phase diagram can be deduced by consider each of the following four limits.

*Pure gauge limit* ($\kappa \ll 1$): In the limit $\kappa = 0$ the gap of the electric point charges diverges, and the theory reduces to 4D $U(1)$ gauge theory. In this limit, the system has a global 1-form symmetry, $A \to A + \lambda$ with $\mathrm{d}\lambda = 0$, called electric symmetry. This symmetry corresponds to electric strings forming closed loops, i.e. electric flux through closed surfaces is conserved. This theory has two phases, a confined phase at small $\beta$ (strong coupling), where the sys-

tem is gapped and electric strings cost energy proportional to their length; and a deconfined phase at large $\beta$ (weak coupling), where the electric strings condense, the electric symmetry is spontaneously broken, and the system has a gapless photon excitation. Turning on a small $\kappa$ explicitly breaks the 1-form symmetry by introducing gapped electric point charges at which open electric strings end. Because the charges are strongly gapped, qualitatively speaking we expect the 1-form symmetry to re-emerge at low energies below the charge gap, thus allowing the deconfined phase to extend to finite $\kappa$.

*Frozen gauge limit* ($\beta \to \infty$): In this limit the gauge fields are completely trivialized by the constraint $dA = 0$, and equivalently the mass of the magnetic monopoles diverges. We can choose a gauge where $A = 0$ and the action turns into that of a 4D XY model, whose dual description is a gas of vortex strings with Coulomb interactions. This theory has a 1-form symmetry corresponding to the closure of these magnetic strings and the corresponding absence of magnetic monopoles, called magnetic symmetry. From the XY model we deduce a gapless superfluid phase at large $\kappa$ and a gapped phase at small $\kappa$ separated by a second-order phase transition, but this description is not gauge-invariant. The gauge-invariant statement is that at small $\kappa$ the magnetic strings (disorder operators o the XY model) condense, spontaneously breaking the magnetic symmetry. For large but finite $\beta$ the magnetic monopoles explicitly break the magnetic symmetry, though one expects it to be effectively restored at low energies below the monopole gap. The result is that the magnetic symmetry is spontaneously broken in the gapless Coulomb phase, while the superfluid at large $\kappa$ is gapped out by the Higgs mechanism.

*Strong coupling (confining) limit* ($\beta = 0$): In this limit the curvature of the gauge field $A$ is not penalized, and we can say that the magnetic monopoles are maximally proliferated. If we fix to unitary gauge ($\theta = $ const.), the action becomes $-\kappa \sum_\ell \cos(A_\ell)$, reducing to a set of completely disconnected link variables. Thus the system is in a trivial phase for all $\kappa$, and there is no bulk phase transition on this line. Strong coupling expansion (in $\beta$) demonstrates that when $\kappa = 0$ Wilson loops have an area law, indicating confinement of static charges by a tensionful electric string. This confinement extends to positive $\kappa$, though particle-anti-particle nucleation cuts off the area law of Wilson loops for large loops. Confinement may still be observed using the Fredenhagen-Marcu order parameter, however [27, 28].

*Infinite Higgs coupling limit* ($\kappa \to \infty$): In this limit we have the constraint $A = d\theta$. This implies that the electric charge creation and annihilation operators, Eq. (14), act as the identity, and the system can be described as an electric condensate. The bulk has no dynamics and is completely frozen, which can be seen by fixing to unitary gauge ($\theta = $ const.), in which $A = 0$ and the Higgs field is frozen. There is therefore no bulk phase transition along this line. For large but finite $\kappa$ the bulk action can be roughly understood as the proca-type action, Eq. (23), describing a massive 1-form field, at least for large $\beta$.

This completes the general outline of the phase diagram in the vicinity of the edges in Fig. 1. The only question that remains is how the two transitions at small $\kappa$ and large $\beta$ reach each other to separate the Coulomb phase (in which the electric and magnetic symmetries are emergent and spontaneously broken) from the Higgs-confined phase. Figure 13 shows results from Monte Carlo simulations measuring the average Wilson plaquette and Wilson link, which fills in the remainder of the phase diagram schematically shown in Fig. 1. The two transitions extend as first-order transition lines and meet at a triple point in the vicinity of $\beta \sim 1.0$ an $\kappa \sim 0.4$. A third first order line extends from the this triple point towards smaller $\beta$ and larger $\kappa$ which ends at a critical endpoint. We show evidence for this first-order line in Fig. 13(c).

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
