# Peer review of "Higgs Phases and Boundary Criticality"

_SciPost Physics_

## Round 1 · Referee Report · Anonymous (Referee 1) · 2024-11-16

Report

This work studies the phase diagram of various gauge theories, focusing in particular on a gapped-phase region which contains two qualitatively different sub-regions --- the Higgs sub-region and the confined sub-region. It is believed that there is no thermodynamic transition between these two subregions, but a phase transition can occur at the boundary of open geometries. The authors use analytical arguments and numerical evidence for the ubiquity of such a boundary phase transition in various cases: 4D U(1) theory, non-Abelian theories, and higher-form theories.

I expect this paper to be a reasonably smooth read for experts in the field, and thank the authors for being careful with their writing. I did not check the work carefully, but see no reason that the paper is wittingly incorrect. This is a solid and thorough paper, and I have only two relatively minor concerns.

1). The authors should provide a Zenodo or Github link to their numerics for others to check and learn.

2). The work could use a bit more context. In particular, the authors could expand the readership if they could link this direction to the use of boundaries in topological error-correcting codes. Could there be any important consequences of this boundary transition in general, and for quantum information processing specifically? Answering this question is not a requirement for acceptance, just something I hope the authors can think about more. I list potentially related papers, part of which discuss boundaries of various 4D Z2 theories, and none of which are required to be cited:
https://arxiv.org/abs/1805.01836
https://arxiv.org/abs/2110.14644
https://arxiv.org/abs/2111.12096
https://arxiv.org/abs/2208.07367
https://arxiv.org/abs/2310.16982
https://arxiv.org/abs/2405.11719
https://arxiv.org/abs/2407.07951

Recommendation

Publish (meets expectations and criteria for this Journal)

  • validity: -
  • significance: -
  • originality: -
  • clarity: -
  • formatting: -
  • grammar: -

Author:  Kristian Chung  on 2025-08-15  [id 5730]

(in reply to Report 1 on 2024-11-16)
Category:
answer to question

We thank the reviewer for a positive report and recommendation to publish our paper. Here we address the points raised by the referee.

1) While we support open sharing of data and code, the Monte Carlo codes used for this project are not well-documented and commented in a form appropriate for posting in a public repository, however we are happy to make the code and data available upon request. We add an availability statement as a comment on the latest arXiv submission as well as at the end of the main text.

2) The suggested papers are very interesting but only tangentially related to the topics we discuss, which focus on continuous Lie group gauge theories—U(1) and SU(N) in particular—whereas error-correcting codes rely on discrete degrees of freedom. It would be very interesting to study the analogous Higgs-SPT relation for discrete non-Abelian groups and relate our results to quantum double models, for instance, but this is beyond the scope of this work. For this reason we prefer not to cite them in this paper, but thank the referee for pointing out this interesting avenue for future work.

---

## Round 1 · Referee Report · Anonymous (Referee 2) · 2025-2-7

Strengths

  1. Studies a recently popular idea of "Higgs=SPT", the idea that Higgs phases of lattice gauge theories can be thought of as a symmetry protected topological phase. The authors carefully study more general cases which are relevant for both condensed matter and high energy physics.

  2. The main technical contributions include its generalization (from previously studied Z_N and U(1) gauge groups) to non-abelian gauge groups and higher-form gauge theories, where the analysis is more complicated. In every case, they find that boundary phase transition that exists at the "frozen bulk" limit persists when one includes bulk fluctuations, which is a nontrivial result.

  3. The numerical analysis (especially for $\beta_\text{bdry}/\beta_{\text{bulk}} \neq 1$ in Fig. 4, 8) is very useful in understanding the relevance of boundary phase transitions as a probe of bulk phenomena. It is difficult to make analytical statements about the termination of the phase boundary, and the numerics convincingly shows that it moves away from the tricritical point in a peculiar way. Given that the nature of this tricritical point is a popular puzzle and not well-understood, this is useful insight and may spark further studies.

  4. The paper is well-written and easy to follow.

Weaknesses

  1. Much of this paper is a careful and structured generalization of previous work on Higgs=SPT, and the results are what one would naively expect. (Although expected, they are certainly nontrivial)

Report

Although the results are expected from previous work, this work is a valuable contribution to the field. The extensive analytical and numerical analysis provides a good starting point for further study, and the paper is very well-written and useful for physicists interested in the subject.

It would be nice if the authors commented about the following:

  1. In condensed matter, we often have gauge theories which are emergent. For example, the Z2 toric code can be thought as equivalent to Z2 gauge theory with matter. In this general context, is it natural to have the specific boundary conditions that realize the boundary phase transition (allows electric flux to pass while conserving electric charge)? In the language of a tensor product Hilbert space, what are the relevant symmetries for this phase transition to exist?

  2. Perhaps a naive question. Why is $\alpha \equiv \beta_\text{bdry}/\beta_{\text{bulk}} = 1$ special in all the numerics that you do, as in it seems to follow the first order coexistence line (or the "bulk rapid crossover" line) starting from the bulk critical point? From a coarse-graining RG perspective we would expect this $\alpha$ to get renormalized as we go to the IR. Is there any subtlety at $\alpha = 1$ that is preventing this?

Requested changes

If relevant, the authors should add some comments about the prior questions in the paper.

Recommendation

Publish (meets expectations and criteria for this Journal)

  • validity: high
  • significance: high
  • originality: ok
  • clarity: top
  • formatting: perfect
  • grammar: perfect

Author:  Kristian Chung  on 2025-08-15  [id 5731]

(in reply to Report 2 on 2025-02-07)
Category:
remark
answer to question

We thank the referee for their positive report and recommendation to publish. We wish to first address the referee’s concern regarding “weakness” of the work, and then respond to the referee’s individual questions below.

“Much of this paper is a careful and structured generalization of previous work on Higgs=SPT, and the results are what one would naively expect. (Although expected, they are certainly nontrivial)”

We would like to make a point of clarification regarding the above assessment. While our results in the Abelian case do align well with the established Higgs=SPT framework—the Higgs phase exhibits boundary criticality which may be associated to the interpretation of the bulk as an SPT—our study of non-Abelian Higgs models raises new challenges. In particular, the Higgs=SPT idea for Abelian groups rests on a mutual anomaly between the bulk matter symmetry, realized as a boundary symmetry due to the Gauss law, and the bulk magnetic symmetry which emerges in the large-beta limit. Identifying the relevant generalized magnetic symmetries of the non-Abelian gauge field and matching the boundary anomaly are the required missing ingredients to extend the Higgs-SPT idea to the non-Abelian setting. We discuss this in more detail in Sec. 5.2.2.

“In condensed matter, we often have gauge theories which are emergent. For example, the Z2 toric code can be thought as equivalent to Z2 gauge theory with matter. In this general context, is it natural to have the specific boundary conditions that realize the boundary phase transition (allows electric flux to pass while conserving electric charge)? In the language of a tensor product Hilbert space, what are the relevant symmetries for this phase transition to exist?”

In the physical Hilbert space—which indeed has a tensor product structure—the boundary conditions are dictated by the nature of mutually-anomalous symmetries. Specifically, the transition relies on the interplay between a zero-form symmetry associated with conservation of matter and a higher-form symmetry associated with conservation of magnetic flux. Crucially, the phase transition is highly sensitive to the zero-form symmetry: even a weak explicit breaking of this symmetry will destroy the transition. In contrast, it remains robust under a weak breaking of the higher-form magnetic symmetry. For the Z_2 case, these aspects are analyzed in detail in Sections 4.4.4 and 4.4.5 of arXiv: 2211.01376.

“Perhaps a naive question. Why is α=1 special in all the numerics that you do, as in it seems to follow the first order coexistence line (or the "bulk rapid crossover" line) starting from the bulk critical point? From a coarse-graining RG perspective we would expect this α to get renormalized as we go to the IR. Is there any subtlety at α=1 that is preventing this?”

This is an excellent question. The numerics points to a system-size independent correspondence between the bulk critical point(s) and the boundary phase transition line for alpha=1. When alpha departs from one, the boundary transition drifts away from the bulk line. However, we do not have a sharp understanding of any renormalization of alpha and why alpha=1 might be protected. We would be glad to see this question answered.

---

## Editorial Decision

resubmitted